# Negatives-Dominant Contrastive Learning for Generalization in Imbalanced Domains

Meng Cao [1 2]   Jiexi Liu [3]   Songcan Chen [1 2]

## Abstract

Imbalanced Domain Generalization (IDG) focuses on mitigating both *domain and label shifts*, both of which fundamentally shape the model's decision boundaries, particularly under heterogeneous long-tailed distributions across domains. Despite its practical significance, it remains underexplored, primarily due to the *technical* complexity of handling their entanglement and the paucity of *theoretical* foundations. In this paper, we begin by *theoretically* establishing the generalization bound for IDG, highlighting the role of posterior discrepancy and decision margin. This bound motivates us to focus on directly steering decision boundaries, marking a clear departure from existing methods. Then, we *technically* propose a novel Negative-Dominant Contrastive Learning (NDCL) for IDG to enhance discriminability while enforce posterior consistency across domains. Specifically, inter-class decision-boundary separation is enhanced by placing greater emphasis on negatives as the primary signal in our contrastive learning, naturally amplifying gradient signals for minority classes to avoid the decision boundary being biased toward majority classes. Intra-class compactness is encouraged through a re-weighted cross-entropy strategy, and posterior consistency across domains is enforced through a prediction-central alignment strategy. Finally, rigorous yet challenging experiments on benchmarks validate the effectiveness of our NDCL. The code is available at https://github.com/Alrash/NDCL.

[1]College of Computer Science and Technology, Nanjing University of Aeronautics and Astronautics, Nanjing, China [2]MIIT Key Laboratory of Pattern Analysis and Machine Intelligence, Nanjing, China [3]School of Computer and Artificial Intelligence, Nanjing University of Finance and Economics, Nanjing, China. Correspondence to: Songcan Chen <s.chen@nuaa.edu.cn>.

*Proceedings of the $43^{rd}$ International Conference on Machine Learning*, Seoul, South Korea. PMLR 306, 2026. Copyright 2026 by the author(s).

## 1. Introduction

Real-world applications are often affected by domain shifts (Krueger et al., 2021), caused by factors such as weather variations (Eastwood et al., 2022) or stylistic changes (Nam et al., 2021), leading to a decline in model generalization performance. To handle this issue, Domain Generalization (DG) has been developed over the past decade (Wang et al., 2022), aiming to learn a model on source domains that generalizes effectively to diverse unseen target domains. Nevertheless, most existing DG methods primarily tackle covariate shift (Lin et al., 2022), where $P^i(\boldsymbol{X}) \neq P^j(\boldsymbol{X}), \forall i \neq j$, while largely neglecting label shift (Cao et al., 2019), where $P^i(\boldsymbol{Y}) \neq P^j(\boldsymbol{Y}), \forall i \neq j$, which frequently arises during real-world data collection. For instance, label shift can occur when rare hematological cell types are significantly underrepresented in samples from certain medical institutions (Umer et al., 2023), resulting in imbalanced class distributions across domains. This highlights the practical significance of Imbalanced Domain Generalization (IDG) (Yang et al., 2022), where label shift often manifests as heterogeneous long-tailed distributions across domains.

Due to the inherent difficulty of handling both entangled shifts, IDG has received limited attention and remains underexplored. This coupling leads to several additional challenges: 1) Label shift, especially under heterogeneous long-tailed class distributions, can compromise the effectiveness of alignment strategies that are primarily designed to mitigate domain shift. For instance, if pneumonia is more prevalent in Hospital A while cardiomegaly dominates in Hospital B, domain alignment strategies (Arjovsky et al., 2019) alone may inadvertently match features of these distinct conditions across domains, i.e., effectively aligning pneumonia with cardiomegaly, leading to poor generalization. 2) Domains with abundant samples tend to dominate the training process, effectively absorbing underrepresented domains and suppressing their unique characteristics. This imbalance biases the learned representation space and harms generalization to both source minorities and unseen target domains. Corresponding empirical analyses on toy datasets are presented in the Appendix E.2.

While prior work proposes transferability-based surrogates

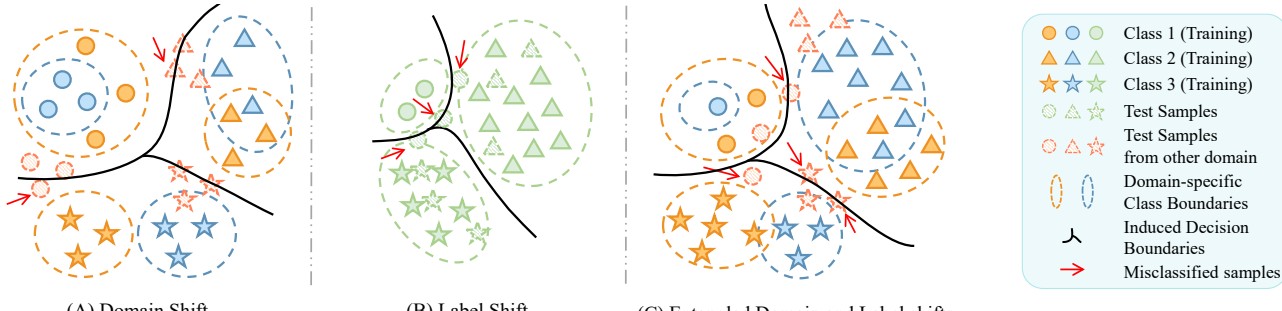

(A) Domain Shift        (B) Label Shift        (C) Entangled Domain and Label shifts

*Figure 1.* Impact of Domain and Label Shifts on Decision Boundaries. Orange, blue, and peach represent three distinct domains. (A) Domain shift leads to domain-specific class boundaries (dashed lines) for each class. (B) Label shift causes majority classes to dominate, compressing margins for minority classes. The dashed and solid lines are equivalent, with the former included for visual consistency. (C) Their interaction amplifies the challenge, requiring robustness to both simultaneously.

(Yang et al., 2022) to heuristically guide representation learning in IDG, it lacks formal generalization guarantees for unseen target domains. To fill this gap, in this paper, we establish the first theoretical generalization bound for IDG based on $\mathcal{H}$-divergence (Ben-David et al., 2010), where posterior discrepancy and decision margin play pivotal roles alongside prior discrepancy. Crucially, departing from conventional domain generalization theory (Albuquerque et al., 2019; Lu et al., 2024), which primarily focuses on aligning prior distributions $P(X)$ across domains, IDG further demands posterior $P(Y|X)$ consistency across domains and well-separated decision boundaries between neighboring classes simultaneously. Meanwhile, in essence, domain shift challenges the model to synthesize a unified decision boundary across heterogeneous domains (Zhu et al., 2022), as shown in Fig. 1 (A), while label shift pulls minority class boundaries toward majority classes due to imbalance-induced bias (Kang et al., 2020), as shown in Fig. 1 (B). Fig. 1 (C) illustrates that when both shifts co-occur, the model's inability to disentangle their respective effects becomes the underlying cause of severely distorted decision boundaries. This theoretical insight and the essential factors motivate us to directly reshape the model's decision boundaries to be domain-consistent and margin-enlarged, thereby enhancing robustness in IDG.

To this end, we propose a novel contrastive learning method at the prediction space for IDG, namely Negative-Dominant Contrastive Learning (NDCL). We observe that the complement of each class, i.e., what it's not (Malisiewicz et al., 2011) or its negative samples, is especially abundant for minority classes. Crucially, as illustrated in Fig. 2, these negatives remain relatively balanced between classes, providing a perspective to *naturally alleviate class imbalance*. In addition, prior studies (Kalantidis et al., 2020; Malisiewicz et al., 2011) have empirically demonstrated that such negatives exhibit favorable transferability across domains. Buliding on these insights, we advocate leveraging the natural structure of the data itself, rather than relying on explicit re-weighting

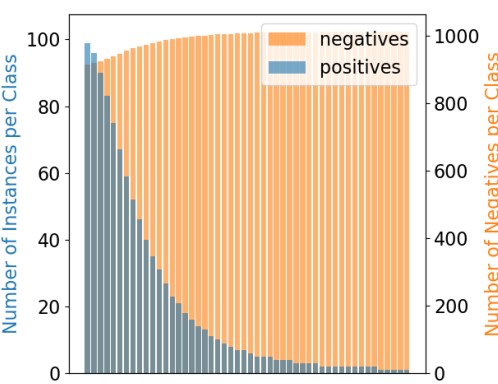

*Figure 2.* Long-Tailed Positives vs. Relatively Balanced yet Abundant Negatives.

(Yang et al., 2022) or re-sampling (Xia et al., 2023) strategies, to shape class boundaries. This offers a unified and principled method to simultaneously mitigate both types of shifts, fundamentally differing from prior works (Su et al., 2024; Xia et al., 2023; Yang et al., 2022) in IDG. Specifically, inter-class decision-boundary separation is enhanced by treating negatives as the primary signal in our contrastive learning, where the InfoNCE objective (Khosla et al., 2020) is reformulated by replacing similarity measure with cosine dissimilarity and positioning negative pairs in the numerator. To reinforce decision-boundary separation, a hard negative augmentation strategy has been employed instead of randomly sampling from all negatives. Meanwhile, intra-class compactness is encouraged via a cross-entropy loss that adaptively re-weights samples within each class, and posterior consistency across domains is enforced through a prediction-central alignment strategy.

In summary, our main contributions can be highlighted as follows:

- Theoretically, a generalization bound for IDG is provided based on $\mathcal{H}$-divergence, highlighting the role of

posterior discrepancy and decision margin.

- Methodologically, a novel Negative-Dominant Contrastive Learning (NDCL) is provided to enhance discriminability while enforce posterior consistency across domains.

- Empirically, the effectiveness of our NDCLis validated through rigorous yet challenging experiments under various forms of imbalance on domain generalization benchmarks.

## 2. Methodology

In this section, a more detailed and rigorous discussion will be presented, including both theoretical formulation and practical methodology.

### 2.1. Preliminaries

In DG, it is commonly assumed that each domain follows a joint distribution $P(\boldsymbol{X}, \boldsymbol{Y})$ (Wang et al., 2022), where $\boldsymbol{X}$ denotes input instances and $\boldsymbol{Y}$ denotes their corresponding labels. Under this assumption, domain shift (Krueger et al., 2021) can be regarded as the discrepancy between these joint distributions, i.e., $P^d(\boldsymbol{X}, \boldsymbol{Y}) \neq P^{d'}(\boldsymbol{X}, \boldsymbol{Y}), \forall d \neq d'$. Given $N$ domains, each domain can sample $N_d$ data, which can be defined as $\left\{\left(\boldsymbol{x}_i^d, y_i^d\right)\right\}_{i=1}^{N_d}$. While existing DG methods typically assume that the marginal label distribution $P(\boldsymbol{Y})$ is shared across domains and focus mainly on the domain shift in the instance space $P(\boldsymbol{X})$, our work explicitly considers the case where both $P(\boldsymbol{X})$ and $P(\boldsymbol{Y})$ may vary across domains. Despite this, the label space remains consistent across domains, containing $K$ semantic classes as in standard DG settings.

For notational simplicity, let $\boldsymbol{X}_k^d = \left\{\left(\boldsymbol{x}_d^k\right)_i\right\}_{i=1}^{n_k^d}$ denote the set of samples from the $k$-th class within the $d$-th domain. Accordingly, the aggregated sample set of the $k$-th class across all $N$ observed domains is then given by $\boldsymbol{X}_k = \bigcup_{d=1}^N \boldsymbol{X}_k^d = \left\{(\boldsymbol{x}_k)_i\right\}_{i=1}^{n_k}$, where $n_k = \sum_{d=1}^N n_k^d$. Let $f_\theta : \mathcal{X} \to \mathcal{Y}_E$ be a mapping from the instance space $\mathcal{X}$ to the encoded label space $\mathcal{Y}_E$, and denote $\boldsymbol{p} = f_\theta(\cdot)$ as the corresponding output coding vector.

### 2.2. Theoretical Guarantee

Building on the definition of a domain as a joint distribution (Wang et al., 2022) and leveraging the $\mathcal{H}$-divergence (Ben-David et al., 2010), we obtain as the follows:

**Theorem 2.1** (Generalization Bound for Imbalanced Domain Generalization). *Given $N$ domains with joint distributions $\left\{P_S^d(\boldsymbol{X}, \boldsymbol{Y})\right\}_{d=1}^N$, the target risk $\epsilon_T(h)$ for any hypothesis $h \in \mathcal{H}$ is defined via their linear mixture. To capture prediction-induced discrepancies, the classical $\mathcal{H}$-divergence is reformulated over mappings $m = yh(x)$ :*

$\boldsymbol{X} \times \boldsymbol{Y} \to \{-1, 1\} \in \mathcal{M}$, *which is an auxiliary class.*

$$
\begin{aligned}
\epsilon_T(h) \leq &\sum_{d=1}^N \pi_d \epsilon_S^d(h) + \lambda_\pi + \underbrace{\ell_{max} \cdot \Pr[\gamma(h) \leq \delta]}_{\text{margin}} \\
&+ \zeta + \underbrace{d_{\mathcal{M} \triangle \mathcal{M}}^{\sum_{d=1}^N \pi_d P_S^d(\boldsymbol{Y}|\boldsymbol{X})}\left(\sum_{d=1}^N \pi_d P_S^d(\boldsymbol{X}), P_T(\boldsymbol{X})\right)}_{\text{prior distribution discrepancy}} \\
&+ \eta + \underbrace{\mathbb{E}_{x \sim P_T(\boldsymbol{X})} d_{\mathcal{M} \triangle \mathcal{M}}\left(\sum_{d=1}^N \pi_d P_S^d(\boldsymbol{Y}|\boldsymbol{X}=x), P_T(\boldsymbol{Y}|\boldsymbol{X}=x)\right)}_{\text{posterior distribution discrepancy}}
\end{aligned}
$$

*where $\sum_{d=1}^N \pi_d = 1$, and $\sum_{d=1}^N \pi_d P_S^d(\boldsymbol{X}, \boldsymbol{Y})$ denotes the closest projection of the target distribution $P_T(\boldsymbol{X}, \boldsymbol{Y})$ onto the convex hull (Albuquerque et al., 2019) of the source domains. Here, $\ell_{max}$ denotes the maximum loss across all source domains, $\gamma(h)$ denotes the margin, and $\delta$ denotes a margin threshold, with $\Pr[\gamma(h) \leq \delta]$ quantifing the fraction of samples with margin less than or equal to $\delta$. Moreover, $\zeta(\eta)$ denotes the maximum $\mathcal{H}$-divergence between any pair of the prior (posterior) distributions. Notably, $d_{\mathcal{M} \triangle \mathcal{M}}^{\sum_{d=1}^N \pi_d P_S^d(\boldsymbol{Y}|\boldsymbol{X})}\left(\sum_{d=1}^N \pi_d P_S^d(\boldsymbol{X}), P_T(\boldsymbol{X})\right)$ denotes the alignment of source and target prior distributions under the source posterior distribution. Finally, $\lambda_\pi := \inf_{h \in \mathcal{H}}\left[\sum_{d=1}^N \pi_d \epsilon_S^d(h) + \epsilon_T(h)\right]$ denotes the error of the ideal joint hypothesis across source and target domains. For the detailed proof, please refer to Appendix A.*

Compared to classical generalization bounds (Albuquerque et al., 2019; Ben-David et al., 2010) for conventional domain generalization, our bound is derived from the joint distribution factorization that **explicitly expresses each key term in posterior form**. This formulation makes the effect of imbalance immediate. Label imbalance perturbs $P(\boldsymbol{Y})$, which alters the posterior $P(\boldsymbol{Y}|\boldsymbol{X})$ (Tian et al., 2020) and skews the decision boundary toward majority classes (Nagarajan et al., 2021). Domain imbalance disrupts cross-domain posterior consistency by yielding unstable posterior estimates for underrepresented domains, a phenomenon empirically reflected in the Appendix E.2, where minor domains exhibit larger losses. Under this decomposition, imbalance naturally manifests in both the posterior discrepancy term and the margin term of our bound.

Importantly, when source and target posteriors coincide, our bound reduces to the classical ones. In this case, the prior terms should be interpreted conditionally, i.e., measuring prior differences under the source posterior. When posteriors differ due to imbalance, however, aligning priors under mismatched posteriors can distort semantics. Thus, our bound offers a principled extension that explicitly accounts for imbalance-induced posterior mismatch, which classical bounds do not model.

Together, *the prior and posterior terms highlight the need*

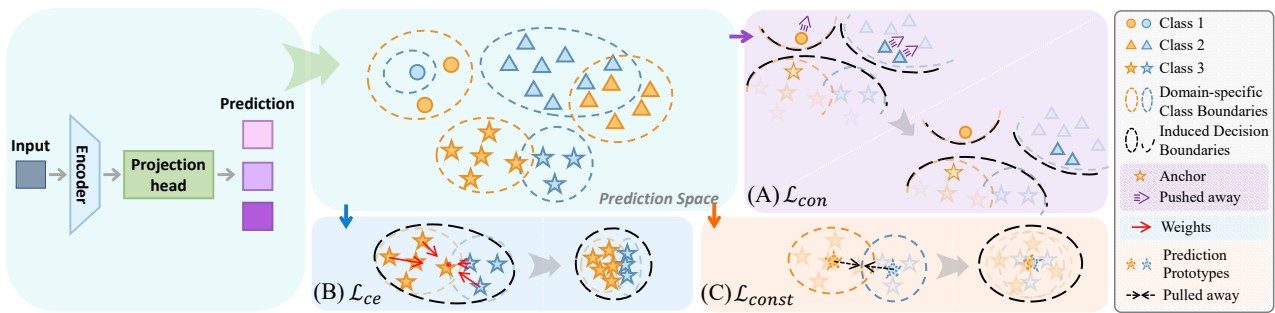

*Figure 3.* An illustration of NDCL, composed of three sub-objectives operating in the prediction space. (A) A contrastive loss that pushes away nearby negatives to enhance inter-class decision-boundary separation. (B) A class-wise re-weighted cross-entropy loss that encourages intra-class compactness and implicitly aligns posteriors. Longer arrows indicate higher weights. (C).A prediction-central alignment strategy that explicitly promotes posterior consistency across domains.

*for prediction consistency across inputs, while the margin term emphasizes learning a larger inter-class separation.* Consequently, our method deliberately avoids explicit alignment of $P(\boldsymbol{X})$, focusing instead on posterior consistency and margin-based separation.

### 2.3. Negative-Dominant Contrastive Learning

Building on the above theoretical insights, we now present our proposed method, Negative-Dominant Contrastive Learning (NDCL), as illustrated in Fig. 3.

**To enhance inter-class decision-boundary separation**, as required for achieving larger margins in our theoretical framework, we reformulate the InfoNCE-like objective:

$$\mathcal{L}_{con} = \sum_{i \in \mathcal{I}} - \log \left\{ \frac{1}{|\mathcal{N}(i)|} \sum_{n \in \mathcal{N}(i)} \frac{1 - \mathrm{s}(\boldsymbol{p}_i, \boldsymbol{p}_n)}{\sum_{a \in \mathcal{A}(i)} (1 - \mathrm{s}(\boldsymbol{p}_i, \boldsymbol{p}_a))} \right\} \tag{1}$$

where $i \in \mathcal{I}$ indexes the instance $\boldsymbol{x}_i$ drawn from all examples across $N$ source domains, $n \in \mathcal{N}(i)$ indexes the $n$-th negative sample from the set of negatives $\boldsymbol{X}_{k^-}$, i.e., the complement of the class $\boldsymbol{X}_k$ corresponding to the label $k$ of $\boldsymbol{x}_i$, and $\mathcal{A}(i) \equiv \mathcal{I} \setminus \{i\}$. $\boldsymbol{p}_{(\cdot)}$ denotes the prediction of the corresponding instance, and $\mathrm{s}(\cdot, \cdot)$ is a similarity function bounded in $[0, 1]$, instantiated as cosine similarity.

From a functional perspective, Eq. (1) can indeed be viewed as a simple modification of the InfoNCE form. However, *this seemingly minor change induces a fundamentally different optimization bias*. While this objective still leverages labeled data to push apart instances from different classes and pull together those from the same class, our formulation shifts the dominant learning signal toward negatives. Unlike SupCon (Khosla et al., 2020), which suggests assigning uniform gradient contributions across all positives, our InfoNCE-like objective purposefully amplifies the influence of informative negatives, especially those that are closer.

Next, a gradient-based interpretation is provided to demonstrate how our proposed Eq. (1) inherently amplifies minority-class signals, mitigating decision boundary dominance by majority classes:

$$\frac{\partial \mathcal{L}_{con}}{\partial \boldsymbol{p}_i} = \frac{1}{Z_i} \left[ - \sum_{p \in \mathcal{P}(i)} \nabla_{\boldsymbol{p}_i} \mathrm{s}(\boldsymbol{p}_i, \boldsymbol{p}_p) \right.$$

$$\left. + \sum_{n \in \mathcal{N}(i)} \nabla_{\boldsymbol{p}_i} \mathrm{s}(\boldsymbol{p}_i, \boldsymbol{p}_n) \cdot \underbrace{\frac{\sum_{p \in \mathcal{P}(i)} 1 - \mathrm{s}(\boldsymbol{p}_i, \boldsymbol{p}_p)}{\sum_{n \in \mathcal{N}(i)} 1 - \mathrm{s}(\boldsymbol{p}_i, \boldsymbol{p}_n)}}_{\text{amplification factor}} \right]$$

where $p \in \mathcal{P}(i)$ indexes the $p$-th positive sample from the set of positives $\boldsymbol{X}_k$, $\nabla_{\boldsymbol{p}_i} \mathrm{s}(\boldsymbol{p}_i, \cdot)$ denotes the gradient of $\mathrm{s}(\boldsymbol{p}_i, \cdot)$ w.r.t. $\boldsymbol{p}_i$, and $Z_i = \sum_{a \in \mathcal{A}(i)} (1 - \mathrm{s}(\boldsymbol{p}_i, \boldsymbol{p}_a))$. This amplification factor dynamically strengthens negative gradients, especially when the local neighborhood of the anchor is dominated by negatives, which leads to a decrease in $\sum_{n \in \mathcal{N}(i)} 1 - \mathrm{s}(\boldsymbol{p}_i, \boldsymbol{p}_n)$. In such cases, minority-class samples are often surrounded by nearby majority-class negatives with small margin, making the denominator relatively small and thus increasing the amplification factor. This enhances repulsion against confusing negatives, helping to preserve class boundaries under imbalance. Meanwhile, such negative-driven contrastive learning encourages local separation from dominant-class negatives, which is particularly beneficial in multi-domain settings where decision boundaries are susceptible to being skewed by domain-specific majority classes, as discussed in (Arjovsky et al., 2019; Nagarajan et al., 2021). In contrast, classical InfoNCE objective primarily strengthens positive compactness via a similar amplification factor while treating all negatives uniformly, which fails to mitigate decision boundary bias. For more details, please refer to Appendix B.

While an abundance of negatives can facilitate learning, those that are easily distinguishable and typically far from the positives tend to render the task trivial, thereby limiting the model's discriminative capacity. To mitigate this issue, we adopt a hard negative mining strategy inspired by

Kalantidis et al. (2020), prioritizing semantically ambiguous samples during contrastive learning. Specifically, for each class $k$, we *rank all instances by their predicted confidence* $p_k$, the $k$-th component of the prediction vector $\boldsymbol{p}$. A refined negative set $\widehat{\mathcal{N}}_k$ is then constructed via mixup (Yan et al., 2020) between low-confidence instances from the positive set, i.e., with small $p_k$, and high-confidence instances from the negative set, i.e., with large $p_k$, encouraging the model to focus on more ambiguous and informative instances:

$$\widehat{\mathcal{N}}_k := \{\hat{\boldsymbol{x}} \,|\, \hat{\boldsymbol{x}} = \lambda \boldsymbol{x}_{k^+} + (1-\lambda)\boldsymbol{x}_{k^-},$$
$$\boldsymbol{x}_{k^+} \sim \boldsymbol{X}_k^{\text{low}}, \; \boldsymbol{x}_{k^-} \sim \boldsymbol{X}_{k^-}^{\text{high}}, \; \lambda \in [0,1]\}$$
(2)

where $\boldsymbol{X}_k^{\text{low}}$ denotes the set of in-class samples with lower $p_k = f_\theta(\boldsymbol{x})_k$, that is uncertain positives, $\boldsymbol{X}_{k^-}^{\text{high}}$ denotes the set of out-of-class samples with higher $p_k$, that is hard negatives, and $\lambda \sim \text{Beta}(\rho, \rho)$ where $\rho$ denotes the coefficient in Beta distribution. Importantly, $p_k$ is adaptively determined within each mini-batch rather than treated as a fixed hyper-parameter.

**To promote posterior consistency across domains**, as required by our theoretical framework, we adopt a prediction-central alignment strategy. By aligning first-order statistics instead of individual samples, this strategy can effectively mitigate imbalance-induced bias by decoupling alignment from sample proportions (Yang et al., 2022). Formally, this alignment objective can be formulated as:

$$\mathcal{L}_{const} = \sum_{i \in \mathcal{I}^\mu} -\frac{1}{|\mathcal{P}^\mu(i)|} \sum_{p \in \mathcal{P}^\mu(i)} \log \frac{\exp(\text{s}(\boldsymbol{\mu}_i, \boldsymbol{\mu}_p))}{\sum_{a \in \mathcal{A}^\mu(i)} \exp(\text{s}(\boldsymbol{\mu}_i, \boldsymbol{\mu}_a))}$$
(3)

where $\boldsymbol{\mu}_i \in \left\{\boldsymbol{\mu}_k^d = \mathbb{E}_{\boldsymbol{x} \in \boldsymbol{X}_k^d} f_\theta(\boldsymbol{x}) \,|\, k \in \{1, \ldots, K\}, \; d \in \{1, \ldots, N\}\right\}$, and $\boldsymbol{\mu}_k^d$ denotes the prediction prototype of $k$-th class in $d$-th domain. This SupCon-inspired loss encourages the alignment of same-class prototypes across domains while repelling those of different classes. Here, $i \in \mathcal{I}^\mu$ indexes the domain-class prototypes, $p \in \mathcal{P}^\mu(i)$ indexes the same-class prototypes from other domains, and $\mathcal{A}^\mu(i) \equiv \mathcal{I}^\mu \setminus \{i\}$.

**To encourage intra-class compactness**, we adopt a cross-entropy loss that adaptively re-weights samples within each class, where its objective can be formulated as:

$$\mathcal{L}_{ce} = \frac{1}{K} \sum_{k=1}^{K} \sum_{i=1}^{n_k} (\omega_k)_i \cdot \ell((\boldsymbol{p}_k)_i, k)$$
(4)

where $(\boldsymbol{p}_k)_i = f_\theta((\boldsymbol{x}_k)_i)$ is the prediction of the $i$-th instance with class $k$, where $(\boldsymbol{x}_k)_i \in \boldsymbol{X}_k$. The weight coefficient $(\omega_k)_i = \frac{\exp(\ell((\boldsymbol{p}_k)_i, k))}{\sum_{j=1}^{n_k} \exp(\ell((\boldsymbol{p}_k)_j, k))}$ adaptively assigns higher weights to samples within $k$-th class that incur larger losses, encouraging the model to focus on hard examples.

This facilitates tighter decision boundaries and promotes better generalization (Lin et al., 2017).

**Finally**, the total objective function can be formulated as:

$$\mathcal{L} = \mathcal{L}_{ce} + \alpha \mathcal{L}_{con} + \beta \mathcal{L}_{const}$$
(5)

where $\alpha$ and $\beta$ are trade-offs. Detailed Algorithm is provided in the Appendix C.

## 3. Experiments

In this section, extensive yet challenging experiments are conducted to comprehensively evaluate the effectiveness of NDCL on three widely used domain generalization benchmarks, involving three distinct types of entangled domain and label shifts.

### 3.1. Datasets and Settings

For the architecture, our NDCL follows the configuration specified in DomainBed (Gulrajani & Lopez-Paz, 2021), utilizing ResNet-50 as the backbone and adopting the training-domain validation method for model selection. Extensive experiments are constructed on three standard Benchmarks, i.e., VLCS, PACS, and OfficeHome. To assess performance under varying imbalance conditions, we define three distinct settings. The GINIDG setting follows the protocol proposed in (Xia et al., 2023), but represents a relatively mild form of imbalance. **To facilitate more rigorous evaluation**, we further propose two challenging settings, namely *TotalHeavyTail* and *Duality*, which depicts long-tailed and compound imbalances, respectively. Detailed descriptions of these settings are provided in the Appendix E.

Twenty-one recent strong comparison methods and another representative methods are deplyed to compare with our NDCL, including representative methods for domain generalization, imbalanced data, and recent methods tailored for IDG. The methods for domain generalization can be divided into five categories: 1) distribution robust method (Group-DRO (Sagawa et al., 2020)), 2) causal methods learning invariance (IRM (Arjovsky et al., 2019), VREx (Krueger et al., 2021), EQRM (Eastwood et al., 2022), TCRI (Salaudeen & Koyejo, 2024)), 3) gradient matching (Fish (Shi et al., 2022), Fishr (Rame et al., 2022), PGrad (Wang et al., 2023)), 4) representation distribution matching (DANN(Ganin et al., 2016), MMD (Li et al., 2018b), CORAL (Sun & Saenko, 2016), RDM (Nguyen et al., 2024)), 5) and other variants (Mixup (Yan et al., 2020), MLDG(Li et al., 2018a), Sag-Net (Nam et al., 2021)). The methods for imbalanced data can be regarded as the re-weighting (Lin et al., 2017; Yang et al., 2022) and margin-based (Cao et al., 2019; Ren et al., 2020) methods. Finally, the methods tailored for IDG can be regarded as the divide-and-conquer strategies (Su et al., 2024; Xia et al., 2023) and the representation-aware weight-

*Table 1.* Overall averaged accuracy under the GINIDG setting on VLCS and PACS Benchmarks. The **bold**, underline, and dashline items are the best, the second-best, and the third-best results, respectively.

| | VLCS | | | | PACS | | | |
|---|---|---|---|---|---|---|---|---|
| | Average | Many | Medium | Few | Average | Many | Medium | Few |
| ERM (Vapnik et al., 1998) | 74.4 ± 1.3 | 73.6 ± 1.2 | 66.0 ± 1.9 | 58.6 ± 2.6 | 82.2 ± 0.3 | 83.9 ± 1.7 | 86.2 ± 0.9 | 71.5 ± 1.0 |
| IRM (Arjovsky et al., 2019) | 76.6 ± 0.4 | 76.8 ± 1.1 | 61.9 ± 1.6 | 58.1 ± 2.9 | 81.2 ± 0.3 | 83.7 ± 1.5 | 83.7 ± 1.6 | 69.9 ± 1.3 |
| GroupDRO (Sagawa et al., 2020) | 75.7 ± 0.4 | 75.1 ± 0.6 | 65.2 ± 0.8 | 60.8 ± 0.7 | 79.7 ± 0.2 | 80.5 ± 1.4 | 84.7 ± 1.0 | 68.2 ± 0.6 |
| Mixup (Yan et al., 2020) | 74.3 ± 0.8 | 75.7 ± 1.3 | 54.7 ± 4.8 | 55.7 ± 1.2 | 82.2 ± 0.7 | 85.7 ± 0.2 | 85.6 ± 0.9 | 64.3 ± 3.6 |
| MLDG (Li et al., 2018a) | 75.7 ± 0.2 | 74.4 ± 0.9 | 62.7 ± 3.2 | 62.5 ± 1.6 | 82.0 ± 0.1 | 84.5 ± 0.6 | 85.2 ± 0.3 | 67.3 ± 0.7 |
| CORAL (Sun & Saenko, 2016) | 75.8 ± 0.3 | 76.7 ± 0.4 | 63.1 ± 1.3 | 60.8 ± 1.1 | 82.8 ± 1.0 | 84.6 ± 1.3 | 86.1 ± 0.6 | 72.6 ± 0.7 |
| MMD (Li et al., 2018b) | 76.2 ± 0.4 | 76.5 ± 0.7 | 62.0 ± 1.4 | 57.5 ± 1.7 | 82.2 ± 0.4 | 83.1 ± 0.2 | 86.8 ± 0.3 | 70.7 ± 1.2 |
| DANN (Ganin et al., 2016) | 76.7 ± 0.3 | 75.9 ± 0.5 | 65.0 ± 1.2 | 62.8 ± 1.6 | 82.8 ± 0.5 | 82.7 ± 0.3 | 86.0 ± 0.7 | 76.1 ± 0.6 |
| SagNet (Nam et al., 2021) | 75.8 ± 0.6 | 76.9 ± 1.0 | 58.9 ± 2.3 | 59.0 ± 0.2 | 81.3 ± 1.4 | 81.6 ± 3.0 | 86.1 ± 0.3 | 68.8 ± 2.3 |
| VREx (Krueger et al., 2021) | 73.7 ± 1.1 | 73.8 ± 1.7 | 63.3 ± 3.0 | 61.3 ± 1.7 | 82.8 ± 0.5 | 84.9 ± 1.2 | 86.2 ± 0.7 | 70.2 ± 1.1 |
| Fish (Shi et al., 2022) | 77.3 ± 0.9 | **78.3 ± 0.9** | 62.8 ± 0.5 | 61.4 ± 1.4 | 80.7 ± 0.9 | 81.3 ± 1.0 | 85.9 ± 0.7 | 66.4 ± 0.3 |
| Fishr (Rame et al., 2022) | 76.2 ± 0.0 | 75.5 ± 0.4 | **66.5 ± 0.2** | 63.7 ± 0.8 | 82.1 ± 0.5 | 81.0 ± 2.1 | 87.5 ± 0.4 | 72.5 ± 1.6 |
| EQRM (Eastwood et al., 2022) | 74.5 ± 0.2 | 75.0 ± 0.6 | 64.5 ± 1.0 | 57.9 ± 1.5 | 82.3 ± 1.2 | 82.3 ± 1.3 | 86.6 ± 0.6 | 74.9 ± 3.2 |
| RDM (Nguyen et al., 2024) | 75.0 ± 0.2 | 74.7 ± 0.4 | 61.3 ± 1.8 | 57.6 ± 1.0 | 83.2 ± 1.1 | 84.4 ± 2.0 | 86.4 ± 0.5 | 73.3 ± 2.2 |
| PGrad (Wang et al., 2023) | 77.6 ± 0.4 | 78.2 ± 0.4 | 63.5 ± 1.4 | 60.3 ± 1.1 | **84.3 ± 0.2** | **87.6 ± 0.4** | 86.6 ± 0.5 | 71.4 ± 2.2 |
| TCRI (Salaudeen & Koyejo, 2024) | 75.6 ± 0.6 | 74.2 ± 1.1 | 64.9 ± 2.5 | 58.7 ± 1.6 | 83.3 ± 0.6 | 82.3 ± 1.2 | **89.2 ± 0.5** | 69.6 ± 1.6 |
| Focal (Lin et al., 2017) | 75.3 ± 0.6 | 76.1 ± 1.5 | 60.1 ± 1.8 | 57.4 ± 0.8 | 81.1 ± 0.3 | 83.2 ± 1.5 | 84.1 ± 1.6 | 68.4 ± 0.5 |
| ReWeight (Yang et al., 2022) | 74.4 ± 0.4 | 72.7 ± 1.5 | 63.1 ± 2.8 | 62.8 ± 0.9 | 82.1 ± 0.8 | 82.7 ± 0.6 | 87.3 ± 1.1 | 70.2 ± 1.1 |
| BSoftmax (Ren et al., 2020) | 73.0 ± 0.4 | 69.9 ± 0.6 | 66.2 ± 3.7 | **68.7 ± 1.0** | 83.2 ± 0.6 | 80.8 ± 1.1 | 87.6 ± 0.5 | **81.8 ± 0.8** |
| LDAM (Cao et al., 2019) | 74.5 ± 0.7 | 73.6 ± 1.3 | 61.0 ± 2.0 | 61.1 ± 1.4 | 83.0 ± 0.5 | 84.3 ± 0.2 | 86.4 ± 0.9 | 74.2 ± 0.8 |
| GINIDG (Xia et al., 2023) | 74.3 ± 0.4 | 73.7 ± 0.5 | 60.7 ± 2.2 | 58.7 ± 1.3 | 81.1 ± 1.0 | 81.8 ± 2.6 | 85.5 ± 1.4 | 69.9 ± 0.5 |
| BoDA (Yang et al., 2022) | 76.3 ± 0.4 | 76.8 ± 0.6 | 62.3 ± 1.8 | 62.8 ± 1.4 | 82.1 ± 0.6 | 85.6 ± 0.8 | 84.1 ± 0.9 | 70.8 ± 2.8 |
| SAMALTDG (Su et al., 2024) | 74.5 ± 0.6 | 72.5 ± 1.1 | 66.4 ± 2.2 | 64.3 ± 1.9 | 82.4 ± 1.3 | 84.5 ± 0.4 | 86.2 ± 1.4 | 71.6 ± 2.3 |
| NDCL (*ours*) | **78.0 ± 0.2** | 77.8 ± 0.9 | 66.0 ± 2.5 | 61.9 ± 1.2 | **85.7 ± 0.4** | 86.5 ± 0.3 | 88.4 ± 0.5 | 79.4 ± 1.9 |

ing method (Yang et al., 2022). All the aforementioned methods are reproduced following the DomainBed training protocol (Gulrajani & Lopez-Paz, 2021), with involved hyperparameters randomly sampled and evaluated over 3 trials, each comprising 10 independent runs. The overall average results are reported from Tab. 1 to 3.

## 3.2. Results and Analyses

**Overall Result Analysis** Tab. 1 to 3 report performance comparisons under the GINIDG, TotalHeavyTail, and Duality settings, respectively. The TotalHeavyTail setting focuses on consistent long-tailed distributions across domains, while the Duality setting introduces heterogeneous label shifts along with imbalanced domain sizes, i.e., major and minor domains. The GINIDG setting, adapted from (Xia et al., 2023), serves as an intermediate case, characterized by a milder degree of imbalance.

From Tab. 1, we can observe the following findings: 1) Our NDCL achieves the best overall performance, outperforming all baselines in average accuracy on both datasets. Notably, it achieves first or second place in most of sub-group categories, including the challenging few-shot classes, where conventional domain generalization methods, such as IRM and GroupDRO, consistently underperform due to their lack of imbalance-awareness. 2) Compared to the methods tailored to imbalanced data, our NDCL demonstrates superior robustness across sub-group categories, especially in PACS where it achieves the second-best few-shot accuracy. While the methods, such as Fish and PGrad, show strong performance for major classes, they drop significantly for medium and minority classes, highlighting their limitations under imbalance. 3) Compared to IDG-specific baselines, i.e., BoDA, SAMALTDG, and GINIDG, our method consistently improves upon them over 1.5% on average, demonstrating the effectiveness of modeling cross-domain posterior alignment while maintaining margin-aware separation across

In Tab. 2, our NDCL still demonstrates remarkable robustness and consistency across multiple datasets and sub-group categories. Nevertheless, under this setting, the performance of other methods has undergone significant reordering. Interestingly, methods specifically designed for imbalanced data, e.g., BSoftmax, emerge among the top 3 performers, especially on minority classes. This is because the Total-HeavyTail setting amplifies inter-class competition due to severe label imbalance, making fine-grained class discrimination more critical than domain alignment. Sinces all domains have similar class distribution skews, domain shift becomes relatively less influential. Conventional domain generalization methods, which perform well on major and medium classes, often struggle on minority classes, further validating this observation.

In Tab.3, apart from the consistently strong performance of our NDCL, the presence of heterogeneous label shift reduces inter-class competition, thereby amplifying the impact of domain shift. As a result, conventional domain generalization methods achieve competitive performance, and the degradation in minority class accuracy is less severe compared to the TotalHeavyTail setting.

*Table 2.* Overall averaged accuracy under the TotalHeavyTail setting on three Benchmarks.

| | VLCS | | | | PACS | | | | OfficeHome | | | |
|---|---|---|---|---|---|---|---|---|---|---|---|---|
| | Average | Many | Medium | Few | Average | Many | Medium | Few | Average | Many | Medium | Few |
| ERM | 73.6 ± 0.8 | 79.2 ± 1.5 | 61.0 ± 3.0 | 50.8 ± 1.2 | 69.6 ± 0.9 | 87.4 ± 0.7 | 62.0 ± 0.5 | 71.2 ± 1.5 | 46.1 ± 0.6 | 74.9 ± 0.7 | 64.8 ± 0.1 | 25.4 ± 1.0 |
| IRM | 70.9 ± 0.7 | 82.0 ± 0.8 | 55.6 ± 1.4 | 40.8 ± 3.1 | 71.8 ± 2.1 | 87.2 ± 1.3 | 62.9 ± 1.7 | 73.4 ± 3.8 | 43.8 ± 1.1 | 73.6 ± 0.4 | 61.2 ± 1.6 | 23.6 ± 1.4 |
| GroupDRO | 73.1 ± 0.6 | 79.4 ± 1.1 | 62.4 ± 1.9 | 47.4 ± 0.9 | 72.3 ± 1.4 | 87.3 ± 1.0 | 67.8 ± 3.6 | 69.9 ± 1.1 | 46.0 ± 0.6 | 75.6 ± 0.5 | 64.3 ± 0.3 | 25.0 ± 0.8 |
| Mixup | 70.9 ± 1.1 | 81.7 ± 0.8 | 56.5 ± 2.6 | 41.9 ± 2.4 | 68.3 ± 1.2 | 86.5 ± 0.7 | 65.0 ± 1.3 | 62.9 ± 1.6 | 44.9 ± 0.6 | 75.0 ± 0.4 | 64.2 ± 0.4 | 23.1 ± 0.9 |
| MLDG | 73.2 ± 0.1 | 78.9 ± 1.4 | 58.9 ± 3.2 | 51.4 ± 1.2 | 70.1 ± 1.2 | 86.7 ± 0.8 | 67.7 ± 1.8 | 64.8 ± 2.1 | 45.4 ± 0.4 | 73.8 ± 1.0 | 63.3 ± 0.7 | 24.9 ± 0.4 |
| CORAL | 73.1 ± 0.7 | 82.7 ± 0.7 | 59.4 ± 1.9 | 46.2 ± 1.0 | 69.8 ± 0.8 | 88.1 ± 1.3 | 66.9 ± 1.8 | 62.3 ± 1.4 | 45.7 ± 0.9 | 78.5 ± 0.2 | 64.9 ± 1.0 | 22.9 ± 0.9 |
| MMD | 72.0 ± 0.4 | 81.3 ± 1.1 | 57.5 ± 1.9 | 45.1 ± 2.6 | 70.3 ± 0.5 | 85.8 ± 1.6 | 66.1 ± 1.0 | 67.9 ± 0.7 | 45.4 ± 0.4 | 74.6 ± 1.2 | 63.9 ± 0.5 | 24.7 ± 0.8 |
| DANN | 72.4 ± 1.1 | 75.9 ± 1.0 | 64.1 ± 1.2 | 49.6 ± 1.8 | 73.9 ± 0.7 | 86.3 ± 2.3 | 71.2 ± 3.5 | 71.1 ± 0.7 | 44.2 ± 0.9 | 74.9 ± 1.2 | 63.1 ± 0.7 | 22.1 ± 0.9 |
| SagNet | 72.7 ± 0.4 | 79.8 ± 0.3 | 61.8 ± 4.3 | 45.7 ± 3.9 | 72.5 ± 0.8 | 85.5 ± 0.5 | 68.2 ± 1.5 | 70.4 ± 2.8 | 46.3 ± 0.5 | 77.6 ± 0.3 | 66.0 ± 1.0 | 24.1 ± 0.4 |
| VREx | 71.6 ± 1.4 | 80.1 ± 1.7 | 61.4 ± 1.7 | 45.2 ± 0.7 | 70.3 ± 0.6 | 85.9 ± 1.6 | 66.0 ± 0.4 | 66.9 ± 2.0 | 45.5 ± 0.4 | 74.7 ± 0.2 | 63.2 ± 0.3 | 25.1 ± 1.0 |
| Fish | 74.3 ± 0.2 | 81.6 ± 0.7 | 59.3 ± 2.1 | 50.3 ± 1.7 | 69.9 ± 1.7 | 86.2 ± 2.0 | 63.0 ± 2.5 | 70.7 ± 2.1 | 46.5 ± 0.2 | 77.3 ± 0.7 | 65.5 ± 1.0 | 24.8 ± 0.2 |
| Fishr | 74.2 ± 0.4 | 80.5 ± 0.3 | 60.6 ± 1.7 | 49.3 ± 0.2 | 71.6 ± 0.3 | 87.1 ± 0.6 | 67.1 ± 0.6 | 71.1 ± 1.9 | 46.2 ± 0.5 | 75.9 ± 0.7 | 64.7 ± 0.4 | 25.3 ± 0.5 |
| EQRM | 72.1 ± 0.4 | 77.9 ± 1.4 | 58.8 ± 3.6 | 47.5 ± 1.3 | 72.5 ± 0.2 | 86.4 ± 0.0 | 69.3 ± 1.8 | 72.0 ± 0.9 | 46.3 ± 0.1 | 77.0 ± 0.2 | 63.4 ± 0.2 | 25.4 ± 0.4 |
| RDM | 71.8 ± 0.2 | 79.7 ± 1.8 | 59.8 ± 2.5 | 44.6 ± 0.9 | 72.8 ± 0.6 | 89.0 ± 0.7 | 66.0 ± 1.1 | 73.5 ± 0.9 | 46.9 ± 0.1 | 74.8 ± 0.5 | 64.1 ± 0.7 | 26.6 ± 0.3 |
| PGrad | 72.7 ± 0.2 | 82.9 ± 0.5 | 58.6 ± 1.1 | 47.0 ± 1.0 | 71.3 ± 0.2 | 91.9 ± 0.5 | 64.1 ± 1.6 | 68.7 ± 1.8 | 48.0 ± 0.4 | 78.2 ± 0.5 | 66.5 ± 0.6 | 26.5 ± 0.6 |
| TCRI | 71.8 ± 0.9 | 78.0 ± 0.7 | 58.6 ± 0.4 | 51.1 ± 0.3 | 73.4 ± 0.7 | 87.7 ± 0.7 | 67.7 ± 2.4 | 72.0 ± 1.5 | 48.0 ± 0.4 | 73.4 ± 0.8 | 65.0 ± 0.4 | 29.4 ± 0.4 |
| Focal | 71.9 ± 0.9 | 81.6 ± 1.4 | 57.8 ± 2.5 | 46.5 ± 1.3 | 69.7 ± 1.3 | 87.9 ± 1.6 | 64.6 ± 2.4 | 66.8 ± 1.8 | 45.1 ± 0.2 | 75.2 ± 0.8 | 62.9 ± 0.5 | 24.2 ± 0.2 |
| ReWeight | 73.0 ± 0.6 | 72.1 ± 2.3 | 64.5 ± 3.4 | 59.2 ± 0.4 | 74.1 ± 1.4 | 83.0 ± 1.5 | 70.3 ± 1.9 | 75.7 ± 1.0 | 47.1 ± 0.6 | 74.3 ± 1.4 | 64.6 ± 0.2 | 27.4 ± 0.5 |
| BSoftmax | 74.7 ± 0.6 | 74.4 ± 0.9 | 62.9 ± 1.5 | 62.1 ± 1.6 | 73.9 ± 1.1 | 77.9 ± 1.5 | 70.7 ± 2.2 | 80.8 ± 1.8 | 48.6 ± 0.5 | 69.5 ± 0.9 | 61.9 ± 0.5 | 33.0 ± 0.5 |
| LDAM | 72.4 ± 1.6 | 80.5 ± 1.0 | 59.2 ± 3.1 | 49.7 ± 2.2 | 71.9 ± 1.8 | 84.6 ± 1.1 | 65.5 ± 3.6 | 73.1 ± 1.2 | 46.3 ± 0.4 | 74.2 ± 0.6 | 64.1 ± 0.5 | 26.0 ± 0.6 |
| GINIDG | 70.0 ± 0.9 | 80.4 ± 1.0 | 56.2 ± 1.4 | 38.3 ± 3.9 | 68.2 ± 0.2 | 88.3 ± 1.5 | 61.6 ± 0.2 | 65.5 ± 0.4 | 44.9 ± 0.5 | 72.1 ± 0.5 | 62.2 ± 0.1 | 25.5 ± 0.8 |
| BoDA | 70.1 ± 0.6 | 82.6 ± 1.6 | 54.9 ± 1.4 | 40.7 ± 1.6 | 70.1 ± 0.8 | 86.1 ± 1.7 | 67.9 ± 1.4 | 64.5 ± 1.9 | 44.9 ± 0.3 | 77.5 ± 0.4 | 64.2 ± 0.9 | 22.0 ± 0.5 |
| SAMALTDG | 72.7 ± 1.5 | 71.4 ± 1.8 | 66.0 ± 1.3 | 55.3 ± 2.3 | 70.7 ± 0.5 | 82.3 ± 1.3 | 66.0 ± 1.7 | 75.2 ± 1.4 | 46.1 ± 0.4 | 74.1 ± 1.0 | 64.0 ± 0.2 | 25.6 ± 0.5 |
| NDCL (*ours*) | **75.6 ± 0.4** | 70.7 ± 0.1 | **66.1 ± 0.4** | 61.0 ± 1.2 | **76.4 ± 0.6** | 84.9 ± 0.8 | **72.2 ± 1.3** | 79.2 ± 0.9 | **49.0 ± 0.2** | 71.6 ± 0.8 | 66.0 ± 0.2 | 30.5 ± 0.3 |

*Table 3.* Overall averaged accuracy under the Duality setting on three Benchmarks.

| | VLCS | | | | PACS | | | | OfficeHome | | | |
|---|---|---|---|---|---|---|---|---|---|---|---|---|
| | Average | Many | Medium | Few | Average | Many | Medium | Few | Average | Many | Medium | Few |
| ERM | 54.9 ± 2.4 | 29.5 ± 4.8 | 72.4 ± 2.9 | 65.3 ± 2.1 | 67.9 ± 1.1 | 58.7 ± 2.3 | 81.2 ± 1.5 | 61.7 ± 0.8 | 52.3 ± 0.2 | 68.9 ± 1.7 | 61.7 ± 0.1 | 42.9 ± 0.7 |
| IRM | 52.6 ± 3.2 | 27.1 ± 6.4 | 73.5 ± 3.1 | 66.4 ± 2.1 | 65.2 ± 0.7 | 51.0 ± 4.5 | 80.3 ± 1.4 | 58.6 ± 1.5 | 49.4 ± 1.0 | 67.1 ± 2.3 | 60.1 ± 1.4 | 39.8 ± 1.0 |
| GroupDRO | 57.8 ± 2.3 | 36.3 ± 4.4 | 70.5 ± 2.2 | 69.2 ± 0.9 | 66.7 ± 0.1 | 52.1 ± 2.5 | 84.0 ± 1.9 | 57.6 ± 1.7 | 51.7 ± 0.1 | 70.5 ± 1.4 | 62.2 ± 0.7 | 41.7 ± 0.5 |
| Mixup | 54.4 ± 2.0 | 30.0 ± 3.7 | 69.2 ± 2.3 | 68.0 ± 0.4 | 63.9 ± 0.2 | 53.0 ± 3.5 | 81.8 ± 1.4 | 54.0 ± 1.7 | 52.2 ± 0.4 | 70.6 ± 0.9 | 62.6 ± 0.1 | 42.2 ± 0.8 |
| MLDG | 51.3 ± 3.2 | 24.2 ± 2.9 | 71.9 ± 1.4 | 67.2 ± 2.4 | 66.1 ± 1.5 | 60.3 ± 5.4 | 82.6 ± 1.4 | 57.9 ± 5.2 | 52.5 ± 0.4 | 68.0 ± 1.2 | 62.9 ± 1.3 | 43.1 ± 0.2 |
| CORAL | 55.3 ± 1.8 | 36.4 ± 1.6 | 68.0 ± 2.7 | 66.1 ± 1.8 | 65.0 ± 1.1 | 60.1 ± 1.0 | 85.3 ± 2.5 | 52.4 ± 1.0 | 54.5 ± 0.1 | 74.3 ± 1.7 | 66.6 ± 0.1 | 43.8 ± 0.4 |
| MMD | 50.5 ± 1.9 | 26.6 ± 2.7 | 69.0 ± 1.9 | 64.3 ± 1.7 | 68.2 ± 0.7 | 59.3 ± 5.2 | 82.0 ± 1.0 | 59.7 ± 2.5 | 50.9 ± 0.3 | 69.0 ± 1.6 | 60.3 ± 0.5 | 41.4 ± 0.3 |
| DANN | 60.9 ± 3.6 | 41.4 ± 5.7 | 70.1 ± 3.1 | 67.2 ± 1.2 | 68.3 ± 1.1 | 65.8 ± 3.1 | 83.0 ± 3.6 | 60.4 ± 2.4 | 51.8 ± 0.3 | 70.0 ± 1.7 | 61.9 ± 1.0 | 42.2 ± 1.1 |
| SagNet | 53.6 ± 2.8 | 27.1 ± 3.7 | 73.4 ± 3.1 | 69.9 ± 1.9 | 66.0 ± 0.9 | 53.3 ± 4.9 | 82.5 ± 1.8 | 56.7 ± 2.3 | 53.9 ± 0.5 | 73.0 ± 0.9 | 63.8 ± 0.3 | 43.9 ± 0.7 |
| VREx | 51.5 ± 2.5 | 25.1 ± 3.4 | 72.0 ± 2.9 | 68.1 ± 1.2 | 65.6 ± 1.5 | 56.5 ± 2.2 | 80.4 ± 2.1 | 59.0 ± 1.8 | 52.9 ± 0.4 | 70.1 ± 1.5 | 62.5 ± 0.6 | 43.3 ± 0.7 |
| Fish | 56.6 ± 2.2 | 31.5 ± 3.4 | 71.2 ± 3.0 | 70.5 ± 1.5 | 66.1 ± 1.0 | 61.4 ± 3.4 | 80.1 ± 2.2 | 57.4 ± 2.4 | 52.9 ± 0.1 | 69.5 ± 1.6 | 62.6 ± 0.4 | 43.3 ± 0.0 |
| Fishr | 52.5 ± 4.1 | 25.9 ± 5.4 | 70.6 ± 0.4 | 68.3 ± 1.4 | 66.5 ± 1.7 | 65.1 ± 2.1 | 79.5 ± 1.0 | 56.8 ± 3.2 | 52.5 ± 0.3 | 71.1 ± 1.5 | 62.0 ± 0.6 | 43.0 ± 0.7 |
| EQRM | 52.9 ± 3.8 | 26.8 ± 4.6 | 70.6 ± 2.5 | 66.9 ± 2.1 | 66.7 ± 1.2 | 52.9 ± 1.2 | 80.8 ± 2.2 | 60.8 ± 1.0 | 52.4 ± 0.2 | 68.0 ± 1.4 | 63.0 ± 0.2 | 42.9 ± 0.3 |
| RDM | 54.7 ± 4.5 | 30.6 ± 6.5 | 71.6 ± 1.4 | 67.1 ± 1.3 | 67.1 ± 1.3 | 56.8 ± 2.9 | 83.5 ± 1.0 | 58.6 ± 1.1 | 52.9 ± 0.3 | 71.1 ± 1.0 | 61.5 ± 0.4 | 43.7 ± 0.8 |
| PGrad | 55.7 ± 4.3 | 30.0 ± 6.2 | 68.6 ± 0.9 | 70.1 ± 1.2 | 70.6 ± 0.8 | 57.0 ± 4.8 | 83.9 ± 1.6 | 63.7 ± 1.2 | 55.4 ± 0.2 | 73.0 ± 0.7 | 65.7 ± 0.6 | 46.0 ± 0.4 |
| TCRI | 52.0 ± 3.9 | 24.7 ± 6.9 | 69.3 ± 1.2 | 70.8 ± 1.6 | 69.0 ± 0.3 | 69.5 ± 2.4 | 84.1 ± 1.8 | 57.1 ± 2.6 | 54.6 ± 0.5 | 70.8 ± 1.4 | 64.5 ± 0.9 | 45.4 ± 0.6 |
| Focal | 51.5 ± 3.3 | 25.0 ± 4.9 | 74.1 ± 1.7 | 67.6 ± 1.7 | 67.6 ± 1.6 | 61.0 ± 4.8 | 80.7 ± 3.5 | 60.6 ± 1.9 | 50.8 ± 0.7 | 67.5 ± 1.4 | 60.3 ± 1.3 | 41.5 ± 0.4 |
| ReWeight | 51.5 ± 2.2 | 25.6 ± 2.5 | 67.8 ± 3.1 | 70.4 ± 1.6 | 65.7 ± 1.4 | 56.8 ± 4.3 | 77.0 ± 3.2 | 59.7 ± 1.8 | 52.4 ± 0.3 | 69.8 ± 1.6 | 61.3 ± 0.8 | 43.7 ± 0.4 |
| BSoftmax | 50.3 ± 1.0 | 22.7 ± 2.3 | 76.0 ± 1.8 | 69.2 ± 0.9 | 67.6 ± 0.9 | 52.4 ± 4.6 | 84.5 ± 2.1 | 62.0 ± 0.2 | 52.8 ± 0.5 | 67.4 ± 1.4 | 62.1 ± 0.6 | 44.4 ± 1.0 |
| LDAM | 50.2 ± 2.4 | 22.0 ± 4.1 | 73.0 ± 0.3 | 68.8 ± 1.1 | 66.6 ± 0.5 | 50.8 ± 4.1 | 79.7 ± 1.2 | 61.7 ± 1.9 | 50.1 ± 0.3 | 68.1 ± 1.3 | 59.6 ± 1.1 | 40.8 ± 0.2 |
| GINIDG | 58.6 ± 1.5 | 40.7 ± 2.6 | 68.4 ± 3.8 | 64.5 ± 1.2 | 63.2 ± 1.9 | 60.4 ± 4.0 | 78.4 ± 2.8 | 55.5 ± 1.0 | 50.2 ± 0.6 | 69.4 ± 0.5 | 59.8 ± 0.6 | 40.5 ± 0.9 |
| BoDA | 58.2 ± 0.7 | 40.2 ± 0.7 | 73.4 ± 2.1 | 65.8 ± 1.0 | 64.5 ± 1.8 | 55.2 ± 1.7 | 84.7 ± 2.7 | 54.1 ± 2.2 | 53.5 ± 0.6 | 74.6 ± 0.9 | 65.5 ± 0.2 | 42.7 ± 0.8 |
| SAMALTDG | 57.5 ± 1.2 | 32.4 ± 1.8 | 76.8 ± 0.1 | 69.2 ± 1.0 | 66.1 ± 0.5 | 49.6 ± 3.0 | 81.7 ± 0.7 | 59.9 ± 0.3 | 51.7 ± 1.0 | 68.9 ± 0.9 | 61.3 ± 1.2 | 42.8 ± 1.1 |
| NDCL (*ours*) | **61.6 ± 0.8** | 38.5 ± 1.5 | 76.4 ± 1.9 | 70.7 ± 1.7 | **71.1 ± 0.7** | 53.3 ± 1.7 | **86.3 ± 1.1** | 65.0 ± 0.7 | **55.7 ± 0.2** | 71.4 ± 1.0 | 65.8 ± 0.5 | **47.6 ± 0.6** |

Surprisingly, the methods tailored to IDG underperform across all three settings. One possible reason is that the random hyperparameter selection in DomainBed amplifies the optimization difficulties inherent to divide-and-conquer strategies commonly used in these methods.

**Margin and Posterior Discrepancy Analysis** Fig. 4 visualizes the averaged margin and posterior discrepancy of each class between the multiple source domains and the target domain during inference, providing an empirical quantification of the theoretical factors in our bound. Overall, our NDCL achieves larger margins and lower posterior discrepancy, represented by the brown line, across most classes under both settings. These findings validate the effectiveness of our NDCL in mitigating the compounded effects of entangled domain and label shifts in IDG. In addition, under the TotalHeavyTail setting, negative margins are more prevalent, even among medium classes, which indicates intensi-

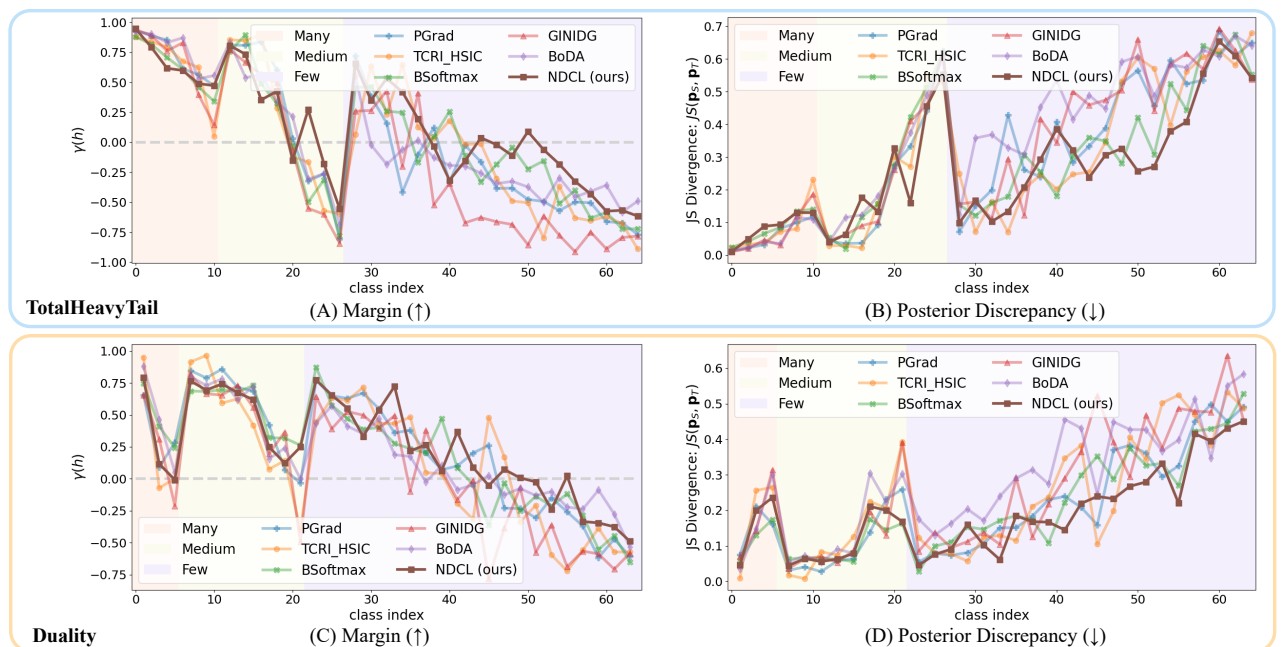

*Figure 4.* Illustrative per-class comparison of margin $\gamma(h)$ and posterior discrepancy based on JS divergence on OfficeHome under two different settings across several representative methods including ours. ($\uparrow$) denotes the larger the better, while ($\downarrow$) denotes the opposite.

*Table 4.* Ablation studies on OfficeHome under two different settings.

| | TotalHeavyTail | | | | Duality | | | |
|---|---|---|---|---|---|---|---|---|
| | Average | Many | Medium | Few | Average | Many | Medium | Few |
| Baseline (ERM) | $46.1 \pm 0.6$ | $74.9 \pm 0.7$ | $64.8 \pm 0.1$ | $25.4 \pm 1.0$ | $52.3 \pm 0.2$ | $68.9 \pm 1.7$ | $61.7 \pm 0.1$ | $42.9 \pm 0.7$ |
| NoWeight (w/o $\omega_k$) | $47.1 \pm 0.5$ | $75.6 \pm 0.2$ | $65.1 \pm 0.9$ | $26.8 \pm 0.6$ | $54.3 \pm 0.2$ | $69.8 \pm 1.3$ | $64.0 \pm 0.2$ | $45.0 \pm 0.5$ |
| NoCon (w/o $\mathcal{L}_{Con}$) | $46.1 \pm 0.4$ | $74.1 \pm 1.0$ | $64.0 \pm 0.2$ | $25.6 \pm 0.5$ | $51.7 \pm 1.0$ | $68.8 \pm 0.8$ | $61.4 \pm 1.3$ | $42.7 \pm 1.1$ |
| NoConst (w/o $\mathcal{L}_{Const}$) | $47.9 \pm 0.1$ | $76.6 \pm 0.4$ | $64.8 \pm 0.4$ | $27.4 \pm 0.5$ | $53.4 \pm 0.2$ | $69.7 \pm 0.4$ | $63.3 \pm 0.4$ | $43.8 \pm 0.4$ |
| NoAug (w/o $\widehat{\mathcal{N}}_k$) | $48.6 \pm 0.4$ | $76.0 \pm 0.4$ | $65.6 \pm 0.5$ | $27.9 \pm 0.6$ | $54.5 \pm 0.3$ | $71.9 \pm 1.5$ | $64.0 \pm 0.4$ | $45.3 \pm 0.5$ |
| NDCL | $49.0 \pm 0.2$ | $71.6 \pm 0.8$ | $66.0 \pm 0.2$ | $30.5 \pm 0.3$ | $55.7 \pm 0.2$ | $71.4 \pm 1.0$ | $65.8 \pm 0.5$ | $47.6 \pm 0.6$ |

fied inter-class competition due to consistent cross-domain imbalance. Correspondingly, posterior discrepancy across domains, measured by JS divergence between $P(\boldsymbol{Y}|\boldsymbol{X})$s, is typically larger in regions with small or negative margins, and gradually decreases as the margin increases, indicating a correlation between class separability and cross-domain prediction stability. In contrast, under the Duality setting, fewer negative margins and lower discrepancies are observed, likely due to its domain-specific imbalance allowing classes to dominate locally, thereby reducing overall competition.

**Ablation Studies**  Tab. 4 reports the ablation studies of our NDCL. The removal of $\mathcal{L}_{con}$ leads to a significant performance drop, particularly for minority classes, demonstrating its essential role in promoting decision boundary separation. Other components, i.e., $\omega_k$ and $\mathcal{L}_{const}$, also contribute, but their impact varies across the TotalHeavyTail and Duality settings. This variation reflects the difference in imbalance patterns, where TotalHeavyTail presents consistent cross-

domain long-tailed distributions, while Duality involves domain-specific imbalance, demanding stronger generalization under heterogeneous class distributions. In addition, compared with NoAug and NDCL, the augmented negative set $\widehat{\mathcal{N}}_k$ further promotes decision boundary separation.

**Extended Experimental Analyses**  Further analyses on sensitivity to hyper-parameters, computational complexity, the effect of contrastive objectives, sampling behaviors on traditional imbalance methods, prior distribution analysis, furhter quantitative evaluation, and significance test are provided in the Appendix E.3-E.10.

## 4. Conclusion

In this paper, we bridge the theoretical gap in imbalanced domain generalization (IDG) by establishing a novel generalization bound that jointly accounts for posterior discrepancy and decision margin, which are two critical factors overlooked by conventional bounds under imbalance. Mo-

tivated by this theoretical insight, we introduce a unified framework that directly addresses entangled domain and label shifts through explicitly steering the decision boundary, eliminating the need for ad-hoc multi-level objective functions. Our key innovation, NDCL, leverages a negative-dominant contrastive learning paradigm to simultaneously enhance discriminability and enforce posterior consistency across domains. Rigorous yet challenging experiments on three benchmarks, involving three distinct types of entangled shifts, demonstrate the effectiveness of our NDCL.

While NDCL indirectly mitigates domain prior discrepancy via governed decision boundaries, explicit modeling of domain priors remains challenging under severe imbalance due to potential ill-posedness in representation alignment. Future directions may integrate causality-inspired mechanisms to disentangle underlying factors, offering a promising path toward more generalizable models.

## Impact Statement

This paper presents work whose goal is to advance the field of Machine Learning. There are many potential societal consequences of our work, none of which we feel must be specifically highlighted here.

## Acknowledgements

This work was supported by the Natural Science Foundation of China (NSFC) (Grant No.62376126).

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

# A. Error Bound for Imbalanced Domain Generalization

We build upon the domain adaptation bound established in (Ben-David et al., 2010)[Thm. 2], which relates the target error to the source error and a distributional discrepancy term. Specifically, for any hypothesis $h \in \mathcal{H}$, the target domain risk $\epsilon_T(h)$ is upper-bounded as:

$$\epsilon_T(h) \leq \epsilon_S(h) + d_{\mathcal{H} \triangle \mathcal{H}}(P_S(\boldsymbol{X}), P_T(\boldsymbol{X})) + \lambda_{\mathcal{H}} \tag{6}$$

where $\epsilon_S(h)$ is the source risk, $d_{\mathcal{H} \triangle \mathcal{H}}(P_S(\boldsymbol{X}), P_T(\boldsymbol{X}))$ is the $\mathcal{H} \triangle \mathcal{H}$-divergence measuring the discrepancy between input distributions, and $\lambda_{\mathcal{H}} := \inf_{h \in \mathcal{H}}[\epsilon_S(h) + \epsilon_T(h)]$ denotes the error of the ideal joint hypothesis over both domains. While this bound has been influential in understanding domain shift, it primarily focuses on aligning marginal input distributions $P(\boldsymbol{X})$ and subsequent variants of this bound have been widely used in existing domain generalization theory analysis (Albuquerque et al., 2019; Lu et al., 2024).

However, in IDG, aligning these marginal input distributions $P(\boldsymbol{X})$ may fail to guarantee effective transfer when label distributions $P(\boldsymbol{Y})$ vary significantly across domains. Even under invariant class-conditional distributions across domains $P(\boldsymbol{X}|\boldsymbol{Y})$, a shift in the label prior $P(\boldsymbol{Y})$ can lead to a substantial change in the posterior $P(\boldsymbol{Y}|\boldsymbol{X})$ via Bayesian rule. Consequently, the classifier trained on the source domain may fail to generalize to the target domain, despite exhibiting a low $\mathcal{H} \triangle \mathcal{H}$-divergence. To illustrate this limitation, we present concrete examples where input alignment alone is insufficient for generalization. Consider a binary classification task with classes $\boldsymbol{Y} \in \{0, 1\}$, corresponding to two categories. Assume that the class-conditional distributions are identical across the source and target domains:

$$P_S(\boldsymbol{X}|\boldsymbol{Y}=0) = P_T(\boldsymbol{X}|\boldsymbol{Y}=0) = \mathcal{N}([-1,0], \boldsymbol{I})$$
$$P_S(\boldsymbol{X}|\boldsymbol{Y}=1) = P_T(\boldsymbol{X}|\boldsymbol{Y}=1) = \mathcal{N}([1,0], \boldsymbol{I})$$

but the class priors differ significantly:

$$P_S(\boldsymbol{Y}=0) = 0.1, P_S(\boldsymbol{Y}=1) = 0.9; \quad P_T(\boldsymbol{Y}=0) = 0.9, P_T(\boldsymbol{Y}=\boldsymbol{I}) = 0.1$$

Under these conditions, the resulting marginal input distributions are:

$$P_S(\boldsymbol{X}) = 0.1 \cdot \mathcal{N}([-1,0], \boldsymbol{I}) + 0.9 \cdot \mathcal{N}([1,0], \boldsymbol{I}),$$
$$P_T(\boldsymbol{X}) = 0.9 \cdot \mathcal{N}([-1,0], \boldsymbol{I}) + 0.1 \cdot \mathcal{N}([1,0], \boldsymbol{I})$$

which clearly diverge due to the shift in label priors, despite the shared class-conditional structure. This example highlights that marginal alignment of input features is insufficient for generalization when posterior distributions are influenced by label imbalance.

Consequently, it is more appropriate to measure the divergence between the joint distributions $P(\boldsymbol{X}, \boldsymbol{Y})$ across domains, rather than focusing solely on the marginal distributions $P(\boldsymbol{X})$.

**Step 1: Refining the error bound in Eq.** (6) **with the joint distributions.** To better capture prediction-induced discrepancies under distribution shift, we reformulate the classical discrepancy $d_{\mathcal{H} \triangle \mathcal{H}}(P_S(\boldsymbol{X}), P_T(\boldsymbol{X}))$ from (Ben-David et al., 2010), where $h: \boldsymbol{X} \to \{0, 1\} \in \mathcal{H}$, into a form defined over the mappings $m = yh(x): \boldsymbol{X} \times \boldsymbol{Y} \to \{-1, 1\} \in \mathcal{M}$, with $h: \boldsymbol{X} \to \{-1, 1\}$ and $y \in \boldsymbol{Y} = \{-1, 1\}$. We then define a new discrepancy $d_{\mathcal{M} \triangle \mathcal{M}}$ over the auxiliary class $\mathcal{M}$, which acts directly on this composed formulation:

$$\triangle := \sup_{m, m'} |\mathbb{E}_{(\boldsymbol{X}, \boldsymbol{Y}) \sim P_S(\boldsymbol{X}, \boldsymbol{Y})}[I[m(\boldsymbol{X}, \boldsymbol{Y}) \neq m'(\boldsymbol{X}, \boldsymbol{Y})]] - \mathbb{E}_{(\boldsymbol{X}, \boldsymbol{Y}) \sim P_T(\boldsymbol{X}, \boldsymbol{Y})}[I[m(\boldsymbol{X}, \boldsymbol{Y}) \neq m'(\boldsymbol{X}, \boldsymbol{Y})]]|$$

Notably, this transformation does not alter the underlying hypothesis $\mathcal{H}$ class but instead reorganizes the evaluation space to emphasize decision-relevant characteristics.

In this way, we reformulate their error bound as follows:

$$|\epsilon_T(h) - \epsilon_S(h)| \leq d_{\mathcal{M} \triangle \mathcal{M}}(P_S(\boldsymbol{X}, \boldsymbol{Y}), P_T(\boldsymbol{X}, \boldsymbol{Y})) \tag{7}$$

**Lemma A.1.** *Let $P_S(\boldsymbol{X}, \boldsymbol{Y})$ and $P_T(\boldsymbol{X}, \boldsymbol{Y})$ be joint distributions from the source and target domains. Then the $\mathcal{M} \triangle \mathcal{M}$-divergence between these joint distributions can be upper bounded by:*

$$d_{\mathcal{M} \triangle \mathcal{M}}(P_S(\boldsymbol{X}, \boldsymbol{Y}), P_T(\boldsymbol{X}, \boldsymbol{Y})) \leq \quad d_{\mathcal{M} \triangle \mathcal{M}}^{P_S(\boldsymbol{Y}|\boldsymbol{X})}(P_S(\boldsymbol{X}), P_T(\boldsymbol{X}))$$
$$+ \mathbb{E}_{x \sim P_T(\boldsymbol{X})}[d_{\mathcal{M} \triangle \mathcal{M}}(P_S(\boldsymbol{Y}|\boldsymbol{X}=x), P_T(\boldsymbol{Y}|\boldsymbol{X}=x))]$$

*where the non-standard $d_{\mathcal{M}\triangle\mathcal{M}}^{P_S(\boldsymbol{Y}|\boldsymbol{X})}\left(P_S\left(\boldsymbol{X}\right),P_T\left(\boldsymbol{X}\right)\right)$ can be defined as:*

$$\sup_{m,m'}\left|\int\left(P_S\left(\boldsymbol{X}\right)-P_T\left(\boldsymbol{X}\right)\right)\mathbb{E}_{y\sim P_S(\boldsymbol{Y}|\boldsymbol{X})}\left[I\left[m\left(\boldsymbol{X},\boldsymbol{Y}\right)\neq m'\left(\boldsymbol{X},\boldsymbol{Y}\right)\right]\right]dx\right|$$

*which denotes a posterior-weighted symmetric difference divergence between the source and target marginal distributions and measures the discrepancy between $P_S\left(\boldsymbol{X}\right)$ and $P_T\left(\boldsymbol{X}\right)$ under the influence of a fixed source posterior $P_S\left(\boldsymbol{Y}|\boldsymbol{X}\right)$.*

*Proof.* By the chain rule of the joint distribution, we have:

$$P\left(\boldsymbol{X},\boldsymbol{Y}\right)=P\left(\boldsymbol{X}\right)\cdot P\left(\boldsymbol{Y}|\boldsymbol{X}\right)$$

Consider the difference between two joint distributions, where $I\left[\cdot\right]$ denotes the indicator and the discrepancy between the hypothesis pairs $m,m'\in\mathcal{M}$ can be regarded as:

$$\triangle:=\sup_{m,m'}\left|\mathbb{E}_{(\boldsymbol{X},\boldsymbol{Y})\sim P_S(\boldsymbol{X},\boldsymbol{Y})}\left[I\left[m\left(\boldsymbol{X},\boldsymbol{Y}\right)\neq m'\left(\boldsymbol{X},\boldsymbol{Y}\right)\right]\right]-\mathbb{E}_{(\boldsymbol{X},\boldsymbol{Y})\sim P_T(\boldsymbol{X},\boldsymbol{Y})}\left[I\left[m\left(\boldsymbol{X},\boldsymbol{Y}\right)\neq m'\left(\boldsymbol{X},\boldsymbol{Y}\right)\right]\right]\right|$$

Then, the chain rule of distributions can be applied as follows:

$$\triangle=\sup\left|\int P_S\left(\boldsymbol{X}\right)P_S\left(\boldsymbol{Y}|\boldsymbol{X}\right)I\,dxdy-\int P_T\left(\boldsymbol{X}\right)P_T\left(\boldsymbol{Y}|\boldsymbol{X}\right)I\,dxdy\right|$$

where $I$ is short for $I\left[m\left(\boldsymbol{X},\boldsymbol{Y}\right)\neq m'\left(\boldsymbol{X},\boldsymbol{Y}\right)\right]$. By introducing the term involving $P_T\left(\boldsymbol{X}\right)P_S\left(\boldsymbol{Y}|\boldsymbol{X}\right)$, and applying the triangle inequality, we obtain:

$$\triangle\leq\left|\int\left(P_S\left(\boldsymbol{X}\right)-P_T\left(\boldsymbol{X}\right)\right)\mathbb{E}_{y\sim P_S(\boldsymbol{Y}|\boldsymbol{X})}\left[I\right]dx\right|+\left|\int P_T\left(\boldsymbol{X}\right)\left(\mathbb{E}_{y\sim P_S(\boldsymbol{Y}|\boldsymbol{X})}\left[I\right]-\mathbb{E}_{y\sim P_T(\boldsymbol{Y}|\boldsymbol{X})}\left[I\right]\right)dxdy\right|$$

So that, we can obtain:

$$\begin{aligned}d_{\mathcal{M}\triangle\mathcal{M}}\left(P_S\left(\boldsymbol{X},\boldsymbol{Y}\right),P_T\left(\boldsymbol{X},\boldsymbol{Y}\right)\right)\leq\quad &d_{\mathcal{M}\triangle\mathcal{M}}^{P_S(\boldsymbol{Y}|\boldsymbol{X})}\left(P_S\left(\boldsymbol{X}\right),P_T\left(\boldsymbol{X}\right)\right)\\&+\mathbb{E}_{x\sim P_T(\boldsymbol{X})}\left[d_{\mathcal{M}\triangle\mathcal{M}}\left(P_S\left(\boldsymbol{Y}|\boldsymbol{X}=x\right),P_T\left(\boldsymbol{Y}|\boldsymbol{X}=x\right)\right)\right]\end{aligned}$$

$\square$

**Theorem A.2** (Generalization Bound with the Joint Distributions). *For any hypothesis $h\in\mathcal{H}$ with an auxiliary class $\mathcal{M}$ where $h:\boldsymbol{X}\to\{-1,1\}\in\mathcal{H}$ and $m=yh\left(x\right):\boldsymbol{X}\times\boldsymbol{Y}\to\{-1,1\}\in\mathcal{M}$, the target domain risk can be bounded by:*

$$\begin{aligned}\epsilon_T(h)\leq\epsilon_S(h)+&d_{\mathcal{M}\triangle\mathcal{M}}^{P_S(\boldsymbol{Y}|\boldsymbol{X})}\left(P_S\left(\boldsymbol{X}\right),P_T\left(\boldsymbol{X}\right)\right)\\&+\mathbb{E}_{x\sim P_T(\boldsymbol{X})}\left[d_{\mathcal{M}\triangle\mathcal{M}}\left(P_S\left(\boldsymbol{Y}|\boldsymbol{X}=x\right),P_T\left(\boldsymbol{Y}|\boldsymbol{X}=x\right)\right)\right]+\lambda_{\mathcal{H}}\end{aligned}$$

where $\lambda_{\mathcal{H}}:=\inf_{h\in\mathcal{H}}\left[\epsilon_S\left(h\right)+\epsilon_T\left(h\right)\right]$. On the one hand, while retaining the original marginal discrepancy term $d_{\mathcal{M}\triangle\mathcal{M}}^{P_S(\boldsymbol{Y}|\boldsymbol{X})}\left(P_S(\boldsymbol{X}),P_T(\boldsymbol{X})\right)$, this extended bound enhances the classical formulation by incorporating a conditional divergence term that captures shifts in $P\left(\boldsymbol{Y}|\boldsymbol{X}\right)$, which are typically overlooked by input alignment alone. Moreover, based on Bayes rule $P\left(\boldsymbol{Y}|\boldsymbol{X}\right)=\frac{P(\boldsymbol{X}|\boldsymbol{Y})P(\boldsymbol{Y})}{P(\boldsymbol{X})}$, aligning the posteriors $P\left(\boldsymbol{Y}|\boldsymbol{X}\right)$ can effectively mitigate the influence of the label prior $P\left(\boldsymbol{Y}\right)$ on the decision function. Even in scenarios where $P\left(\boldsymbol{Y}\right)$ varies significantly *across domains*, maintaining consistency in $P\left(\boldsymbol{Y}|\boldsymbol{X}\right)$ helps preserve a stable and generalizable decision boundary. On the other hand, $d_{\mathcal{M}\triangle\mathcal{M}}^{P_S(\boldsymbol{Y}|\boldsymbol{X})}\left(P_S(\boldsymbol{X}),P_T(\boldsymbol{X})\right)$ characterizes the alignment of the marginal distributions under the source conditional distribution $P_S\left(\boldsymbol{Y}|\boldsymbol{X}\right)$. In other words, aligning only the marginals without taking the conditional distributions into account may distort the original representation structure, thereby weakening the discriminative information encoded in the feature space. Such distortions can in turn destabilize the decision boundary and hinder its ability to generalize across domains. Therefore, ensuring posterior consistency becomes crucial.

**Step 2: Extending to multiple source domains**

**Theorem A.3** (Generalization Bound with the Joint Distributions on multiple source domains)**.** *For any hypothesis $h \in \mathcal{H}$ with an auxiliary class $\mathcal{M}$, the target domain risk can be defined based on a linear mixture across multiple source domains:*

$$\epsilon_T(h) \leq \sum_{i=1}^{N} \pi_i \epsilon_S^i(h) + \underbrace{\zeta_{\text{in}} + \zeta_{\text{out}}}_{\text{prior distribution discrepancy}} + \underbrace{\eta_{\text{in}} + \eta_{\text{out}}}_{\text{posterior distribution discrepancy}} + \lambda_\pi$$

where $\sum_{i=1}^{N} \pi_i = 1$, $\lambda_\pi = \inf_{h \in \mathcal{H}} \left[ \sum_{i=1}^{N} \pi_i \epsilon_S^i(h), \epsilon_T(h) \right]$. $\zeta_{\text{in}}/\eta_{\text{in}}$ denotes the maximum $\mathcal{H}$-divergence between any pair of the prior/posterior distributions $P(\boldsymbol{X})/P(\boldsymbol{Y}|\boldsymbol{X}=x)$. $\zeta_{\text{out}} = d_{\mathcal{M}\triangle\mathcal{M}}^{P_S(\boldsymbol{Y}|\boldsymbol{X})} \left( \sum_{i=1}^{N} \pi_i P_S^i(\boldsymbol{X}), P_T(\boldsymbol{X}) \right)$ and $\eta_{\text{out}} = d_{\mathcal{M}\triangle\mathcal{M}} \left( \sum_{i=1}^{N} \pi_i P_S^i(\boldsymbol{Y}|\boldsymbol{X}=x), P_T(\boldsymbol{Y}|\boldsymbol{X}=x) \right)$.

*Proof.*

*Remark* A.4 (Bounding the $\mathcal{H}$-divergence between domains in the convex hull (Albuquerque et al., 2019))*.* Let a set $S$ of source domains such that $|S| = N$ be denoted by $P_S^i, i \in [N]$. The convex hull $\Lambda_S$ of $S$ is defined as the set of mixture distributions given by: $\Lambda_S = \left\{ \overline{P} := \overline{P}(\cdot) = \sum_{i=1}^{N} \pi_i P_S^i(\cdot), \pi_i \in \triangle_N \right\}$. Let $d_{\mathcal{H}\triangle\mathcal{H}} \left( P_S^i(\boldsymbol{X}), P_S^j(\boldsymbol{X}) \right) \leq \epsilon, \forall i, j \in [N]$. The following holds for the $\mathcal{H}$-divergence between any pair of domains $P', P'' \in \Lambda_S^2$:

$$d_{\mathcal{H}\triangle\mathcal{H}}(P', P'') \leq \epsilon$$

According to this remark, Albuquerque et al. (2019) further introduce unseen target $\overline{P}_T$, the element of $\Lambda_S$ which is closest to $P_T$, i.e., $\overline{P}_T$ is given by $\arg\min_{\pi_1, \ldots, \pi_N} d_{\mathcal{H}\triangle\mathcal{H}} \left( P_T, \sum_{i=1}^{N} \pi_i P_S^i \right)$.

Along this line, we extend this divergence analysis to the joint distributions $P_i(\boldsymbol{X}, \boldsymbol{Y}), \forall i \in [N]$, in order to capture both covariate and label shifts. Specifically, the covnex hull $\Lambda_S$ can be defined as $\Lambda_S = \left\{ \overline{P} := \overline{P}(\boldsymbol{X}, \boldsymbol{Y}) = \sum_{i=1}^{N} \pi_i P_S^i(\boldsymbol{X}, \boldsymbol{Y}), \pi_i \in \triangle_N \right\}$. This definition is well-motivated, as under the two-stage sampling perspective (Cao & Chen, 2024), one first samples a joint distribution from the real-world domain mixture, and then draws specific input–label pairs from the selected joint distribution.

Consequently, Eq. (7) can be reformulated in terms of a linear mixture with Lemma A.1:

$$\sum_{i=1}^{N} \pi_i \epsilon_T(h) \leq \sum_{i=1}^{N} \pi_i \left[ \epsilon_S^i(h) + d_{\mathcal{M}\triangle\mathcal{M}} \left( P_S^i(\boldsymbol{X}, \boldsymbol{Y}), P_T(\boldsymbol{X}, \boldsymbol{Y}) \right) \right] + \lambda_\pi$$

$$\leq \sum_{i=1}^{N} \pi_i \left[ \epsilon_S^i(h) + d_{\mathcal{M}\triangle\mathcal{M}} \left( P_S^i(\boldsymbol{X}, \boldsymbol{Y}), \overline{P}_T(\boldsymbol{X}, \boldsymbol{Y}) \right) \right.$$

$$\left. + d_{\mathcal{M}\triangle\mathcal{M}} \left( \overline{P}_T(\boldsymbol{X}, \boldsymbol{Y}), P_T(\boldsymbol{X}, \boldsymbol{Y}) \right) \right] + \lambda_\pi$$

$$\leq \sum_{i=1}^{N} \pi_i \left[ \epsilon_S^i(h) + d_{\mathcal{M}\triangle\mathcal{M}}^{P_S^i(\boldsymbol{X})} + \mathbb{E}_{x \sim P_T(\boldsymbol{X})} d_{\mathcal{M}\triangle\mathcal{M}}^{P_S^i(\boldsymbol{Y}|\boldsymbol{X}=x)} \right.$$

$$\left. + d_{\mathcal{M}\triangle\mathcal{M}}^{P_T(\boldsymbol{X})} + \mathbb{E}_{x \sim P_T(\boldsymbol{X})} d_{\mathcal{M}\triangle\mathcal{M}}^{P_T(\boldsymbol{Y}|\boldsymbol{X}=x)} \right] + \lambda_\pi$$

$$\leq \sum_{i=1}^{N} \pi_i \epsilon_S^i(h) + \zeta_{\text{in}} + \zeta_{\text{out}} + \eta_{\text{in}} + \eta_{\text{out}} + \lambda_\pi$$

where the last inequality is based on Remark A.4, and $d_{\mathcal{M}\triangle\mathcal{M}}^{P(\boldsymbol{X})} = d_{\mathcal{M}\triangle\mathcal{M}}^{P(\boldsymbol{Y}|\boldsymbol{X})} \left( P(\boldsymbol{X}), \overline{P}_T(\boldsymbol{X}) \right)$, $d_{\mathcal{M}\triangle\mathcal{M}}^{P(\boldsymbol{Y}|\boldsymbol{X}=x)} = d_{\mathcal{M}\triangle\mathcal{M}} \left( P(\boldsymbol{Y}|\boldsymbol{X}=x), \overline{P}_T(\boldsymbol{Y}|\boldsymbol{X}=x) \right)$. $\qquad \square$

**Step 3: Introducing a structural regularization term via margin surrogate error.** Note that the standard risk $\epsilon_S(h)$ measures the empirical error on the source domain. However, when $P(\boldsymbol{Y})$ is imbalanced, majority classes dominate the

optimization of the empirical loss, which biases the decision boundary toward them. As a result, the margin around minority samples shrinks, reducing classification confidence and increasing error on rare classes. To address this, we introduce a convex surrogate error (Yuan et al., 2021) on the source domain, denoted by $\gamma\left(h\right)$, defined as:

$$\gamma\left(h\right) := \phi\left(h_y\left(x\right) - \max_{k \neq y} h_k\left(x\right)\right)$$

where $\phi\left(\cdot\right)$ denotes a margin surrogate function, such as $\phi\left(z\right) = \max\left(0, -z\right)$ or $\log\left(1 + e^z\right)$. This term encourages the model to maintain a sufficiently large margin between classes.

Consequently, $\epsilon_S\left(h\right)$ can be decomposed through margin $\gamma\left(h\right)$ as follows:

$$\begin{aligned}
\epsilon_S\left(h\right) &= \mathbb{E}_{(x,y)\sim P_S(\boldsymbol{X},\boldsymbol{Y})}\ell\left(h\left(x\right), y\right) \\
&= \underbrace{\mathbb{E}_{(x,y)\sim P_S(\boldsymbol{X},\boldsymbol{Y})}\left[\ell\left(h\left(x\right), y\right) \cdot 1_{\gamma(h)\leq\delta}\right]}_{\text{small-margin loss}} + \underbrace{\mathbb{E}_{(x,y)\sim P_S(\boldsymbol{X},\boldsymbol{Y})}\left[\ell\left(h\left(x\right), y\right) \cdot 1_{\gamma(h)>\delta}\right]}_{\text{large-margin loss}} \\
&\leq \ell_{max} \cdot \Pr\left[\gamma\left(h\right) \leq \delta\right] + \mathbb{E}_{(x,y)\sim P_S(\boldsymbol{X},\boldsymbol{Y})}\left[\ell\left(h\left(x\right), y\right) \cdot 1_{\gamma(h)>\delta}\right] \\
&\leq \ell_{max} \cdot \Pr\left[\gamma\left(h\right) \leq \delta\right] + \epsilon_S
\end{aligned}$$

where $\delta$ is a margin threshold, $\ell_{max}$ denotes the upper bound of the loss function, and $\Pr\left[\gamma\left(h\right) \leq \delta\right]$ quantifies the proportion of samples with margin less than or equal to the threshold $\delta$.

Notably, this inequality holds under the assumption that the loss function is monotonically decreasing with respect to the margin $\gamma\left(h\right)$, whose property can be satisfied by commonly used surrogate losses, such as cross-entropy loss.

Combined with the above, the final upper bound can be defined as:

$$\begin{aligned}
\epsilon_T\left(h\right) &\leq \sum_{i=1}^{N} \pi_i \left(\epsilon_S^i\left(h\right) + \ell_{max}^i \cdot \Pr\left[\gamma^i\left(h\right) \leq \delta\right]\right) + \zeta_{\text{in}} + \zeta_{\text{out}} + \eta_{\text{in}} + \eta_{\text{out}} + \lambda_\pi \\
&\leq \sum_{i=1}^{N} \pi_i \epsilon_S^i\left(h\right) + \ell_{max} \cdot \Pr\left[\gamma\left(h\right) \leq \delta\right] + \zeta_{\text{in}} + \zeta_{\text{out}} + \eta_{\text{in}} + \eta_{\text{out}} + \lambda_\pi
\end{aligned}$$

where $\ell_{max}$ denotes the upper bound of the loss function across all source domains, and $\gamma\left(h\right)$ denotes the expected margin over the source domains.

## B. Gradient Derivation

Our proposed InfoNCE-like contrastive loss can be formulated as follows:

$$\mathcal{L}_{con}^{\text{InfoNCE-ND}} = \sum_{i\in\mathcal{I}} -\log\left\{\frac{1}{|\mathcal{N}\left(i\right)|} \sum_{n\in\mathcal{N}(i)} \frac{1 - \mathrm{s}\left(\boldsymbol{p}_i, \boldsymbol{p}_n\right)}{\sum_{a\in\mathcal{A}(i)}\left(1 - \mathrm{s}\left(\boldsymbol{p}_i, \boldsymbol{p}_a\right)\right)}\right\}$$

In analogy, SupCon-like contrastive loss can be formulated as follows:

$$\mathcal{L}_{con}^{\text{SupCon-ND}} = \sum_{i\in\mathcal{I}} -\frac{1}{|\mathcal{N}\left(i\right)|} \sum_{n\in\mathcal{N}(i)} \log\left\{\frac{1 - \mathrm{s}\left(\boldsymbol{p}_i, \boldsymbol{p}_n\right)}{\sum_{a\in\mathcal{A}(i)}\left(1 - \mathrm{s}\left(\boldsymbol{p}_i, \boldsymbol{p}_a\right)\right)}\right\}$$

And, classical InfoNCE contrastive loss dominated by positives can be formulated as follows:

$$\mathcal{L}_{con}^{\text{InfoNCE}} = \sum_{i\in\mathcal{I}} -\log\left\{\frac{1}{|\mathcal{P}\left(i\right)|} \sum_{p\in\mathcal{P}(i)} \frac{\mathrm{s}\left(\boldsymbol{p}_i, \boldsymbol{p}_p\right)}{\sum_{a\in\mathcal{A}(i)} \mathrm{s}\left(\boldsymbol{p}_i, \boldsymbol{p}_a\right)}\right\}$$

Let $f_{ip} = \mathrm{s}\left(\boldsymbol{p}_i, \boldsymbol{p}_p\right)$, $Z_i = \sum_{a \in \mathcal{A}(i)} \mathrm{s}\left(\boldsymbol{p}_i, \boldsymbol{p}_a\right)$, and $Q_i = \frac{1}{|\mathcal{P}(i)|} \sum_{p \in \mathcal{P}(i)} \frac{f_{ip}}{Z_i}$. Then, the derivation of $\mathcal{L}_{con}^{\text{InfoNCE}}$ w.r.t. $\boldsymbol{p}_i$ can be expressed as:

$$
\begin{aligned}
\frac{\partial \mathcal{L}_{con}^{\text{InfoNCE}}}{\partial \boldsymbol{p}_i} = \nabla_{\boldsymbol{p}_i} \mathcal{L}_{con}^{\text{InfoNCE}} &= -\frac{1}{Q_i} \cdot \nabla_{\boldsymbol{p}_i} Q_i \\
&= -\frac{1}{Q_i} \cdot \frac{1}{|\mathcal{P}(i)|} \sum_{p \in \mathcal{P}(i)} \nabla_{\boldsymbol{p}_i} \left( \frac{f_{ip}}{Z_i} \right) \\
&\stackrel{\text{Quotient rule}}{=} -\frac{1}{Q_i} \cdot \frac{1}{|\mathcal{P}(i)|} \sum_{p \in \mathcal{P}(i)} \frac{\nabla_{\boldsymbol{p}_i} f_{ip} \cdot Z_i - f_{ip} \cdot \nabla_{\boldsymbol{p}_i} Z_i}{Z_i^2} \\
&= -\frac{1}{Q_i} \cdot \frac{1}{|\mathcal{P}(i)|} \cdot \frac{1}{Z_i^2} \sum_{p \in \mathcal{P}(i)} \left[ \nabla_{\boldsymbol{p}_i} f_{ip} \cdot Z_i - f_{ip} \cdot \nabla_{\boldsymbol{p}_i} Z_i \right]
\end{aligned}
$$

Next, we organize the final summation terms, and let $\Psi = \sum_{p \in \mathcal{P}(i)} \left[ \nabla_{\boldsymbol{p}_i} f_{ip} \cdot Z_i - f_{ip} \cdot \nabla_{\boldsymbol{p}_i} Z_i \right]$ and $f_{in} = \mathrm{s}\left(\boldsymbol{p}_i, \boldsymbol{p}_n\right)$:

$$
\begin{aligned}
\Psi &= \sum_{p \in \mathcal{P}(i)} \left[ \nabla_{\boldsymbol{p}_i} f_{ip} \cdot Z_i - f_{ip} \cdot \left( \sum_{p' \in \mathcal{P}(i)} \nabla_{\boldsymbol{p}_i} f_{ip'} + \sum_{n \in \mathcal{N}(i)} \nabla_{\boldsymbol{p}_i} f_{in} \right) \right] \\
&= \sum_{p \in \mathcal{P}(i)} \nabla_{\boldsymbol{p}_i} f_{ip} \cdot Z_i - \sum_{p \in \mathcal{P}(i)} \sum_{p' \in \mathcal{P}(i)} \nabla_{\boldsymbol{p}_i} f_{ip'} \cdot f_{ip} - \sum_{p \in \mathcal{P}(i)} \sum_{n \in \mathcal{N}(i)} \nabla_{\boldsymbol{p}_i} f_{in} \cdot f_{ip} \\
&= \sum_{p \in \mathcal{P}(i)} \nabla_{\boldsymbol{p}_i} f_{ip} \cdot Z_i - \sum_{p' \in \mathcal{P}(i)} \nabla_{\boldsymbol{p}_i} f_{ip'} \cdot \sum_{p \in \mathcal{P}(i)} f_{ip} - \sum_{n \in \mathcal{N}(i)} \nabla_{\boldsymbol{p}_i} f_{in} \cdot \sum_{p \in \mathcal{P}(i)} f_{ip} \\
&= \sum_{p \in \mathcal{P}(i)} \nabla_{\boldsymbol{p}_i} f_{ip} \cdot Z_i - \sum_{p \in \mathcal{P}(i)} \nabla_{\boldsymbol{p}_i} f_{ip} \cdot \sum_{p \in \mathcal{P}(i)} f_{ip} - \sum_{n \in \mathcal{N}(i)} \nabla_{\boldsymbol{p}_i} f_{in} \cdot \sum_{p \in \mathcal{P}(i)} f_{ip} \\
&= -\left( \sum_{p \in \mathcal{P}(i)} \nabla_{\boldsymbol{p}_i} f_{ip} \cdot \left( \sum_{p \in \mathcal{P}(i)} f_{ip} - Z_i \right) + \sum_{n \in \mathcal{N}(i)} \nabla_{\boldsymbol{p}_i} f_{in} \cdot \sum_{p \in \mathcal{P}(i)} f_{ip} \right) \\
&= -\left( -\sum_{p \in \mathcal{P}(i)} \nabla_{\boldsymbol{p}_i} f_{ip} \cdot \sum_{n \in \mathcal{N}(i)} f_{in} + \sum_{n \in \mathcal{N}(i)} \nabla_{\boldsymbol{p}_i} f_{in} \cdot \sum_{p \in \mathcal{P}(i)} f_{ip} \right)
\end{aligned}
$$

Therefore, we can obtain:

$$
\begin{aligned}
\frac{\partial \mathcal{L}_{con}^{\text{InfoNCE}}}{\partial \boldsymbol{p}_i} &= -\frac{1}{Q_i} \cdot \frac{1}{|\mathcal{P}(i)|} \cdot \frac{1}{Z_i^2} \left[ -\left( -\sum_{p \in \mathcal{P}(i)} \nabla_{\boldsymbol{p}_i} f_{ip} \cdot \sum_{n \in \mathcal{N}(i)} f_{in} + \sum_{n \in \mathcal{N}(i)} \nabla_{\boldsymbol{p}_i} f_{in} \cdot \sum_{p \in \mathcal{P}(i)} f_{ip} \right) \right] \\
&= \frac{1}{Q_i} \cdot \frac{1}{|\mathcal{P}(i)|} \cdot \frac{1}{Z_i} \left[ -\sum_{p \in \mathcal{P}(i)} \nabla_{\boldsymbol{p}_i} f_{ip} \cdot \frac{\sum_{n \in \mathcal{N}(i)} f_{in}}{Z_i} + \sum_{n \in \mathcal{N}(i)} \nabla_{\boldsymbol{p}_i} f_{in} \cdot \frac{\sum_{p \in \mathcal{P}(i)} f_{ip}}{Z_i} \right] \\
&= \frac{1}{Z_i} \left[ -\sum_{p \in \mathcal{P}(i)} \nabla_{\boldsymbol{p}_i} f_{ip} \cdot \frac{\sum_{n \in \mathcal{N}(i)} f_{in}}{\sum_{p \in \mathcal{P}(i)} f_{ip}} + \sum_{n \in \mathcal{N}(i)} \nabla_{\boldsymbol{p}_i} f_{in} \right] \\
&= \frac{1}{Z_i} \left[ -\sum_{p \in \mathcal{P}(i)} \nabla_{\boldsymbol{p}_i} \mathrm{s}\left(\boldsymbol{p}_i, \boldsymbol{p}_p\right) \cdot \frac{\sum_{n \in \mathcal{N}(i)} f_{in}}{\sum_{p \in \mathcal{P}(i)} f_{ip}} + \sum_{n \in \mathcal{N}(i)} \nabla_{\boldsymbol{p}_i} \mathrm{s}\left(\boldsymbol{p}_i, \boldsymbol{p}_n\right) \right]
\end{aligned}
$$

Similar to $\mathcal{L}_{con}^{\text{InfoNCE}}$, the gradient of $\mathcal{L}_{con}^{\text{InfoNCE-ND}}$ can be expressed in terms of the notations $f'_{ip} = 1 - \mathrm{s}\left(\boldsymbol{p}_i, \boldsymbol{p}_p\right)$, $f'_{in} =$

$1 - \mathrm{s}\,(\boldsymbol{p}_i, \boldsymbol{p}_n)$, $Z'_i = \sum_{a \in \mathcal{A}(i)} (1 - \mathrm{s}\,(\boldsymbol{p}_i, \boldsymbol{p}_a))$, and $Q'_i = \frac{1}{|\mathcal{N}(i)|} \sum_{n \in \mathcal{N}(i)} \frac{f'_{in}}{Z'_i}$:

$$
\begin{aligned}
\frac{\partial \mathcal{L}_{con}^{\text{InfoNCE-ND}}}{\partial \boldsymbol{p}_i} &= \frac{1}{Z'_i} \left[ \sum_{p \in \mathcal{P}(i)} \nabla_{\boldsymbol{p}_i} f'_{ip} - \sum_{n \in \mathcal{N}(i)} \nabla_{\boldsymbol{p}_i} f'_{in} \cdot \frac{\sum_{p \in \mathcal{P}(i)} f'_{ip}}{\sum_{n \in \mathcal{N}(i)} f'_{in}} \right] \\
&= \frac{1}{Z'_i} \left[ - \sum_{p \in \mathcal{P}(i)} \nabla_{\boldsymbol{p}_i} \mathrm{s}\,(\boldsymbol{p}_i, \boldsymbol{p}_p) + \sum_{n \in \mathcal{N}(i)} \nabla_{\boldsymbol{p}_i} \mathrm{s}\,(\boldsymbol{p}_i, \boldsymbol{p}_n) \cdot \frac{\sum_{p \in \mathcal{P}(i)} f'_{ip}}{\sum_{n \in \mathcal{N}(i)} f'_{in}} \right]
\end{aligned}
$$

These two gradient formulations exhibit fundamentally different optimization focuses, each expressed as a term modulated by a weighted coefficient. $\partial \mathcal{L}_{con}^{\text{InfoNCE}} / \partial \boldsymbol{p}_i$ adaptively emphasizes positive pair tightening by amplifying gradients for hard positives with low similarity or surrounded by similar negatives, while applying uniform repulsion to all negatives. In contrast, $\mathcal{L}con^{\text{InfoNCE-ND}}$ prioritizes hard negative repulsion by adaptively scaling negative gradients according to overall similarity, while applying uniform attraction to all positives. This contrast reveals a fundamental shift: **from enforcing positive compactness to enhancing negative separation.** This distinction is particularly relevant in the context of class imbalance. Under label shift, minority-class samples tend to be absorbed by majority-class clusters, requiring stronger emphasis on negative separation. $\mathcal{L}con^{\text{InfoNCE-ND}}$ addresses this issue by amplifying the repulsion of confusing majority-class negatives when $\sum_{n \in \mathcal{N}(i)} f_{in}$ is small, thereby preserving minority-class decision boundaries. In contrast, the gradient of $\mathcal{L}con^{\text{InfoNCE}}$ can explode when $\sum p \in \mathcal{P}(i) f_{ip}$ becomes vanishingly small, which is exacerbated by the scarcity of positive samples in minority classes. Consequently, $\mathcal{L}_{con}^{\text{InfoNCE-ND}}$ appears more suitable for IDG.

Finally, the gradient of $\mathcal{L}_{con}^{\text{SupCon-ND}}$ can also be expressed in terms of $f'_{ip} = 1 - \mathrm{s}\,(\boldsymbol{p}_i, \boldsymbol{p}_p)$, $f'_{in} = 1 - \mathrm{s}\,(\boldsymbol{p}_i, \boldsymbol{p}_n)$, and $Z'_i = \sum_{a \in \mathcal{A}(i)} (1 - \mathrm{s}\,(\boldsymbol{p}_i, \boldsymbol{p}_a))$:

$$
\begin{aligned}
\frac{\partial \mathcal{L}_{con}^{\text{SupCon-ND}}}{\partial \boldsymbol{p}_i} &= \frac{1}{Z'_i} \left[ \sum_{p \in \mathcal{P}(i)} \nabla_{\boldsymbol{p}_i} f'_{ip} - \sum_{n \in \mathcal{N}(i)} \nabla_{\boldsymbol{p}_i} f'_{in} \cdot \left( \frac{Z_i}{|\mathcal{N}(i)| \sum_{n \in \mathcal{N}(i)} f_{in}} - 1 \right) \right] \\
&= \frac{1}{Z'_i} \left[ - \sum_{p \in \mathcal{P}(i)} \nabla_{\boldsymbol{p}_i} \mathrm{s}\,(\boldsymbol{p}_i, \boldsymbol{p}_p) + \sum_{n \in \mathcal{N}(i)} \nabla_{\boldsymbol{p}_i} \mathrm{s}\,(\boldsymbol{p}_i, \boldsymbol{p}_n) \cdot \left( \frac{Z'_i}{|\mathcal{N}(i)| \sum_{n \in \mathcal{N}(i)} f'_{in}} - 1 \right) \right]
\end{aligned}
$$

Similar to the conclusion in SupCon (Khosla et al., 2020), this gradient enforces equal treatment across all negatives, which can suppress the gradient signal from rare but critical minority-class negatives. In contrast, our proposed $\mathcal{L}_{con}^{\text{InfoNCE-ND}}$ naturally emphasizes closer negatives, thereby more confusing, leading to stronger repulsion for hard negatives and attenuated updates for already-separated ones. This dynamic aligns well with the goal of maintaining class separation in IDG. In summary, while SupCon emphasizes balanced treatment of positives, we shift focus by dynamically weighting negatives based on similarity. This inversion transforms SupCon's design into a complementary strength under imbalance, enabling targeted repulsion of hard negatives and improved minority-class separation. Meanwhile, an empirical experiment has been provided in E.5.

## C. Algorithm

Note that the number of the augmented negative set $\widehat{\mathcal{N}}_k$ is $\hat{n}_k$, which is inversely proportional to the number of observed samples of the $k$-th class across all domains. The underlying intuition is that the classifier is easier to learn on the major classes.

Thus, $p_k$ in Eq. (2) is *not a tunable hyper-parameter*, which would be dynamically determined in the current mini-batch. For example, when mining hard negatives for class 1, we first aim to augment $\hat{n}_k = 100$ instances. Then, we sort all instances by their predicted probability $p_1$, and subsequently select the *bottom quarter of those* labeled as class 1, e.g., 10 instances, as $X_k^{\text{low}}$. Accordingly, $X_k^{\text{high}}$ consists of the top 10 non-class-1 instances with the highest predicted $p_1$ scores, since $100/10 = 10$. This adaptive mechanism enables the dynamic selection of boundary samples per class and per iteration, ensuring robustness to both class imbalance and evolving model predictions.

---

**Algorithm 1** NDCL

---

**Input:** Training set, batch size $B$, number of source domains $N$, number of class $K$, maximum iteration $T$, the trade-offs $\alpha$ and $\beta$, and Beta distribution parameter $\rho$

**Parameters:** A learnable network $f_\theta$

**Output:** $f_\theta$

1: **for** $t \leftarrow 1$ to $T$ **do**
2:     Randomly sample a batch $\boldsymbol{S} = \{(\boldsymbol{x}_b, y_b, d_b)\}_{b=1}^{B}$
3:     $\boldsymbol{p}_b \leftarrow f_\theta(\boldsymbol{x}_b)$
    *# calculate sub-objective function $\mathcal{L}_{ce}$*
4:     $\omega_b \leftarrow \frac{\exp(\ell_{ce}(\boldsymbol{p}_b, y_b))}{\sum_{\boldsymbol{x}_i \in \boldsymbol{X}_k} \exp(\ell_{ce}(\boldsymbol{p}_i, k))}$
5:     $\mathcal{L}_{ce} \leftarrow$ Eq. (4) with $\{(\boldsymbol{p}_b, y_b, \omega_b)\}_{b=1}^{B}$
    *# calculate sub-objective function $\mathcal{L}_{const}$*
6:     $\boldsymbol{\mu}_k^d \leftarrow \frac{1}{n_k^d} \sum_{\boldsymbol{x}_i \in \boldsymbol{X}_k^d} \boldsymbol{p}_i$
7:     $\mathcal{L}_{const} \leftarrow$ Eq. (3) with $\{\boldsymbol{\mu}_k^d\}_{d=1,\ldots,N; k=1,\ldots,K}$
    *# calculate sub-objective function $\mathcal{L}_{con}$*
8:     **for** a **do**
9:         resort all instances by $p_k$ in batch $\boldsymbol{S}$
10:       $\widehat{\mathcal{N}}_k \leftarrow$ Eq. (2)
11:       $(\widehat{\boldsymbol{p}}_k)_i \leftarrow f_\theta((\boldsymbol{x}_k)_i), \forall (\boldsymbol{x}_k)_i \in \widehat{\mathcal{N}}_k$
12:     **end for**
13:     $\mathcal{L}_{con} \leftarrow$ Eq. (1) with $\bigcup_{k=1}^{K} \{(\widehat{\boldsymbol{p}}_k)_i\}_{i=1}^{\hat{n}_k}$
    *# total loss*
14:     $\mathcal{L}_{total} \leftarrow \mathcal{L}_{ce} + \alpha \mathcal{L}_{con} + \beta \mathcal{L}_{const}$
15:     Update network parameters
16: **end for**

---

## D. Related Works

Existing methods in conventional Domain Generalization can be divided into four taxonomies. *1) Domain-invariant representation (DIR).* MMD (Li et al., 2018b) leveraged the kernel mean embedding to align arbitrary order moments of two distributions. CORAL (Sun & Saenko, 2016) achieved good performance by transferring only second-order moment. DANN (Ganin et al., 2016) introduced an additional domain classifier to interfere with the representations, so that they do not contain domain information. IRM (Arjovsky et al., 2019) constructed a causal diagram from a causal perspective to obtain invariant representations of causality. RDM (Nguyen et al., 2024) aligned risk distribution to indirectly eliminate domain information in the representation space. TCRI (Salaudeen & Koyejo, 2024) proposed a causal strategy to learn domain general representation, which is similar to DIR. *2) Data augmentation.* Mixup (Yan et al., 2020) provided more interpolated instances to enhance the smoothness, thereby improving generalizability. EDM (Cao & Chen, 2024) enlarged the supported region spanned by source domains through generated extrapolated domains. *3) Optimization-based.* Fish (Shi et al., 2022) proposed aligning gradients across domains by minimizing domain-specific gradient discrepancies. PGrad (Wang et al., 2023) leveraged principal gradients to align feature distributions across multiple source domains, enabling better transferability. *4) Meta learning.* MLDG (Li et al., 2018a) utilized the meta-learning to simulate the marginal shift between domains. Despite their remarkable success, these methods appear insufficient to fully address the complex challenges posed by the entangled domain and label shifts in IDG.

Recent studies have begun tackling IDG through a divide-and-conquer strategy. Xia et al. (2023) propose a two-stage framework focused on synthesizing reliable samples to compensate for minority domains or classes, while incorporating domain-invariant representation learning via adversarial training. Su et al. (2024) seek flat local minima while amplifying gradients for low-confidence samples, leveraging domain-invariant representations to address long-tailed data across domains. While intuitive, the former incurs substantial computational overhead, and the latter does not effectively address those coupling effects. Moreover, the inherent limitation of domain-invariant representations, where target domains may lie outside the support of source domains (Cao & Chen, 2024), is often exacerbated under such coupling. In contrast, Yang et al. (2022) propose a unified method that adaptively re-weights each domain-class pair through a designed transferability graph in the representation space to jointly address both shifts. this method remains biased toward majority classes, and its

overemphasis on underrepresented minority classes may lead to overfitting and degraded generalization performance. In this paper, we first present the generalization bound for IDG to reveal the key factors that influence the generalization of the target domain. Building on this theoretical insight, our designed NDCL leverages negative signals to naturally repel samples from neighboring classes, thereby enhancing generalization. In addition, **this design cleverly incorporates the structure of InfoNCE objective**, without requiring any modifications to the original data distribution.

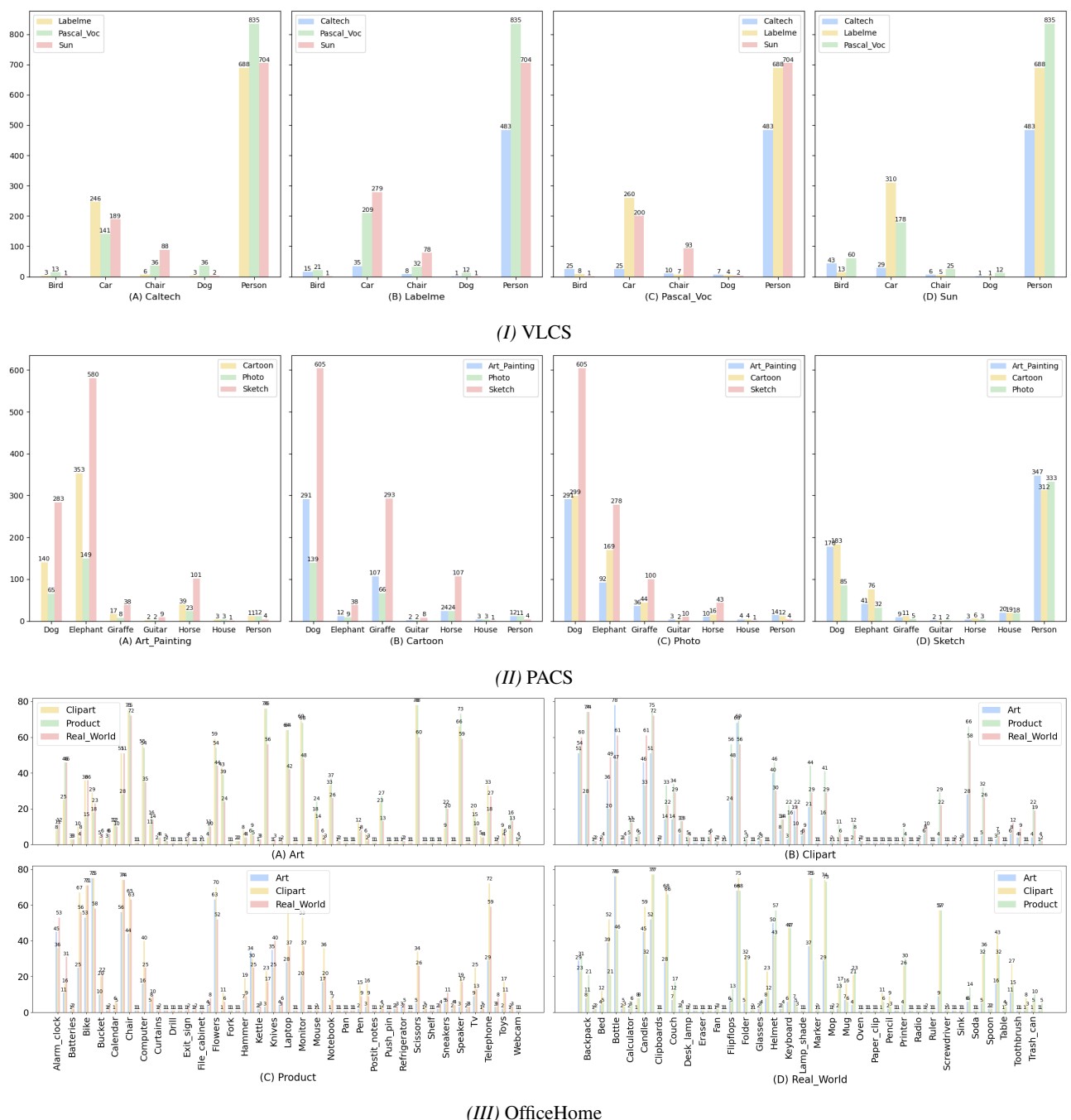

*Figure 5.* TotalHeavyTail setting on three Benchmark.

# E. Addtional Experiments

In this section, we provide a more detailed experimental analysis.

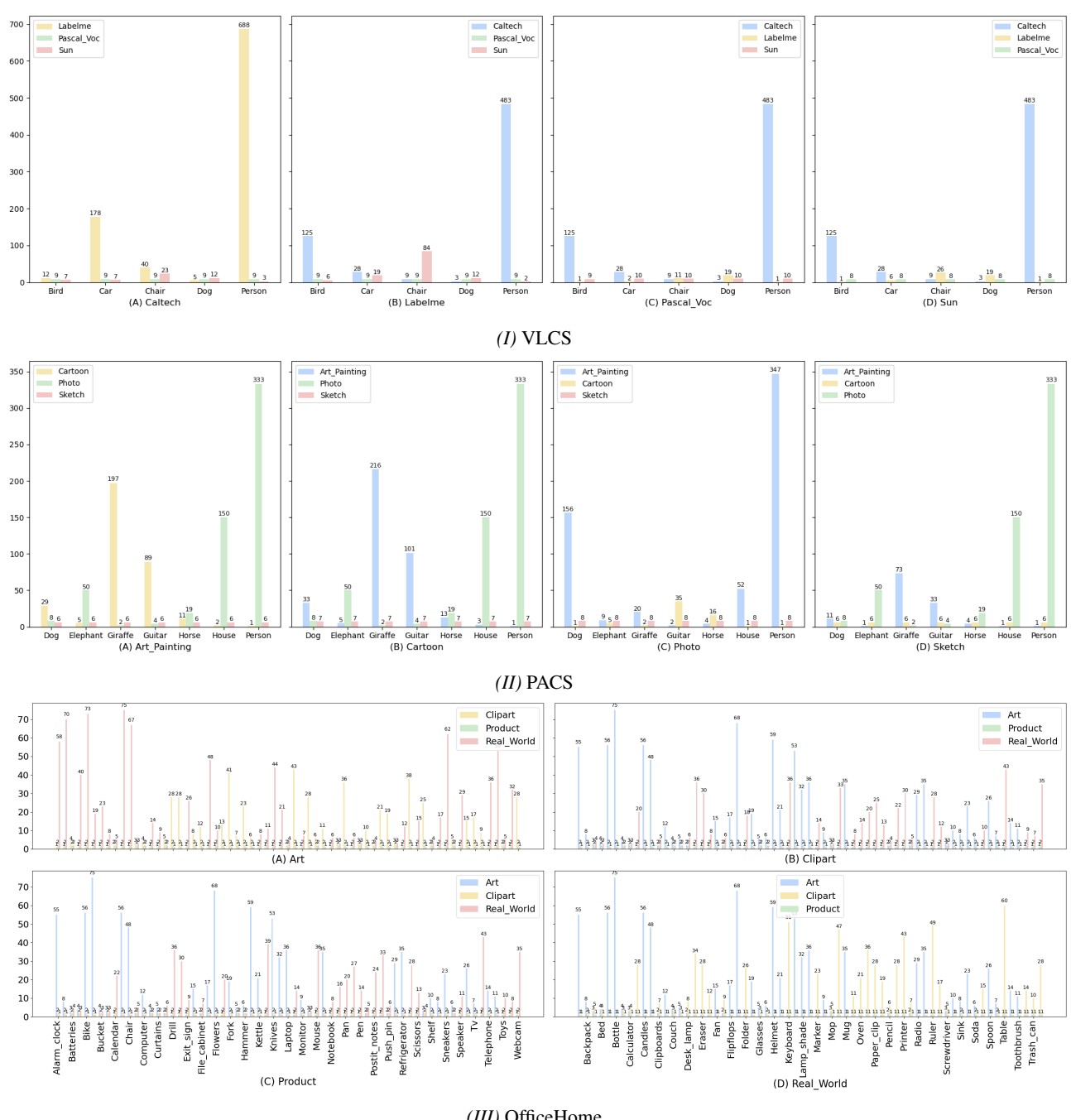

*Figure 6.* Duality setting on three Benchmark.

## E.1. Datasets and Settings

VLCS is a widely used domain generalization benchmark that combines four distinct datasets: VOC2007 (V), LabelMe (L), Caltech (C), and SUN09 (S). It contains 5 object categories shared across domains, with significant variations in scene style and object depiction, making it suitable for evaluating cross-domain recognition performance.

PACS is a more challenging DG dataset consisting of four visually diverse domains: Photo (P), Art painting (A), Cartoon (C), and Sketch (S). It includes 7 object categories and features large domain shifts due to differences in texture, abstraction, and artistic style.

*Table 5.* Statistical information about imbalance ratios across all benchmarks under three different settings. CR denotes the overall class ratio, i.e., $\frac{\max\left(\{n_k\}_{k=1}^{K}\right)}{\min\left(\{n_k\}_{k=1}^{K}\right)}$. DR denotes the sampling ratio across domains, i.e., $\frac{\max\left(\{N^d\}_{d=1}^{N}\right)}{\min\left(\{N^d\}_{d=1}^{N}\right)}$. ECR denotes the class ratio in each source domain, i.e., $\frac{\max\left(\{n_k^d\}_{k=1}^{K}\right)}{\min\left(\{n_k^d\}_{k=1}^{K}\right)}$.

| | | GINIDG | | | TotalHeavyTail | | | Duality | | |
| --- | --- | --- | --- | --- | --- | --- | --- | --- | --- | --- |
| | | CR | DR | ECR | CR | DR | ECR | CR | DR | ECR |
| VLCS | C | 10.15 | 1.38 | [30.95, 90.28, 4.90] | 131.00 | 1.21 | [229.33, 64.23, 704.00] | 26.92 | 20.51 | [137.60, 1.00, 7.67] |
| | L | 10.00 | 2.45 | [14.08, 99.42, 7.36] | 144.42 | 2.04 | [483.00, 69.58, 704.00] | 20.58 | 14.40 | [161.00, 1.00, 42.00] |
| | S | 10.00 | 2.25 | [19.32, 42.87, 5.22] | 143.28 | 1.97 | [483.00, 688.00, 69.58] | 16.40 | 16.20 | [161.00, 26.00, 1.00] |
| | V | 26.66 | 2.08 | [13.23, 33.36, 107.14] | 144.23 | 1.81 | [69.00, 172.00, 704.00] | 15.43 | 16.20 | [161.00, 22.00, 1.11] |
| PACS | A | 12.50 | 3.35 | [8.54, 3.65, 46.44] | 154.57 | 3.88 | [176.50, 74.50, 580.00] | 26.92 | 20.51 | [137.60, 1.00, 7.67] |
| | C | 12.36 | 3.35 | [7.98, 4.12, 48.27] | 147.86 | 4.16 | [145.50, 69.50, 605.00] | 20.58 | 14.40 | [161.00, 1.00, 42.00] |
| | P | 13.00 | 2.34 | [7.27, 8.20, 40.75] | 132.78 | 2.31 | [97.00, 149.50, 605.00] | 16.40 | 16.20 | [161.00, 26.00, 1.00] |
| | S | 16.50 | 1.80 | [19.47, 25.00, 9.00] | 198.40 | 1.27 | [173.50, 312.00, 166.50] | 13.60 | 13.47 | [73.00, 1.00, 166.50] |
| OfficeHome | A | - | - | - | 74.00 | 1.16 | [78.00, 78.00, 72.00] | 9.75 | 7.42 | [43.00, 1.00, 75.00] |
| | C | - | - | - | 66.00 | 1.54 | [78.00, 75.00, 74.00] | 9.75 | 7.04 | [75.00, 1.00, 43.00] |
| | P | - | - | - | 69.33 | 1.63 | [75.00, 75.00, 74.00] | 9.75 | 7.04 | [75.00, 1.00, 43.00] |
| | R | - | - | - | 70.33 | 1.80 | [76.00, 77.00, 77.00] | 9.62 | 14.08 | [75.00, 60.00, 1.00] |

OfficeHome is another more challenging benchmark for domain generalization, consisting of 65 categories across four diverse domains: Art, Clipart, Product, and Real-World. It exhibits substantial domain discrepancies and category imbalance, thus providing a comprehensive evaluation setting for generalization algorithms.

Since existing works do not provide standardized data splitting protocols, particularly on the OfficeHome Benchmark, and typicallyomit performance breakdowns across both coarse-and-fine-grained class levels (Su et al., 2024; Xia et al., 2023), it becomes challenging to ensure systematic evaluation and reproducibility. To fill this gap, we provide a script, namely idg_generate.py, in the code folder of the supplementary materials, which enables the generation of diverse and controllable dataset configurations.

```
1    # class TotalHeavyTail() and main() are defined in idg_generate.py
2    generator = TotalHeavyTail(num_of_validation, percent_of_test, heavytail_distribution_parameter)
3    stats = main('PACS', num_of_validation, thred_many, thred_few, generator, file=sys.stdout)
```

*Listing 1.* Example of generating the TotalHeavyTail setting on PACS

TotalHeavyTail and Duality are two representative settings generated by this script, where Fig. 5 and Fig. 6 visualize the number of training samples per class in each setting, respectively. Specifically, the TotalHeavyTail setting represents a cross-domain consistent long-tailed distribution, with relatively balanced domain sampling. In contrast, the Duality setting introduces a symmetric long-tailed distribution with noticeable domain-level sampling discrepancies. Tab. 5 compares the imbalance ratios across the three settings used in the main experiments. It can be observed that, compared to the GINIDG setting, our TotalHeavyTail and Duality settings place increased demands on model robustness and generalization capacity.

We adopt the DomainBed protocol (Gulrajani & Lopez-Paz, 2021) to reproduce all methods, aiming to systematically evaluate which approaches are effective under which settings. ResNet-50 architecture has been adopted as the backbone, and each input image is resized and cropped to $224\times224$. Adam is utilized as the optimizer. Training data is randomly sampled from the full source domains under each setting. Unlike the original DomainBed training-domain validation protocol, which utilizes the remaining data for model selection, we utilize **a cross-domain balanced validation set** for model selection. And, the remaining data from the training domains serves as the in-distribution test set, while the held-out domains are used as target domains. Batch size is set to 32 for each source domain and 64 for test. Maximum iteration $T$ is set to 5,000. The default learning rate is set to $5e$-5. The range of the trade-offs involved in each method can be referred to the supplementary material. All codes are run on Python 3.9, PyTorch 1.13 on Arch Linux with NVIDIA GeForce RTX 4090 GPUs.

### E.2. Illustrating IDG Failure Modes on Toy Data

In this subsection, we conduct additional experiments to demonstrate the challenges of IDG, mentioned in section 1.

To demonstrate **how heterogeneous long-tailed distributions can lead to misalignment**, we design a toy experiment on the PACS dataset with a symmetric cross-domain long-tail distribution. Specifically, an imbalance ratio of 400:1 is used, as illustrated in Fig. 7 (C), where person is the dominant class in Domain A and a minority class in Domain C, and vice versa

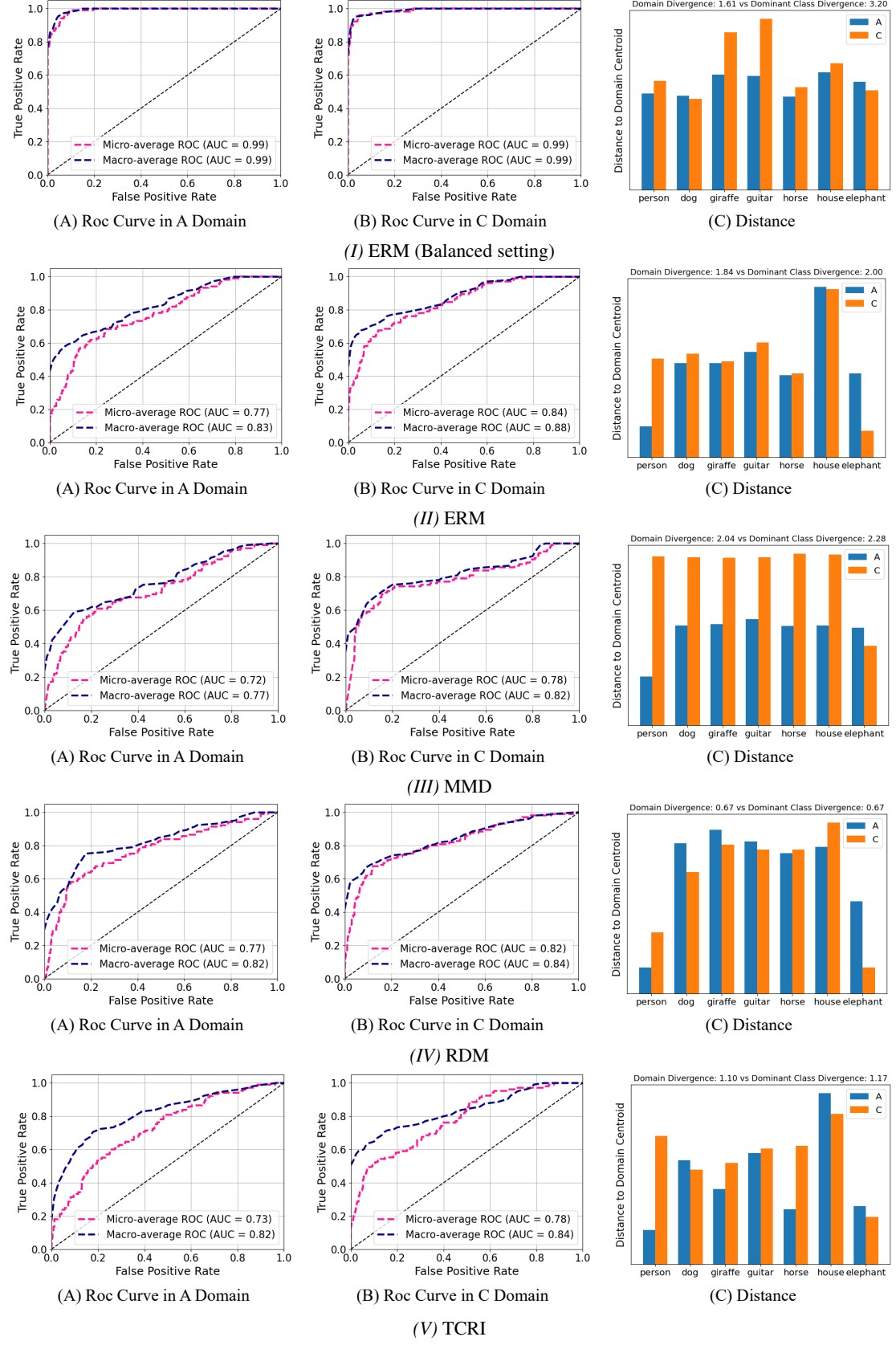

*Figure 7.* An illustration of missalignment caused by heterogeneous long-tailed distributions. (A) and (B) illustrates the Roc curves on the validation set for each source domain. (C) illustrates the distance between each class and its corresponding domain centroid. In Domain A, class sample sizes decrease from left to right along the x-axis, whereas in Domain C, they increase accordingly.

for the other classes. Several key observations emerge: 1) Under this imbalanced setting, the ROC-AUC on the validation set decreases significantly. 2) Employing domain alignment strategies further reduces the ROC-AUC, suggesting that majority classes dominate the alignment process and likely cause mismatches. 3) The dominant class consistently lies closest to its domain centroid, as illustrated by the bar plots in Fig. 7 (C). And, the Wasserstein distance between domains is comparable to the Wasserstein distance (Cao & Chen, 2024) between their dominant classes. These findings further imply that majority classes play a dominant role in the domain alignment process.

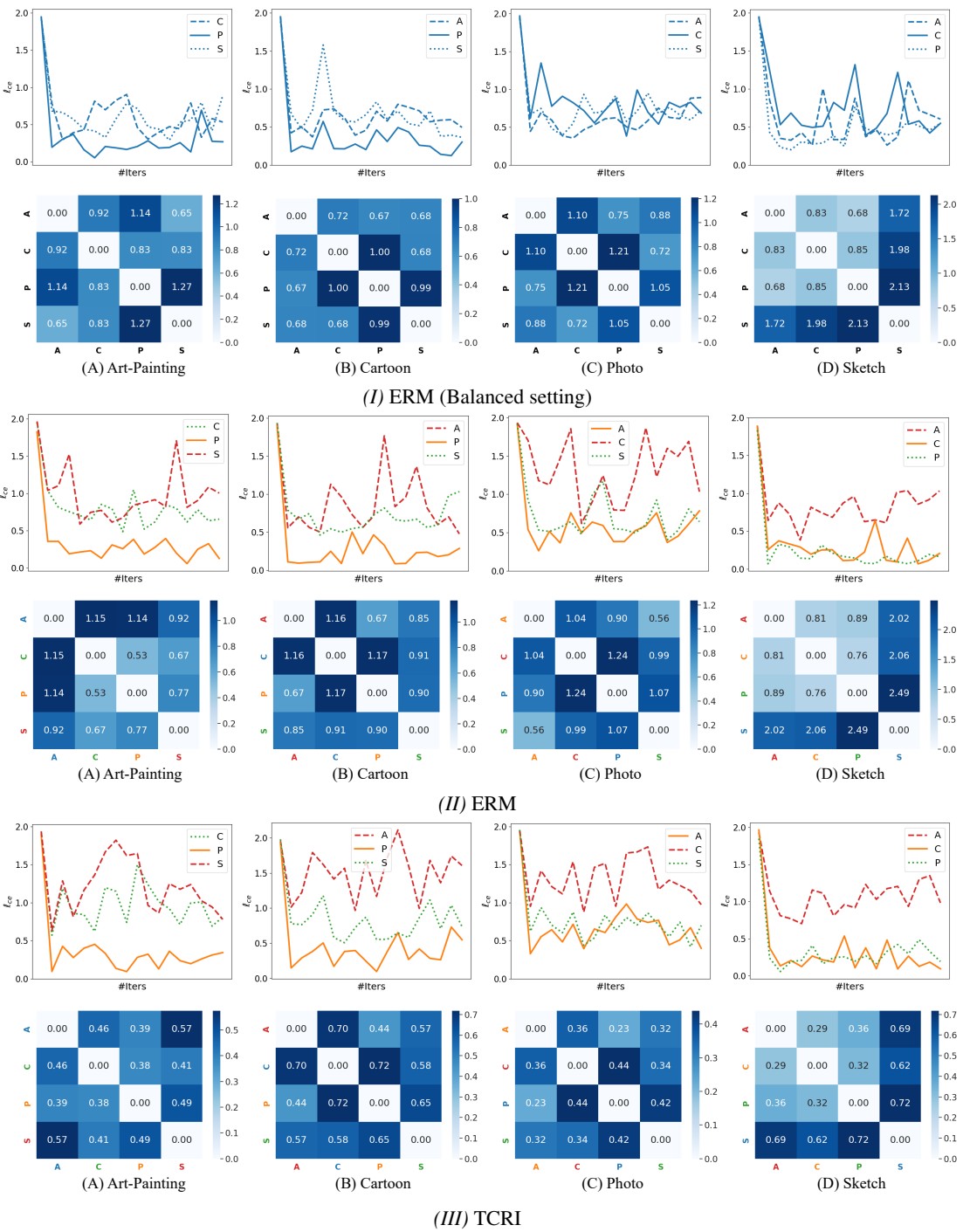

*Figure 8.* An illustration of underrepresented domains being absorbed by more populous ones. Minority, majority, and medium domains are color-coded in red, orange, and green, respectively. The line plot tracks the validation cross-entropy loss over training epochs for each source domain, while the heatmap quantifies inter-domain discrepancies under the optimal model, measured by the Wasserstein distance(Cao & Chen, 2024).

*Table 6.* Impact of domain-level resampling on BSoftmax under TotalHeavyTail and Duality settings.

| | TotalHeavyTail | | | | Duality | | | |
|---|---|---|---|---|---|---|---|---|
| | Average | Many | Medium | Few | Average | Many | Medium | Few |
| Applying | $48.6 \pm 0.5$ | $69.5 \pm 0.9$ | $61.9 \pm 0.5$ | $33.0 \pm 0.5$ | $52.8 \pm 0.5$ | $67.4 \pm 1.4$ | $62.1 \pm 0.6$ | $44.4 \pm 1.0$ |
| Omitting | $48.4 \pm 0.5$ | $69.1 \pm 0.7$ | $61.8 \pm 1.0$ | $32.4 \pm 0.5$ | $51.9 \pm 0.3$ | $65.4 \pm 1.2$ | $59.1 \pm 0.8$ | $43.7 \pm 0.3$ |

*Table 7.* Discrepancy of prior distributions in learned embeddings with on OfficeHome under TotalHeavyTail and Duality settings across several representative methods. SS indicates source-source divergence. ST indicates source-target divergence.

| | | ERM | MMD | RDM | PGrad | TCRI | BSoftmax | GINIDG | BoDA | NDCL (*ours*) |
|---|---|---|---|---|---|---|---|---|---|---|
| TotalHeavyTail | SS | $0.45 \pm 0.02$ | $0.43 \pm 0.05$ | $0.55 \pm 0.09$ | $0.36 \pm 0.06$ | $0.17 \pm 0.03$ | $0.38 \pm 0.06$ | $0.77 \pm 0.16$ | $0.13 \pm 0.00$ | $0.30 \pm 0.03$ |
| | ST | $0.99 \pm 0.08$ | $1.03 \pm 0.15$ | $1.37 \pm 0.13$ | $0.78 \pm 0.13$ | $0.38 \pm 0.07$ | $0.99 \pm 0.14$ | $2.32 \pm 0.19$ | $0.29 \pm 0.01$ | $0.69 \pm 0.08$ |
| Duality | SS | $0.84 \pm 0.10$ | $0.70 \pm 0.07$ | $0.82 \pm 0.16$ | $0.65 \pm 0.05$ | $0.42 \pm 0.04$ | $0.63 \pm 0.06$ | $1.38 \pm 0.14$ | $0.36 \pm 0.02$ | $0.53 \pm 0.05$ |
| | ST | $1.10 \pm 0.09$ | $1.06 \pm 0.09$ | $0.90 \pm 0.17$ | $0.72 \pm 0.07$ | $0.55 \pm 0.06$ | $0.81 \pm 0.11$ | $2.36 \pm 0.19$ | $0.43 \pm 0.05$ | $0.70 \pm 0.08$ |

To demonstrate **how inter-domain sampling imbalance can cause underrepresented domains to be absorbed by more populous ones**, thereby supressing inter-domain support set and impairing generalization, we design a toy experiment with imbalanced domain sampling but balanced class sampling within each domain. Specifically, we sample 100 examples per class for the major domain, 20 per class for the medium domain, and 4 per class for the minority domain. This setting is compared against a fully balanced baseline in which each domain contributes only 10 samples per class. As illustrated in Fig. 8, several key observations emerge: 1) Under the balanced setting, losses across domains remain similar. In contrast, the imbalanced setting leads to substantial loss discrepancies, with the minority domain exhibiting the highest and most unstable validation loss. This indicates that domain-level sampling imbalance negatively affects the generalization ability of each source domain. 2) Comparing inter-domain discrepancies in SubFig. 8 (I) and (II), we observe that the divergence between majority and minority domains under the imbalanced setting is significantly smaller than in the balanced case, suggesting that the minority domain has been absorbed by the majority. 3) Regarding source-to-target domain shift, the imbalanced setting exhibits notably larger discrepancies compared to the balanced case, confirming that domain-level imbalance undermines cross-domain generalization. 4) Even with the use of a recent distribution alignment method, i.e., TRCI (Salaudeen & Koyejo, 2024), the loss gap between majority and minority domains persists, although the overall inter-domain distributional divergence is noticeably reduced.

### E.3. Impact of Domain-Level Resampling

To ensure a fair comparison, a domain-level resampling strategy is implicitly applied to all long-tailed methods. Specifically, each training iteration consists of samples from all domains. This strategy is commonly adopted in domain generalization, particularly by alignment-based approaches. Tab. 6 reports the impact of applying or omitting this resampling strategy on the BSoftmax method. These results indicate that under the TotalHeavyTail setting, which is dominated by label imbalance, this strategy has minimal effect, suggesting that class discriminability should be prioritized in this case. In contrast, under the Duality setting, where heterogeneous label shift reduces inter-class competition and amplifies domain shift, this resampling strategy helps to mitigate domain divergence during training, thereby improving the robustness of long-tailed methods.

### E.4. Prior Distribution Discrepancies in Learned Embedding

In Tab. 7, we investigate the marginal distribution discrepancies, i.e., prior distribution discrepancies, of learned representations across several representative methods. Despite not explicitly aligning prior distributions, our proposed NDCL achieves lower prior discrepancies both among source domains (SS) and between source and target domains (ST), compared to existing domain alignment approaches such as MMD and RDM. This suggests that our posterior-alignment strategy implicitly mitigates prior mismatches to some extent. Another potential explanation, discussed in Subsection E.2, is that the methods, such as MMD and RDM, may suffer from misalignment due to label shift, particularly under the TotalHeavyTail setting. TCRI achieves lower prior discrepancy than ours, possibly due to its causal approach, which aligns support sets rather than full representations and thus avoids misalignment when sampling is sufficient for each domain. BoDA, which directly manipulates the representations, yields the lowest prior discrepancy overall.

*Table 8.* Comparison of Contrastive Loss Variants for our NDCL. NDCL[InfoNCE-ND] denotes our proposed method in the main manuscript, dominated by negatives with InfoNCE-like objective. NDCL[SupCon-ND] denotes NDCL with SupCon-like objective dominated by negatives, while NDCL[SupCon] denotes NDCL with SupCon objective dominated by positives.

| | TotalHeavyTail | | | | Duality | | | |
| --- | --- | --- | --- | --- | --- | --- | --- | --- |
| | Average | Many | Medium | Few | Average | Many | Medium | Few |
| NDCL[SupCon] | $47.3 \pm 0.2$ | $76.1 \pm 0.7$ | $64.8 \pm 0.9$ | $26.7 \pm 0.4$ | $53.1 \pm 0.3$ | $70.2 \pm 1.4$ | $62.6 \pm 0.5$ | $43.7 \pm 0.7$ |
| NDCL[SupCon-ND] | $48.0 \pm 0.3$ | $75.3 \pm 1.3$ | $65.3 \pm 0.4$ | $27.7 \pm 0.5$ | $53.2 \pm 0.2$ | $69.7 \pm 1.4$ | $63.1 \pm 0.6$ | $43.6 \pm 0.4$ |
| NDCL[InfoNCE-ND] | $49.0 \pm 0.2$ | $71.6 \pm 0.8$ | $66.0 \pm 0.2$ | $30.5 \pm 0.3$ | $55.7 \pm 0.2$ | $71.4 \pm 1.0$ | $65.8 \pm 0.5$ | $47.6 \pm 0.6$ |

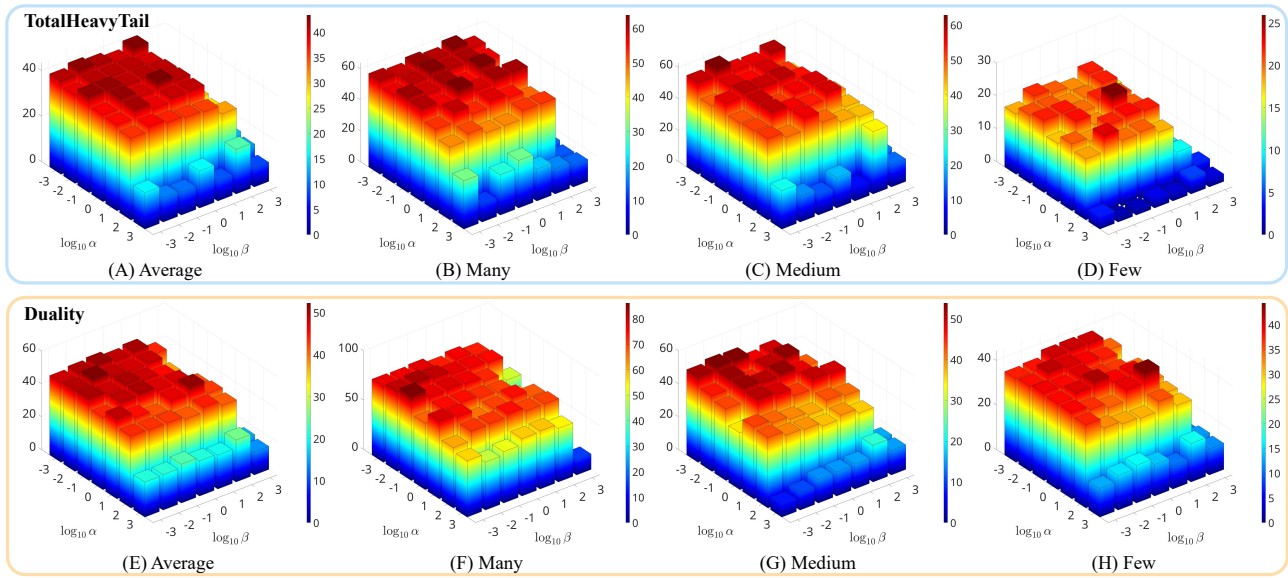

*Figure 9.* Joint influence of hyperparameters $\alpha$ and $\beta$ on OfficeHome under two different settings. The upper part corresponds to the TotalHeavyTail setting, and the lower part to the Duality setting. Log-scale axes are used, e.g., $0 = \log_{10} 1$.

### E.5. Empirical Evidence of Proposed Contrastive Objective

The results in Tab. 8 demonstrate the performance differences among three contrastive loss variants integrated into our NDCL framework. Both NDCL[SupCon-ND] and NDCL[InfoNCE-ND] show clear improvements over NDCL[SupCon], particularly on minority classes, confirming that negative-dominated separation alleviates the absorption of small clusters into majority ones. Among them, NDCL[InfoNCE-ND], which is our proposed method in the main manuscript, achieves the best balance, yielding the highest overall accuracy while **substantially boosting *Few* class performance** across both datasets. These findings highlight the importance of posterior-aligned negative separation in maintaining stable and generalizable decision boundaries under imbalanced domain shifts, which is consistent with our gradient analysis.

### E.6. Parameter Studies

Fig. 9 reports the joint influence of the two hyperparameters involved in our NDCL. These results indicate that both of them are effective within the range of $10^{[-3,2]}$, with optimal average performance typically around $10^{-1}$ and $10^{-2}$, respectively. Notably, under the severely imbalanced TotalHeavyTail setting, larger values lead to improved performance on minority classes, indicating that our design can effectively enhance class separability for underrepresented categories.

Fig. 10 reports the influence of Beta Distribution parameter $\rho$ involved in our NDCL. Note that $\rho$ is the Beta distribution parameter, not the mixing coefficient $\lambda$ itself. When $\rho < 1$, the sampled mixing coefficients are biased toward the extremes; when $\rho > 1$, most mixed samples lie near the midpoint of the two inputs. From this table, we observe that under the TotalHeavyTail setting, performance is relatively insensitive to $\rho$, with the best range around $\{0.2, \ldots, 2.0\}$. In contrast, under the Duality setting, the effective range narrows to $\{0.1, \ldots, 0.5\}$. This difference arises because in TotalHeavyTail, the class imbalance is more severe, causing minority samples to cluster near majority ones; thus, mixtures closer to the middle region are more beneficial. In Duality, however, preserving the semantic structure of the original samples is more critical, favoring more extreme mixing.

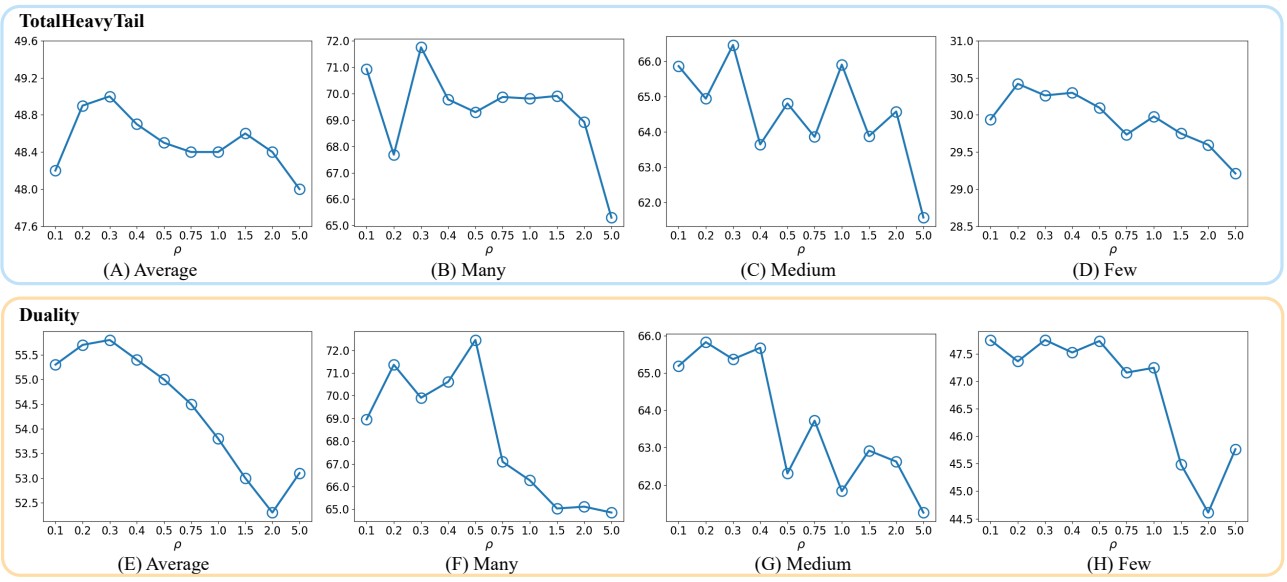

*Figure 10.* The influence of Beta Distribution parameter $\rho$ on OfficeHome under two different settings. The horizontal axis denotes different values of $\rho$, and the vertical axis denotes accuracy.

*Table 9.* Average per-batch training cost (in seconds).

|  | RDM | PGrad | TRCI | BSoftMax | GINIDG | BoDA | SAMALTDG | NDCL | NDCL–NoCon | NDCL–NoAug | NDCL–NoConst |
|---|---|---|---|---|---|---|---|---|---|---|---|
| VLCS | 0.224 | 0.679 | 0.433 | 0.248 | 0.617 | 0.315 | 0.489 | 0.500 | 0.229 | 0.263 | 0.483 |
| PACS | 0.208 | 0.597 | 0.415 | 0.219 | 0.606 | 0.267 | 0.478 | 0.468 | 0.215 | 0.260 | 0.453 |
| OfficeHome | 0.228 | 0.654 | 0.466 | 0.254 | 0.613 | 0.298 | 0.493 | 0.505 | 0.221 | 0.296 | 0.494 |

### E.7. Computational Complexity Analysis

**First**, we would like to clarify that the number of hyper-parameters involved in our NDCL is comparable to those in other IDG methods. NDCL has 3, GINIDG has 1 but involves 3 sub-objective functions, SAMALTDG has 4, BoDA has 7. **Second**, in Tab. 9, we have empirically investigated the computational complexity of our NDCL. From this table, we can observe that: 1) Overall, NDCL exhibits moderate training cost, significantly lower than PGrad with gradient manipulation and GINIDG with adversarial generation. 2) Compared to the IDG methods, NDCL achieves comparable efficiency to SAMALTDG, and although its cost is marginally higher than BoDA, NDCL consistently outperforms these methods across all cases. We believe this constitutes a reasonable trade-off between efficiency and effectiveness. 3) Compared with NDCL and NDCL-NoConst, the cost of predictive center alignment is relatively minor. 4) Comparing NDCL-NoCon and NDCL-NoAug, the cost of hard negative mining is also relatively modest, and both variants are among the more efficient in terms of runtime. These findings indicate that the primary computational cost of NDCL arises from training on augmented data.

According to Tab. 4 in the main manuscript, NDCL-NoAug, which does not employ hard negative mining, exhibits a training cost comparable to that of BoDA, while still achieving consistently better performance.

### E.8. Stability of Prototypes under Few-Sample Regimes

The quality of prediction prototypes $\left\{\boldsymbol{\mu}_k^d \mid k \in \{1, \ldots, K\}, d \in \{1, \ldots, N\}\right\}$ naturally affects model performance. However, our NDCL remains robust even under extreme data scarcity, as evidenced in Tab. 10.

Under the TotalHeavyTail setting on OfficeHome, approximately 40% of the classes contain fewer than two samples per source domain. Despite this severe limitation for accurate prototype estimation, NDCL still achieves competitive performance on minority classes, as shown in Tabs. 2, 3, and Fig. 4. In contrast, under the Duality setting, where more samples are available and prototype quality improves, NDCL further achieves SOTA performance on minority classes.

To directly assess stability, Tab. 10 tracks the step-wise behavior of extreme minority classes (with fewer than two samples)

*Table 10.* The step-wise behavior of extreme minority classes, whose header denotes the step ($\times 100$).

| | 0 | 3 | 6 | 9 | 12 | 15 | 18 | 21 | 24 | 27 | 30 | 33 | 36 | 39 | 42 | 45 | 48 | 50 |
|---|---|---|---|---|---|---|---|---|---|---|---|---|---|---|---|---|---|---|
| Sim | - | 0.14 | 0.93 | 0.96 | 0.99 | 0.99 | 0.99 | 0.99 | 0.99 | 0.98 | 0.99 | 0.98 | 0.98 | 0.99 | 0.98 | 0.99 | 0.99 | 0.99 |
| Loss | 0.62 | 0.39 | 0.39 | 0.39 | 0.38 | 0.39 | 0.38 | 0.38 | 0.39 | 0.39 | 0.38 | 0.38 | 0.38 | 0.39 | 0.38 | 0.38 | 0.39 | 0.39 |

*Table 11.* Quantitative Evaluation of Margin and Posterior Discrepancy on OfficeHome under the TotalHeavyTail and Duality settings. Avg $\gamma(h)$ denotes the average per-instance margin, and $\Pr[\gamma(h) \leq \delta]$ denotes the proportion of samples whose margin is at most $\delta$ where $\delta = 0$. PD denotes the posterior discrepancy measured using the Jensen–Shannon divergence. ($\uparrow$) means higher is better, and ($\downarrow$) means lower is better.

| | TotalHeavyTail | | | Duality | | |
|---|---|---|---|---|---|---|
| | Avg $\gamma(h)$ ($\uparrow$) | $\Pr[\gamma(h) \leq \delta]$ ($\downarrow$) | PD ($\downarrow$) | Avg $\gamma(h)$ ($\uparrow$) | $\Pr[\gamma(h) \leq \delta]$ ($\downarrow$) | PD ($\downarrow$) |
| PGrad | 0.046 | 0.540 | 0.300 | 0.222 | 0.427 | 0.208 |
| TCRI | 0.008 | 0.545 | 0.291 | 0.111 | 0.489 | 0.234 |
| BSoftmax | 0.086 | 0.514 | 0.272 | 0.171 | 0.463 | 0.221 |
| GINIDG | $-0.084$ | 0.564 | 0.309 | 0.092 | 0.499 | 0.246 |
| BoDA | 0.047 | 0.568 | 0.351 | 0.214 | 0.459 | 0.274 |
| NDCL | 0.136 | 0.485 | 0.254 | 0.230 | 0.432 | 0.189 |

*Table 12.* Pearson correlation between theoretical quantities and target accuracy using $\alpha = 0.05$.

| | Avg $\gamma(h)$ | $\Pr[\gamma(h) \leq \delta]$ | PD |
|---|---|---|---|
| Correlation | 0.869 | $-0.934$ | $-0.889$ |
| $p$-value | $2.439 \times 10^{-4}$ | $8.616 \times 10^{-6}$ | $1.099 \times 10^{-4}$ |

under the TotalHeavyTail setting on OfficeHome. The prototype similarity (Sim) between consecutive steps rapidly stabilizes, while the corresponding $\mathcal{L}_{con}$ (Loss) exhibits smooth convergence. These results demonstrate that our prototypes do not suffer from high-variance fluctuations and remain statistically reliable even in low-data regimes.

### E.9. Quantitative Evaluation of Margin and Posterior Discrepancy

Fig. 4 has visualized per-class margins and posterior discrepancies on the target domain of OfficeHome, where NDCL achieves noticeably better separation and posterior alignment than competing methods, especially on the Few classes.

To further quantify how margin and posterior discrepancy relate to generalization performance, we report the corresponding metrics with $\delta = 0$ for six representative methods in Tab. 11, and examine their Pearson correlations with target-domain accuracy in Tab. 12. These quantitative results reveal two key findings.

1. *Margin terms strongly correlate with accuracy.* Across both settings, NDCL achieves the largest average margin and the smallest proportion of small-margin samples, while competing methods exhibit lower margins and higher small-margin probability, consistent with their weaker target-domain accuracy. Pearson correlation further confirms a strong positive association between average margin and accuracy, and a strong negative association for small-margin probability, both statistically significant. This supports the relevance of the margin term in our bound.

2. *Posterior discrepancy negatively correlates with accuracy.* NDCL consistently yields the lowest posterior discrepancy, whereas other baselines, including alignment-based competitors, suffer larger discrepancies. The negative correlation with accuracy is statistically significant, validating the role of the posterior discrepancy term in our theoretical formulation.

In summary, the results provide clear empirical support for the theory: larger margins and smaller posterior discrepancies reliably predict higher target-domain accuracy, confirming the practical relevance of both terms in our bound.

### E.10. Significance Test

To complement performance metrics, we additionally report the Friedman test (Miller Jr, 1997) with post-hoc analysis, which statistically ranks all 24 methods with $\alpha = 0.05$. The results are visualized in Fig. 11, from which several observations

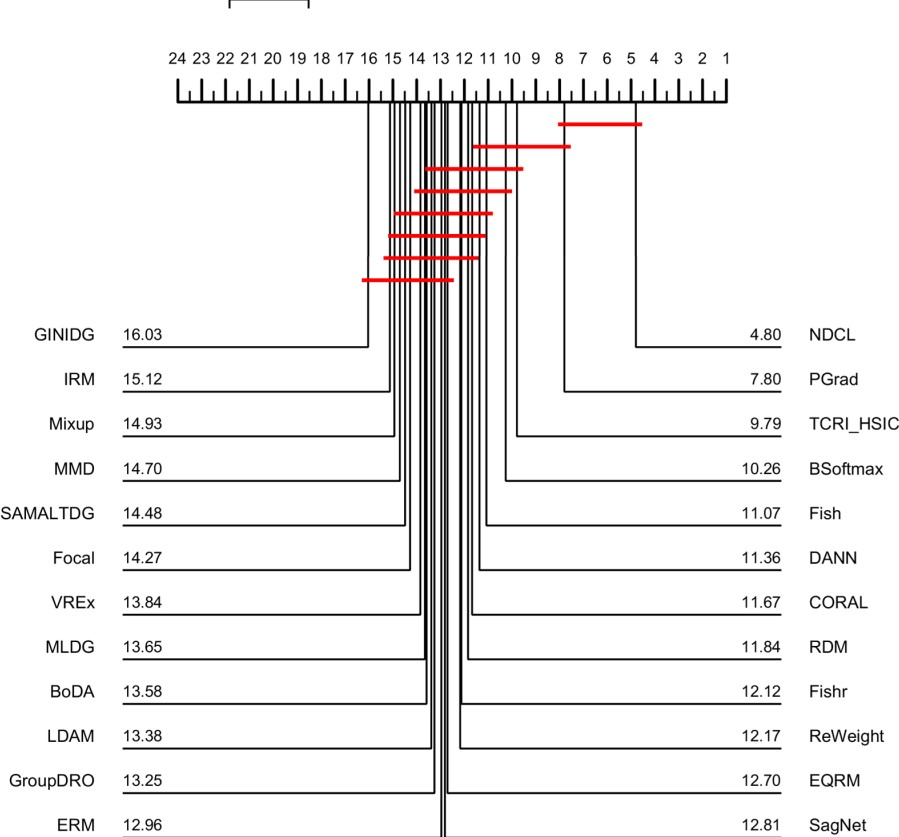

*Figure 11.* Nonparametric hypothesis testing using the Friedman test on the performance ranks of all compared methods across cases. Here, CD denotes the critical difference, which is 3.71.

emerge: 1) *NDCL achieves the best overall ranking by a clear margin.* NDCL obtains an average rank of 4.80, markedly better than all 23 baselines. The gap between NDCL and the second-best method (PGrad, 7.80) is substantial, highlighting the consistently strong performance of NDCL across diverse domains and class-imbalance conditions. 2) *Most existing DG methods form a middle tier with similar and noticeably worse ranks.* A large cluster of methods exhibit average ranks between 11 and 15, indicating that while they offer moderate improvements, their performance is far from competitive with the top-tier methods. 3) *In conjunction with CD = 3.71, NDCL is statistically superior to almost all methods.* Only PGrad falls within the non-significant region; all remaining 22 methods are significantly worse than NDCL. **Overall**, these findings provide strong and comprehensive evidence for the effectiveness and robustness of our NDCL across diverse settings.

### E.11. Overall Results for Large-scale Benchmark

In this subsection, we further evaluate the proposed NDCL on the large-scale benchmark DomainNet (Peng et al., 2019) under the TotalHeavyTail and Duality settings, as reported in Tab. 13. We additionally include one very recent work, i.e., PgCL (Wang et al., 2026) for comparison.

From these results, we observe that: 1) our NDCL consistently achieves the highest average accuracy across both settings, demonstrating a clear overall advantage; 2) on minority classes, it delivers top-tier performance, ranking first under the Duality setting and second under the TotalHeavyTail setting, slightly behind BSoftmax. These results collectively demonstrate that our method remains effective on large-scale datasets.

### E.12. Comprehensive Results

In this subsection, we report more comprehensive results of our designed experiments.

*Table 13.* Overall accuracy of representative methods under two different setting on DomainNet.

| | TotalHeavyTail | | | | Duality | | | |
|---|---|---|---|---|---|---|---|---|
| | Average | Many | Medium | Few | Average | Many | Medium | Few |
| ERM | $24.7 \pm 0.6$ | $57.2 \pm 0.7$ | $50.2 \pm 0.3$ | $15.0 \pm 0.1$ | $24.6 \pm 0.2$ | $48.5 \pm 0.6$ | $41.1 \pm 0.5$ | $19.3 \pm 0.1$ |
| TCRI_HSIC | $25.4 \pm 0.5$ | $59.0 \pm 0.9$ | $51.9 \pm 0.6$ | $15.2 \pm 0.4$ | $25.9 \pm 0.2$ | $49.5 \pm 0.7$ | $41.8 \pm 0.7$ | $20.7 \pm 0.3$ |
| BSoftmax | $26.4 \pm 0.5$ | $45.3 \pm 0.4$ | $45.6 \pm 0.2$ | $19.4 \pm 0.2$ | $26.2 \pm 0.6$ | $41.8 \pm 1.3$ | $37.3 \pm 0.7$ | $22.7 \pm 0.5$ |
| BoDA | $25.9 \pm 0.4$ | $59.7 \pm 0.5$ | $52.8 \pm 0.3$ | $15.6 \pm 0.1$ | $26.6 \pm 0.2$ | $53.8 \pm 0.5$ | $44.7 \pm 0.2$ | $21.0 \pm 0.2$ |
| PgCL | $25.7 \pm 0.2$ | $59.6 \pm 0.1$ | $53.9 \pm 0.2$ | $15.5 \pm 0.5$ | $26.5 \pm 0.3$ | $47.7 \pm 0.2$ | $38.1 \pm 0.4$ | $22.1 \pm 0.3$ |
| NDCL | $27.9 \pm 0.3$ | $54.4 \pm 0.5$ | $52.2 \pm 0.4$ | $18.6 \pm 0.2$ | $27.3 \pm 0.2$ | $48.6 \pm 0.2$ | $40.4 \pm 0.8$ | $23.0 \pm 0.1$ |

*Table 14.* Worst-domain *in-distribution* accuracy per target under the Duality setting on three benchmarks. The **bold**, underline, and dashline items are the best, the second-best, and the third-best results, respectively. Column V on VLCS denotes the in-distribution accuracy of the worst source domain when V is used as the target domain.

| | VLCS | | | | PACS | | | | OfficeHome | | | |
|---|---|---|---|---|---|---|---|---|---|---|---|---|
| | C | L | S | V | A | C | P | S | A | C | P | R |
| ERM | $68.1 \pm 0.4$ | $62.9 \pm 1.7$ | $62.0 \pm 2.9$ | $33.7 \pm 2.5$ | $77.8 \pm 0.6$ | $79.6 \pm 0.5$ | $76.6 \pm 1.5$ | $75.8 \pm 0.7$ | $67.0 \pm 0.3$ | $66.5 \pm 0.6$ | $49.4 \pm 0.3$ | $55.9 \pm 0.8$ |
| IRM | $67.1 \pm 4.6$ | $64.1 \pm 1.1$ | $60.3 \pm 0.9$ | $31.3 \pm 2.8$ | $75.1 \pm 1.2$ | $75.1 \pm 2.9$ | $75.8 \pm 2.0$ | $74.9 \pm 1.9$ | $64.9 \pm 0.6$ | $65.5 \pm 0.7$ | $46.9 \pm 0.9$ | $54.4 \pm 0.9$ |
| GroupDRO | $67.6 \pm 1.5$ | $64.7 \pm 2.1$ | $60.8 \pm 0.4$ | $41.7 \pm 5.9$ | $75.7 \pm 0.2$ | $78.4 \pm 0.9$ | $78.8 \pm 1.0$ | $76.9 \pm 1.7$ | $66.1 \pm 0.8$ | $66.7 \pm 0.9$ | $48.5 \pm 1.1$ | $56.9 \pm 0.6$ |
| Mixup | $71.1 \pm 2.5$ | $64.4 \pm 2.5$ | $57.8 \pm 1.6$ | $36.7 \pm 1.8$ | $76.2 \pm 1.1$ | $77.3 \pm 1.8$ | $78.5 \pm 0.7$ | $76.2 \pm 1.2$ | $67.0 \pm 1.3$ | $67.1 \pm 0.2$ | $48.5 \pm 0.2$ | $59.5 \pm 0.2$ |
| MLDG | $70.2 \pm 2.8$ | $63.9 \pm 2.0$ | $60.3 \pm 1.5$ | $35.7 \pm 1.8$ | $76.9 \pm 1.2$ | $77.9 \pm 1.2$ | $76.3 \pm 1.6$ | $75.3 \pm 2.4$ | $66.9 \pm 0.7$ | $67.7 \pm 0.4$ | $48.3 \pm 1.0$ | $56.5 \pm 0.8$ |
| CORAL | $69.8 \pm 2.1$ | $65.3 \pm 0.6$ | $63.7 \pm 0.2$ | $38.4 \pm 0.8$ | $76.1 \pm 1.1$ | $75.5 \pm 0.9$ | $74.8 \pm 1.3$ | $73.6 \pm 0.7$ | $68.4 \pm 0.8$ | $69.2 \pm 0.9$ | $51.8 \pm 0.4$ | $59.7 \pm 0.5$ |
| MMD | $63.3 \pm 2.8$ | $64.5 \pm 2.6$ | $59.9 \pm 2.1$ | $31.4 \pm 0.7$ | $79.7 \pm 1.4$ | $79.9 \pm 0.7$ | $76.9 \pm 2.2$ | $74.9 \pm 1.1$ | $66.5 \pm 1.1$ | $66.6 \pm 0.1$ | $48.4 \pm 0.4$ | $53.7 \pm 0.5$ |
| DANN | $71.9 \pm 2.7$ | $62.3 \pm 3.0$ | $61.1 \pm 3.5$ | $41.0 \pm 5.1$ | $78.4 \pm 0.3$ | $80.5 \pm 0.6$ | $75.4 \pm 3.1$ | $74.9 \pm 1.2$ | $66.8 \pm 0.5$ | $66.9 \pm 0.8$ | $49.1 \pm 0.5$ | $55.8 \pm 0.7$ |
| SagNet | $71.7 \pm 0.3$ | $66.4 \pm 0.8$ | $64.8 \pm 1.9$ | $32.0 \pm 1.6$ | $77.6 \pm 1.0$ | $78.0 \pm 0.6$ | $76.7 \pm 2.3$ | $75.3 \pm 2.3$ | $68.6 \pm 0.1$ | $69.5 \pm 0.3$ | $52.3 \pm 0.7$ | $59.4 \pm 0.8$ |
| VREx | $65.7 \pm 3.9$ | $63.8 \pm 0.6$ | $64.9 \pm 0.9$ | $32.0 \pm 2.5$ | $74.6 \pm 2.1$ | $77.2 \pm 0.8$ | $75.4 \pm 1.1$ | $78.2 \pm 1.4$ | $66.1 \pm 0.7$ | $66.4 \pm 0.5$ | $49.7 \pm 0.8$ | $57.8 \pm 0.4$ |
| Fish | $72.6 \pm 2.3$ | $66.1 \pm 2.4$ | $62.0 \pm 1.4$ | $34.5 \pm 2.3$ | $76.1 \pm 1.2$ | $77.6 \pm 0.7$ | $77.3 \pm 1.8$ | $77.6 \pm 0.8$ | $67.6 \pm 0.3$ | $68.8 \pm 0.2$ | $50.2 \pm 0.8$ | $59.7 \pm 1.2$ |
| Fishr | $68.8 \pm 2.3$ | $66.3 \pm 2.5$ | $64.0 \pm 1.5$ | $36.8 \pm 1.8$ | $77.3 \pm 0.7$ | $77.8 \pm 1.1$ | $76.4 \pm 1.6$ | $76.8 \pm 1.1$ | $67.6 \pm 0.7$ | $66.4 \pm 0.2$ | $49.9 \pm 0.6$ | $58.4 \pm 0.5$ |
| EQRM | $69.7 \pm 3.8$ | $61.0 \pm 1.5$ | $62.6 \pm 1.1$ | $34.2 \pm 1.9$ | $79.0 \pm 1.3$ | $80.9 \pm 0.4$ | $76.9 \pm 1.3$ | $77.3 \pm 1.4$ | $66.1 \pm 1.1$ | $67.1 \pm 0.1$ | $50.0 \pm 0.4$ | $56.7 \pm 0.3$ |
| RDM | $72.5 \pm 1.8$ | $65.1 \pm 1.8$ | $59.5 \pm 0.4$ | $35.5 \pm 2.4$ | $76.9 \pm 0.2$ | $78.6 \pm 0.1$ | $79.4 \pm 1.2$ | $75.6 \pm 1.6$ | $67.8 \pm 0.7$ | $67.6 \pm 1.2$ | $49.3 \pm 0.9$ | $57.2 \pm 0.6$ |
| PGrad | $74.8 \pm 1.4$ | $65.5 \pm 2.3$ | $63.5 \pm 0.9$ | $33.1 \pm 1.5$ | $79.8 \pm 0.5$ | $81.1 \pm 0.5$ | $80.4 \pm 1.7$ | $78.3 \pm 2.5$ | $69.2 \pm 0.4$ | $69.9 \pm 0.5$ | $52.1 \pm 0.9$ | $62.0 \pm 0.8$ |
| TCRI | $65.9 \pm 3.0$ | $67.0 \pm 1.6$ | $64.7 \pm 3.1$ | $31.5 \pm 1.5$ | $76.2 \pm 1.1$ | $77.8 \pm 1.1$ | $77.5 \pm 0.9$ | $79.0 \pm 1.0$ | $69.2 \pm 0.4$ | $69.5 \pm 0.5$ | $50.1 \pm 1.2$ | $59.9 \pm 0.5$ |
| Focal | $68.0 \pm 0.7$ | $63.9 \pm 2.2$ | $61.1 \pm 1.3$ | $32.5 \pm 0.3$ | $79.6 \pm 0.6$ | $75.6 \pm 0.6$ | $77.4 \pm 1.4$ | $77.0 \pm 0.4$ | $66.7 \pm 0.4$ | $66.6 \pm 0.4$ | $48.1 \pm 0.8$ | $56.0 \pm 0.2$ |
| ReWeight | $64.7 \pm 1.1$ | $64.4 \pm 0.9$ | $63.3 \pm 1.2$ | $33.0 \pm 0.9$ | $79.2 \pm 0.5$ | $79.4 \pm 0.3$ | $78.2 \pm 0.5$ | $76.0 \pm 1.8$ | $68.2 \pm 0.7$ | $67.6 \pm 0.3$ | $48.8 \pm 0.5$ | $57.1 \pm 0.8$ |
| BSoftmax | $66.8 \pm 0.9$ | $67.3 \pm 1.9$ | $61.4 \pm 1.3$ | $38.7 \pm 2.4$ | $80.2 \pm 0.7$ | $78.6 \pm 1.7$ | $77.4 \pm 1.9$ | $77.0 \pm 0.6$ | $68.0 \pm 0.6$ | $67.4 \pm 0.5$ | $49.2 \pm 0.8$ | $57.6 \pm 0.5$ |
| LDAM | $68.9 \pm 1.1$ | $63.8 \pm 1.3$ | $60.5 \pm 1.1$ | $37.3 \pm 1.0$ | $78.7 \pm 1.5$ | $78.6 \pm 2.1$ | $78.5 \pm 0.2$ | $74.9 \pm 0.7$ | $65.8 \pm 0.4$ | $66.5 \pm 0.3$ | $47.0 \pm 0.5$ | $55.5 \pm 1.0$ |
| BoDA | $71.6 \pm 3.0$ | $65.1 \pm 2.8$ | $64.3 \pm 1.6$ | $31.4 \pm 2.4$ | $74.9 \pm 0.3$ | $76.2 \pm 0.9$ | $76.3 \pm 0.5$ | $74.2 \pm 2.2$ | $68.2 \pm 0.9$ | $68.5 \pm 0.6$ | $51.8 \pm 1.1$ | $58.8 \pm 0.2$ |
| GINIDG | $72.9 \pm 1.8$ | $62.6 \pm 0.4$ | $63.0 \pm 3.4$ | $31.7 \pm 3.2$ | $69.2 \pm 3.2$ | $73.7 \pm 1.1$ | $73.9 \pm 1.6$ | $72.6 \pm 1.7$ | $65.2 \pm 1.3$ | $65.8 \pm 0.8$ | $49.5 \pm 1.5$ | $54.1 \pm 0.7$ |
| SAMALTDG | $70.7 \pm 0.7$ | $63.0 \pm 2.6$ | $63.4 \pm 1.7$ | $34.3 \pm 0.8$ | $79.9 \pm 0.3$ | $79.3 \pm 0.3$ | $77.8 \pm 1.3$ | $75.7 \pm 1.6$ | $66.3 \pm 0.8$ | $66.9 \pm 0.2$ | $49.2 \pm 0.5$ | $58.0 \pm 0.6$ |
| NDCL | $74.9 \pm 0.5$ | $68.1 \pm 1.2$ | $62.9 \pm 1.5$ | $41.0 \pm 1.8$ | $80.0 \pm 0.4$ | $81.9 \pm 0.2$ | $81.2 \pm 1.7$ | $78.3 \pm 1.3$ | $69.4 \pm 0.7$ | $69.8 \pm 0.3$ | $51.3 \pm 0.5$ | $60.0 \pm 0.4$ |

Tab. 14 reports the worst-domain in-distribution accuracy under the Duality setting, that is, the performance on the minority source domain in each experiment. These results highlight several key findings: 1) Our proposed NDCL consistently improves performance on the worst-case domain, demonstrating its effectiveness in mitigating the absorption of minority domains by majority ones, as discussed in Subsection E.2. 2) Domain generalization methods generally perform well, underscoring domain shift as the primary bottleneck in this setting. 3) Some other methods achieve competitive results sporadically, suggesting that enhancing class discriminability alone can offer limited gains, though less reliably than addressing domain shift directly under this setting for IDG.

Tab. 15 and 16 report detailed results on VLCS and PACS benchmarks under the GINIDG setting, which are the performance on the target domains. Tab. 17, 18, and 19 report detailed results on three benchmarks under the TotalHeavyTail setting. Tab. 20, 21, and 22 report detailed results on three benchmarks under the Duality setting. From the detailed tables, we observe some variability in performance across different domain combinations and settings. Nevertheless, our NDCL consistently ranks among the top three in most cases. *Notably, when there is a large discrepancy between source and target domains, such as when domain S is selected as the target domain on PACS, our NDCL achieves consistently strong results across all settings.* This demonstrates the effectiveness and robustness of our NDCL under various forms of label shift.

*Table 15.* Detailed results of the target domain on VLCS benchmark under the GINIDG setting.

| | C | | | | L | | | | S | | | | V | | | |
|---|---|---|---|---|---|---|---|---|---|---|---|---|---|---|---|---|
| | Average | Many | Median | Few | Average | Many | Median | Few | Average | Many | Median | Few | Average | Many | Median | Few |
| ERM | 97.4 ± 0.7 | 98.5 ± 0.9 | 87.4 ± 0.7 | 96.7 ± 0.7 | 62.4 ± 0.4 | 64.3 ± 0.6 | 25.9 ± 0.8 | 50.0 ± 1.5 | 70.1 ± 0.6 | 65.2 ± 1.7 | 72.8 ± 5.7 | 23.3 ± 4.1 | 67.9 ± 4.4 | 66.6 ± 3.1 | 78.0 ± 3.8 | 64.6 ± 8.7 |
| IRM | 97.2 ± 0.6 | 99.4 ± 0.2 | 88.1 ± 2.5 | 91.5 ± 2.1 | 64.5 ± 0.5 | 67.0 ± 0.6 | 21.7 ± 3.1 | 45.0 ± 2.1 | 73.0 ± 0.6 | 70.0 ± 2.3 | 65.2 ± 0.4 | 26.8 ± 5.1 | 71.7 ± 0.4 | 70.9 ± 1.6 | 72.6 ± 6.0 | 69.1 ± 7.9 |
| GroupDRO | 97.5 ± 0.4 | 99.3 ± 0.3 | 89.9 ± 4.2 | 95.9 ± 0.4 | 60.1 ± 0.9 | 65.3 ± 1.2 | 25.3 ± 2.5 | 48.8 ± 0.5 | 67.5 ± 1.8 | 60.4 ± 3.4 | 74.0 ± 2.2 | 26.9 ± 3.2 | 74.9 ± 1.8 | 75.5 ± 1.2 | 71.6 ± 1.7 | 71.6 ± 5.0 |
| Mixup | 95.6 ± 1.5 | 99.7 ± 0.1 | 60.1 ± 19.2 | 96.4 ± 1.6 | 63.2 ± 1.1 | 62.2 ± 1.1 | 18.4 ± 3.5 | 47.5 ± 2.1 | 68.8 ± 1.8 | 64.7 ± 3.6 | 69.5 ± 3.4 | 23.0 ± 3.5 | 72.7 ± 0.7 | 76.1 ± 0.9 | 70.7 ± 2.2 | 55.9 ± 4.3 |
| MLDG | 97.5 ± 0.5 | 99.4 ± 0.1 | 83.5 ± 7.1 | 97.5 ± 0.9 | 65.4 ± 1.3 | 67.6 ± 1.6 | 26.1 ± 1.8 | 47.8 ± 1.4 | 68.5 ± 1.0 | 62.3 ± 1.5 | 66.7 ± 3.4 | 31.1 ± 8.2 | 71.3 ± 1.2 | 68.3 ± 2.0 | 74.5 ± 3.1 | 73.6 ± 6.2 |
| CORAL | 96.7 ± 0.5 | 98.4 ± 0.2 | 88.4 ± 4.1 | 96.3 ± 0.8 | 63.3 ± 0.7 | 65.5 ± 0.5 | 25.0 ± 0.7 | 47.0 ± 0.8 | 68.6 ± 0.9 | 67.7 ± 1.6 | 62.0 ± 0.7 | 30.7 ± 3.3 | 74.3 ± 0.8 | 75.2 ± 0.5 | 76.9 ± 2.8 | 69.2 ± 2.3 |
| MMD | 97.5 ± 0.3 | 98.7 ± 0.5 | 87.3 ± 2.8 | 95.5 ± 1.3 | 63.3 ± 1.3 | 65.6 ± 1.3 | 24.7 ± 2.2 | 45.3 ± 1.9 | 68.3 ± 1.2 | 64.6 ± 2.7 | 67.7 ± 1.7 | 22.0 ± 4.5 | 75.8 ± 1.8 | 77.1 ± 0.6 | 68.3 ± 1.4 | 67.3 ± 8.8 |
| DANN | 97.7 ± 0.1 | 99.5 ± 0.2 | 94.6 ± 1.8 | 95.9 ± 0.8 | 63.4 ± 0.4 | 65.5 ± 0.6 | 22.2 ± 1.8 | 51.1 ± 0.8 | 71.0 ± 0.1 | 66.5 ± 1.4 | 66.8 ± 3.6 | 33.3 ± 0.6 | 74.5 ± 1.3 | 72.0 ± 2.3 | 76.2 ± 3.5 | 70.8 ± 5.4 |
| SagNet | 95.3 ± 0.2 | 98.6 ± 0.4 | 80.5 ± 3.8 | 90.2 ± 3.1 | 65.3 ± 1.6 | 67.7 ± 1.6 | 21.5 ± 0.9 | 48.1 ± 0.3 | 69.9 ± 1.1 | 66.9 ± 2.8 | 61.5 ± 4.9 | 32.1 ± 3.0 | 72.8 ± 0.7 | 74.5 ± 0.7 | 72.0 ± 0.9 | 65.6 ± 3.8 |
| VREx | 93.8 ± 2.1 | 96.2 ± 1.6 | 86.6 ± 3.4 | 95.6 ± 1.5 | 63.2 ± 0.4 | 65.3 ± 0.2 | 24.1 ± 3.4 | 50.5 ± 0.4 | 66.5 ± 0.5 | 61.6 ± 2.3 | 68.7 ± 5.5 | 26.7 ± 7.2 | 71.3 ± 0.4 | 72.1 ± 4.4 | 73.8 ± 6.7 | 72.1 ± 4.4 |
| Fish | 97.8 ± 0.2 | 98.8 ± 0.7 | 90.6 ± 2.4 | 96.3 ± 0.6 | 64.3 ± 1.2 | 66.4 ± 1.3 | 23.0 ± 0.9 | 51.1 ± 0.6 | 71.9 ± 0.4 | 65.6 ± 3.0 | 64.2 ± 3.5 | 32.0 ± 1.7 | 75.3 ± 2.5 | 78.0 ± 1.3 | 73.5 ± 1.5 | 66.3 ± 6.2 |
| Fishr | 97.8 ± 0.5 | 99.6 ± 0.3 | 95.7 ± 0.6 | 93.1 ± 2.0 | 62.9 ± 0.5 | 64.9 ± 0.4 | 24.9 ± 4.3 | 49.8 ± 1.3 | 70.4 ± 0.9 | 65.6 ± 3.0 | 68.6 ± 3.1 | 34.6 ± 1.6 | 73.7 ± 1.0 | 71.9 ± 3.1 | 76.6 ± 3.4 | 77.1 ± 3.0 |
| EQRM | 96.9 ± 0.7 | 98.8 ± 0.6 | 88.3 ± 4.3 | 95.8 ± 0.7 | 60.9 ± 0.9 | 62.8 ± 0.8 | 28.4 ± 1.0 | 48.0 ± 0.7 | 70.2 ± 0.5 | 68.8 ± 1.7 | 60.2 ± 7.4 | 24.4 ± 2.2 | 70.2 ± 1.5 | 69.4 ± 3.5 | 81.1 ± 2.4 | 63.4 ± 4.2 |
| RDM | 96.6 ± 0.8 | 99.7 ± 0.2 | 74.6 ± 7.1 | 93.7 ± 2.4 | 62.9 ± 0.7 | 65.2 ± 0.8 | 20.8 ± 2.6 | 47.0 ± 0.7 | 68.4 ± 0.6 | 62.6 ± 1.0 | 73.3 ± 3.2 | 25.6 ± 4.3 | 72.1 ± 0.8 | 71.2 ± 1.3 | 76.4 ± 0.8 | 64.2 ± 2.8 |
| PGrad | 98.7 ± 0.1 | 99.6 ± 0.2 | 95.3 ± 1.8 | 97.6 ± 0.8 | 63.8 ± 1.0 | 66.0 ± 0.9 | 23.5 ± 0.5 | 49.5 ± 1.2 | 71.9 ± 0.5 | 59.5 ± 1.5 | 64.3 ± 3.6 | 26.8 ± 2.8 | 75.8 ± 0.5 | 78.1 ± 0.6 | 70.8 ± 2.2 | 67.4 ± 1.6 |
| TCRI | 97.0 ± 0.9 | 95.6 ± 3.5 | 92.0 ± 2.2 | 92.9 ± 2.8 | 63.0 ± 1.7 | 65.1 ± 2.0 | 24.5 ± 3.7 | 48.3 ± 1.4 | 67.4 ± 1.2 | 59.5 ± 1.5 | 67.2 ± 3.1 | 24.4 ± 4.0 | 75.2 ± 1.5 | 76.8 ± 3.2 | 75.9 ± 3.4 | 69.3 ± 1.3 |
| Focal | 97.5 ± 0.4 | 99.2 ± 0.4 | 86.2 ± 3.7 | 95.4 ± 1.5 | 63.1 ± 0.9 | 65.3 ± 0.9 | 22.1 ± 1.7 | 48.3 ± 0.4 | 69.0 ± 0.8 | 57.5 ± 3.7 | 64.7 ± 7.6 | 21.7 ± 3.4 | 71.7 ± 3.2 | 72.4 ± 3.6 | 67.5 ± 1.4 | 64.1 ± 5.2 |
| ReWeight | 95.7 ± 0.4 | 97.8 ± 0.4 | 87.9 ± 6.1 | 92.1 ± 2.6 | 63.0 ± 0.6 | 64.9 ± 0.5 | 25.2 ± 1.7 | 50.8 ± 0.7 | 66.0 ± 1.8 | 57.7 ± 3.1 | 75.4 ± 1.5 | 33.3 ± 2.5 | 73.1 ± 2.4 | 70.3 ± 4.2 | 63.9 ± 4.6 | 75.1 ± 4.9 |
| BSoftmax | 97.4 ± 0.8 | 99.1 ± 0.4 | 81.5 ± 12.2 | 97.1 ± 0.8 | 62.5 ± 1.3 | 64.2 ± 1.1 | 26.2 ± 1.5 | 55.7 ± 1.7 | 65.0 ± 1.6 | 54.4 ± 3.7 | 81.7 ± 3.4 | 43.9 ± 5.4 | 67.0 ± 2.1 | 61.9 ± 2.6 | 75.5 ± 2.0 | 78.1 ± 1.2 |
| LDAM | 96.8 ± 1.4 | 99.1 ± 0.5 | 71.5 ± 10.0 | 97.3 ± 0.6 | 59.6 ± 1.1 | 61.2 ± 1.0 | 26.2 ± 1.1 | 48.6 ± 0.8 | 68.4 ± 1.6 | 62.3 ± 3.1 | 68.4 ± 4.9 | 26.7 ± 4.5 | 73.4 ± 0.8 | 71.7 ± 2.7 | 74.1 ± 3.7 | 71.9 ± 2.9 |
| GINIDG | 96.4 ± 0.6 | 97.3 ± 0.8 | 82.8 ± 6.0 | 95.3 ± 1.5 | 65.2 ± 0.3 | 67.4 ± 0.4 | 22.7 ± 2.7 | 50.3 ± 0.9 | 66.0 ± 1.3 | 60.9 ± 3.1 | 65.0 ± 4.1 | 25.7 ± 4.2 | 69.6 ± 0.7 | 69.3 ± 2.1 | 72.2 ± 3.6 | 63.4 ± 3.7 |
| BoDA | 96.9 ± 1.0 | 98.8 ± 0.1 | 80.8 ± 8.0 | 97.0 ± 0.6 | 64.6 ± 1.0 | 66.8 ± 0.9 | 29.5 ± 1.2 | 46.9 ± 1.4 | 68.6 ± 0.8 | 66.9 ± 1.6 | 62.2 ± 3.7 | 33.0 ± 7.1 | 75.2 ± 0.7 | 75.0 ± 0.7 | 76.8 ± 1.8 | 74.3 ± 2.9 |
| SAMALTDG | 98.3 ± 0.3 | 99.6 ± 0.1 | 89.4 ± 2.4 | 97.2 ± 1.0 | 60.4 ± 1.9 | 62.6 ± 2.1 | 18.0 ± 2.5 | 49.7 ± 1.9 | 69.3 ± 1.1 | 62.0 ± 1.9 | 73.7 ± 5.4 | 33.6 ± 4.4 | 69.8 ± 1.8 | 66.0 ± 2.8 | 84.6 ± 1.9 | 76.8 ± 3.5 |
| NDCL | 99.1 ± 0.4 | 100.0 ± 0.0 | 94.7 ± 2.9 | 98.3 ± 0.7 | 64.8 ± 0.7 | 66.9 ± 1.0 | 28.0 ± 1.9 | 48.9 ± 0.8 | 72.2 ± 0.4 | 67.6 ± 1.4 | 65.9 ± 3.1 | 25.4 ± 5.0 | 76.1 ± 0.1 | 77.8 ± 0.9 | 65.1 ± 2.5 | 61.8 ± 1.2 |

*Table 16.* Detailed results of the target domain on PACS benchmark under the GINIDG setting.

| | A | | | | C | | | | P | | | | S | | | |
|---|---|---|---|---|---|---|---|---|---|---|---|---|---|---|---|---|
| | Average | Many | Median | Few | Average | Many | Median | Few | Average | Many | Median | Few | Average | Many | Median | Few |
| ERM | 82.5 ± 1.0 | 89.0 ± 0.3 | 86.4 ± 2.0 | 62.1 ± 3.0 | 73.5 ± 1.0 | 78.6 ± 1.3 | 83.0 ± 0.3 | 54.7 ± 3.8 | 96.0 ± 0.1 | 95.9 ± 0.5 | 92.8 ± 0.4 | 99.4 ± 0.5 | 76.8 ± 0.7 | 72.0 ± 4.9 | 82.5 ± 1.3 | 69.6 ± 5.4 |
| IRM | 82.0 ± 1.8 | 88.0 ± 1.0 | 81.8 ± 4.2 | 67.6 ± 0.3 | 71.7 ± 2.8 | 76.4 ± 2.7 | 80.1 ± 1.1 | 57.7 ± 9.5 | 96.3 ± 0.3 | 95.6 ± 1.0 | 94.0 ± 0.9 | 99.5 ± 0.4 | 74.7 ± 2.4 | 74.8 ± 4.3 | 78.9 ± 2.4 | 54.9 ± 6.4 |
| GroupDRO | 80.8 ± 1.1 | 86.6 ± 1.8 | 87.0 ± 1.6 | 60.0 ± 6.2 | 73.2 ± 1.4 | 79.8 ± 1.0 | 82.6 ± 1.6 | 48.0 ± 2.9 | 95.3 ± 0.6 | 93.6 ± 0.9 | 92.9 ± 1.6 | 100.0 ± 0.0 | 69.3 ± 1.7 | 61.9 ± 6.0 | 76.4 ± 5.7 | 64.7 ± 6.2 |
| Mixup | 82.4 ± 0.8 | 89.3 ± 0.3 | 88.2 ± 0.8 | 57.7 ± 3.7 | 78.5 ± 0.7 | 84.5 ± 1.6 | 85.5 ± 1.1 | 57.8 ± 7.1 | 95.8 ± 1.0 | 95.0 ± 1.0 | 93.5 ± 1.3 | 99.3 ± 0.6 | 71.9 ± 1.2 | 74.1 ± 1.5 | 75.3 ± 2.0 | 42.3 ± 17.1 |
| MLDG | 83.9 ± 1.1 | 89.0 ± 0.2 | 86.3 ± 1.1 | 66.1 ± 3.0 | 75.8 ± 1.4 | 82.4 ± 0.8 | 82.9 ± 1.6 | 55.3 ± 2.4 | 95.3 ± 0.3 | 93.7 ± 0.7 | 92.5 ± 1.5 | 100.0 ± 0.0 | 73.1 ± 1.8 | 72.9 ± 3.5 | 77.2 ± 0.7 | 47.6 ± 5.6 |
| CORAL | 83.6 ± 1.8 | 89.5 ± 0.6 | 86.6 ± 0.8 | 66.2 ± 7.2 | 75.8 ± 1.4 | 82.4 ± 0.8 | 81.5 ± 1.1 | 53.6 ± 2.1 | 96.6 ± 0.4 | 95.6 ± 0.4 | 94.5 ± 1.0 | 100.0 ± 0.0 | 75.4 ± 1.7 | 69.1 ± 4.8 | 82.0 ± 3.2 | 70.4 ± 4.6 |
| MMD | 82.8 ± 1.3 | 87.8 ± 1.0 | 88.9 ± 1.2 | 61.5 ± 3.7 | 73.6 ± 2.6 | 79.6 ± 2.0 | 82.6 ± 0.5 | 51.8 ± 7.3 | 95.8 ± 0.5 | 95.9 ± 0.8 | 91.9 ± 0.5 | 100.0 ± 0.0 | 76.6 ± 0.5 | 69.3 ± 1.5 | 83.7 ± 0.5 | 69.7 ± 4.3 |
| DANN | 84.6 ± 1.5 | 88.3 ± 1.4 | 88.1 ± 0.4 | 71.1 ± 4.5 | 76.4 ± 0.7 | 80.8 ± 1.3 | 81.0 ± 0.3 | 67.8 ± 1.6 | 96.5 ± 0.6 | 95.5 ± 0.8 | 94.5 ± 1.5 | 100.0 ± 0.0 | 73.7 ± 1.0 | 66.4 ± 3.4 | 80.5 ± 1.6 | 65.6 ± 6.8 |
| SagNet | 79.3 ± 2.3 | 84.5 ± 1.4 | 84.7 ± 0.6 | 58.0 ± 8.1 | 75.9 ± 0.3 | 84.5 ± 0.9 | 82.7 ± 1.2 | 67.8 ± 2.7 | 95.1 ± 0.3 | 93.4 ± 0.6 | 92.5 ± 1.4 | 100.0 ± 0.0 | 74.7 ± 3.4 | 64.2 ± 10.1 | 79.7 ± 1.5 | 67.5 ± 5.5 |
| VREx | 85.4 ± 0.5 | 89.5 ± 1.2 | 88.6 ± 0.8 | 71.0 ± 5.1 | 75.9 ± 0.1 | 82.8 ± 0.6 | 83.6 ± 1.2 | 52.1 ± 2.1 | 95.2 ± 0.4 | 94.4 ± 0.4 | 93.0 ± 0.5 | 98.3 ± 1.2 | 74.8 ± 1.8 | 72.7 ± 3.9 | 79.7 ± 1.5 | 59.4 ± 5.2 |
| Fish | 83.1 ± 1.0 | 89.2 ± 1.5 | 85.9 ± 1.9 | 65.1 ± 4.8 | 72.2 ± 3.2 | 77.9 ± 3.8 | 82.6 ± 2.3 | 49.5 ± 1.8 | 97.4 ± 0.3 | 96.8 ± 0.4 | 95.5 ± 0.5 | 100.0 ± 0.0 | 69.9 ± 1.9 | 61.3 ± 4.1 | 79.6 ± 1.1 | 50.9 ± 7.0 |
| Fishr | 84.1 ± 1.4 | 89.7 ± 0.4 | 89.2 ± 1.1 | 64.0 ± 6.0 | 78.1 ± 0.5 | 81.7 ± 1.1 | 85.4 ± 0.9 | 65.0 ± 8.0 | 96.4 ± 0.3 | 96.3 ± 0.9 | 93.2 ± 0.2 | 100.0 ± 0.0 | 70.0 ± 1.8 | 56.2 ± 7.5 | 81.9 ± 1.8 | 61.0 ± 3.4 |
| EQRM | 82.6 ± 1.6 | 87.3 ± 1.1 | 86.3 ± 1.1 | 68.1 ± 7.0 | 77.3 ± 0.8 | 81.0 ± 1.0 | 85.0 ± 1.2 | 63.1 ± 3.5 | 97.0 ± 0.1 | 96.8 ± 0.4 | 94.5 ± 0.6 | 99.9 ± 0.1 | 72.5 ± 2.8 | 64.0 ± 5.2 | 80.5 ± 1.4 | 68.4 ± 7.0 |
| RDM | 82.3 ± 2.0 | 90.2 ± 0.9 | 82.1 ± 1.7 | 64.9 ± 6.4 | 78.0 ± 0.4 | 81.7 ± 1.9 | 87.0 ± 1.3 | 59.9 ± 3.9 | 96.1 ± 0.4 | 95.7 ± 0.9 | 93.0 ± 0.1 | 100.0 ± 0.0 | 76.5 ± 2.4 | 70.2 ± 5.4 | 83.6 ± 0.9 | 68.4 ± 8.3 |
| PGrad | 87.8 ± 0.3 | 93.9 ± 0.4 | 89.1 ± 0.9 | 72.5 ± 0.8 | 76.5 ± 0.9 | 83.5 ± 0.7 | 83.4 ± 1.0 | 54.5 ± 5.2 | 96.9 ± 0.3 | 97.1 ± 0.2 | 94.2 ± 0.8 | 99.7 ± 0.2 | 76.0 ± 0.9 | 76.0 ± 1.7 | 79.8 ± 1.1 | 58.8 ± 7.5 |
| TCRI | 84.5 ± 0.7 | 88.5 ± 1.3 | 88.6 ± 0.9 | 69.6 ± 6.4 | 76.8 ± 0.9 | 81.8 ± 0.3 | 86.5 ± 1.4 | 54.4 ± 3.4 | 96.7 ± 0.6 | 95.6 ± 1.2 | 95.0 ± 1.2 | 100.0 ± 0.0 | 75.0 ± 1.7 | 63.3 ± 5.2 | 86.6 ± 1.2 | 54.5 ± 5.8 |
| Focal | 82.4 ± 0.6 | 89.9 ± 1.0 | 85.5 ± 1.4 | 60.7 ± 2.1 | 77.6 ± 1.0 | 83.1 ± 1.1 | 84.4 ± 1.4 | 60.4 ± 1.0 | 95.8 ± 0.8 | 95.0 ± 1.2 | 92.8 ± 1.2 | 100.0 ± 0.0 | 68.4 ± 1.5 | 64.7 ± 7.0 | 73.8 ± 3.7 | 52.3 ± 0.9 |
| ReWeight | 83.9 ± 1.9 | 90.1 ± 0.9 | 89.9 ± 0.7 | 61.2 ± 8.0 | 76.4 ± 0.4 | 82.6 ± 0.5 | 85.4 ± 0.9 | 53.0 ± 1.7 | 96.3 ± 0.5 | 96.1 ± 0.5 | 93.4 ± 1.1 | 99.6 ± 0.3 | 71.8 ± 2.2 | 62.0 ± 2.1 | 80.6 ± 5.1 | 67.1 ± 3.8 |
| BSoftmax | 84.0 ± 1.1 | 87.9 ± 2.2 | 87.0 ± 1.5 | 70.6 ± 3.3 | 77.3 ± 1.8 | 77.9 ± 3.8 | 84.5 ± 0.9 | 73.2 ± 2.5 | 96.0 ± 0.5 | 94.3 ± 1.1 | 94.2 ± 0.4 | 100.0 ± 0.0 | 75.7 ± 0.9 | 63.3 ± 2.7 | 84.7 ± 0.3 | 83.3 ± 0.6 |
| LDAM | 85.6 ± 1.4 | 90.4 ± 0.6 | 90.1 ± 0.2 | 68.8 ± 5.1 | 78.3 ± 1.3 | 83.9 ± 0.8 | 85.2 ± 1.0 | 58.3 ± 0.5 | 96.0 ± 0.5 | 94.9 ± 0.5 | 93.6 ± 0.9 | 99.9 ± 0.1 | 72.0 ± 0.4 | 67.8 ± 0.8 | 76.5 ± 2.4 | 70.0 ± 8.2 |
| GINIDG | 75.3 ± 1.2 | 80.0 ± 4.9 | 81.2 ± 3.5 | 54.9 ± 7.3 | 76.2 ± 1.0 | 83.6 ± 1.4 | 82.1 ± 1.8 | 54.7 ± 3.3 | 96.0 ± 0.6 | 93.7 ± 0.9 | 94.8 ± 0.8 | 99.9 ± 0.1 | 76.9 ± 3.0 | 69.9 ± 4.9 | 84.0 ± 1.2 | 70.2 ± 6.5 |
| BoDA | 83.8 ± 1.3 | 90.2 ± 0.2 | 84.4 ± 1.5 | 67.7 ± 3.7 | 76.3 ± 0.5 | 82.1 ± 1.3 | 82.0 ± 0.9 | 49.7 ± 0.6 | 95.5 ± 0.7 | 93.2 ± 1.2 | 93.9 ± 1.1 | 100.0 ± 0.0 | 74.7 ± 1.7 | 76.8 ± 1.2 | 75.9 ± 1.6 | 66.0 ± 10.5 |
| SAMALTDG | 81.3 ± 2.1 | 87.9 ± 0.6 | 87.4 ± 1.8 | 59.1 ± 8.3 | 77.3 ± 0.7 | 83.7 ± 0.9 | 83.7 ± 1.4 | 58.3 ± 2.9 | 96.7 ± 0.5 | 94.9 ± 0.3 | 95.6 ± 1.0 | 100.0 ± 0.0 | 74.2 ± 2.5 | 71.5 ± 1.2 | 78.0 ± 4.0 | 68.9 ± 4.0 |
| NDCL | 87.7 ± 0.6 | 90.3 ± 0.6 | 89.0 ± 1.1 | 80.4 ± 4.8 | 78.7 ± 0.5 | 84.4 ± 0.4 | 85.2 ± 0.6 | 60.4 ± 2.6 | 97.3 ± 0.3 | 96.6 ± 0.3 | 95.5 ± 1.2 | 100.0 ± 0.0 | 79.3 ± 0.4 | 74.8 ± 1.8 | 83.8 ± 0.9 | 76.9 ± 7.0 |

Table 17. Detailed results of the target domain on VLCS benchmark under the TotalHeavyTail setting.

| | C | | | | L | | | | S | | | | V | | | |
|---|---|---|---|---|---|---|---|---|---|---|---|---|---|---|---|---|
| | Average | Many | Median | Few | Average | Many | Median | Few | Average | Many | Median | Few | Average | Many | Median | Few |
| ERM | 91.0 ± 2.1 | 99.8 ± 0.1 | 78.2 ± 4.7 | 81.5 ± 6.6 | 66.3 ± 0.3 | 45.5 ± 1.8 | 54.7 ± 0.9 | 37.5 ± 0.8 | 66.8 ± 0.8 | 81.8 ± 3.1 | 49.7 ± 5.4 | 36.8 ± 1.7 | 70.4 ± 0.9 | 89.8 ± 1.5 | 61.5 ± 2.1 | 47.5 ± 3.1 |
| IRM | 89.3 ± 0.9 | 98.7 ± 1.1 | 78.9 ± 2.8 | 68.7 ± 10.1 | 65.1 ± 0.2 | 42.3 ± 1.7 | 53.3 ± 2.0 | 32.9 ± 4.5 | 60.8 ± 1.3 | 91.7 ± 0.5 | 38.6 ± 2.0 | 18.2 ± 2.8 | 68.5 ± 2.5 | 95.3 ± 1.1 | 51.4 ± 4.0 | 43.1 ± 7.7 |
| GroupDRO | 92.7 ± 1.7 | 98.7 ± 0.9 | 84.5 ± 6.0 | 77.5 ± 5.9 | 65.3 ± 0.4 | 38.0 ± 2.0 | 57.6 ± 0.9 | 37.4 ± 1.5 | 65.8 ± 1.8 | 87.1 ± 2.7 | 49.9 ± 9.8 | 33.0 ± 1.3 | 69.6 ± 0.3 | 93.9 ± 0.8 | 57.6 ± 1.1 | 41.7 ± 1.3 |
| Mixup | 87.7 ± 1.2 | 99.8 ± 0.1 | 70.9 ± 6.2 | 70.1 ± 3.8 | 65.3 ± 0.4 | 40.9 ± 1.2 | 58.2 ± 2.1 | 37.7 ± 1.2 | 62.2 ± 2.0 | 91.4 ± 1.6 | 39.2 ± 3.3 | 23.9 ± 3.0 | 68.6 ± 1.5 | 94.7 ± 1.1 | 57.7 ± 3.1 | 35.7 ± 4.6 |
| MLDG | 92.2 ± 1.0 | 99.9 ± 0.1 | 78.8 ± 8.7 | 82.3 ± 5.7 | 65.2 ± 0.7 | 40.3 ± 1.4 | 58.3 ± 0.9 | 36.9 ± 3.6 | 65.5 ± 0.4 | 86.9 ± 1.6 | 44.1 ± 3.4 | 28.9 ± 1.2 | 69.9 ± 0.6 | 88.4 ± 3.2 | 54.5 ± 3.7 | 57.3 ± 3.6 |
| CORAL | 92.5 ± 0.4 | 99.9 ± 0.1 | 86.0 ± 2.3 | 76.4 ± 0.6 | 67.2 ± 0.8 | 46.6 ± 1.6 | 54.7 ± 0.7 | 36.9 ± 1.2 | 63.9 ± 1.1 | 90.1 ± 1.7 | 41.4 ± 3.0 | 31.6 ± 2.1 | 68.9 ± 0.5 | 94.4 ± 1.6 | 55.3 ± 2.7 | 40.0 ± 1.5 |
| MMD | 91.6 ± 1.2 | 99.5 ± 0.1 | 82.6 ± 4.4 | 78.5 ± 6.1 | 64.2 ± 1.8 | 40.0 ± 5.5 | 55.3 ± 1.2 | 36.4 ± 1.3 | 64.1 ± 0.1 | 91.7 ± 0.5 | 41.4 ± 0.7 | 28.1 ± 1.6 | 68.0 ± 1.2 | 93.8 ± 1.4 | 55.0 ± 2.7 | 37.3 ± 2.9 |
| DANN | 91.7 ± 2.6 | 98.9 ± 0.9 | 83.0 ± 5.5 | 82.4 ± 4.5 | 62.3 ± 1.9 | 31.8 ± 4.5 | 58.2 ± 1.4 | 41.3 ± 2.0 | 64.6 ± 2.7 | 84.7 ± 1.0 | 49.7 ± 3.0 | 29.9 ± 1.4 | 70.8 ± 0.6 | 88.1 ± 2.0 | 65.5 ± 1.1 | 44.7 ± 0.3 |
| SagNet | 95.1 ± 1.5 | 99.9 ± 0.1 | 86.7 ± 7.7 | 85.5 ± 7.3 | 64.4 ± 0.5 | 38.3 ± 1.4 | 57.0 ± 0.7 | 37.0 ± 2.1 | 62.7 ± 1.2 | 88.7 ± 2.3 | 42.2 ± 6.1 | 27.3 ± 3.5 | 68.5 ± 1.6 | 92.3 ± 3.8 | 61.2 ± 4.6 | 32.9 ± 7.7 |
| VREx | 91.4 ± 1.1 | 96.8 ± 2.3 | 85.2 ± 4.6 | 80.6 ± 3.6 | 67.4 ± 1.0 | 47.4 ± 3.7 | 57.0 ± 1.9 | 34.8 ± 2.5 | 65.4 ± 0.1 | 84.8 ± 2.5 | 49.1 ± 5.7 | 29.1 ± 3.4 | 66.3 ± 1.8 | 91.6 ± 3.7 | 54.4 ± 3.7 | 36.4 ± 4.4 |
| Fish | 94.8 ± 0.6 | 100.0 ± 0.0 | 81.7 ± 4.9 | 90.3 ± 3.1 | 65.6 ± 0.5 | 42.6 ± 2.0 | 55.0 ± 1.7 | 36.2 ± 1.8 | 65.6 ± 1.9 | 91.3 ± 1.6 | 39.5 ± 5.0 | 28.0 ± 2.6 | 71.4 ± 0.8 | 92.4 ± 1.5 | 60.9 ± 1.7 | 46.8 ± 4.3 |
| Fishr | 92.0 ± 1.1 | 100.0 ± 0.0 | 77.0 ± 3.8 | 84.1 ± 3.1 | 65.7 ± 0.6 | 44.9 ± 0.1 | 55.1 ± 1.9 | 29.5 ± 0.9 | 63.7 ± 1.4 | 83.2 ± 1.8 | 49.3 ± 2.7 | 31.5 ± 2.7 | 73.4 ± 0.9 | 94.0 ± 0.6 | 60.9 ± 1.3 | 52.3 ± 2.5 |
| EQRM | 92.1 ± 0.6 | 99.9 ± 0.1 | 74.3 ± 2.0 | 80.5 ± 5.2 | 64.6 ± 0.3 | 39.2 ± 1.3 | 57.4 ± 1.1 | 38.0 ± 0.9 | 60.6 ± 2.3 | 83.4 ± 4.2 | 46.2 ± 7.0 | 27.3 ± 4.6 | 67.9 ± 2.0 | 89.3 ± 2.1 | 57.1 ± 6.2 | 44.1 ± 5.5 |
| RDM | 91.7 ± 0.4 | 100.0 ± 0.0 | 82.6 ± 3.5 | 73.0 ± 6.5 | 64.0 ± 1.8 | 38.2 ± 4.9 | 57.2 ± 0.6 | 35.8 ± 0.9 | 60.1 ± 0.5 | 88.0 ± 2.0 | 39.3 ± 6.3 | 24.0 ± 1.8 | 70.9 ± 2.0 | 92.5 ± 0.9 | 59.9 ± 1.9 | 45.7 ± 7.8 |
| PGrad | 92.5 ± 0.8 | 100.0 ± 0.0 | 83.8 ± 3.3 | 81.5 ± 3.7 | 65.6 ± 0.3 | 42.2 ± 0.7 | 56.8 ± 0.3 | 35.6 ± 0.5 | 61.4 ± 1.9 | 94.0 ± 0.6 | 36.0 ± 2.4 | 22.0 ± 1.5 | 72.7 ± 0.8 | 95.5 ± 0.8 | 57.9 ± 1.9 | 48.8 ± 1.2 |
| TCRI | 91.0 ± 0.8 | 98.1 ± 1.5 | 79.3 ± 5.6 | 83.7 ± 1.5 | 63.5 ± 0.9 | 36.0 ± 2.2 | 58.6 ± 3.1 | 40.8 ± 0.5 | 61.4 ± 1.9 | 90.4 ± 2.7 | 35.0 ± 2.5 | 25.5 ± 1.3 | 71.3 ± 1.4 | 87.7 ± 5.4 | 61.7 ± 3.4 | 54.3 ± 1.0 |
| Focal | 89.5 ± 1.3 | 98.7 ± 1.1 | 82.3 ± 3.0 | 76.6 ± 3.8 | 65.5 ± 0.6 | 43.2 ± 1.2 | 53.7 ± 0.4 | 37.2 ± 3.1 | 62.2 ± 3.4 | 88.6 ± 3.2 | 41.1 ± 4.8 | 26.7 ± 4.1 | 70.5 ± 1.2 | 96.0 ± 0.7 | 53.9 ± 3.4 | 45.3 ± 2.7 |
| ReWeight | 94.8 ± 1.1 | 99.9 ± 0.1 | 80.3 ± 7.8 | 93.5 ± 0.7 | 58.5 ± 0.9 | 23.6 ± 2.2 | 60.6 ± 1.5 | 45.7 ± 3.3 | 63.4 ± 2.9 | 75.5 ± 7.2 | 49.8 ± 5.7 | 39.0 ± 2.8 | 75.3 ± 2.0 | 89.5 ± 0.1 | 67.4 ± 1.6 | 58.5 ± 7.5 |
| BSoftmax | 94.8 ± 0.1 | 99.0 ± 0.0 | 76.2 ± 0.9 | 95.0 ± 1.3 | 65.9 ± 0.7 | 42.2 ± 2.1 | 57.9 ± 0.4 | 42.1 ± 1.7 | 65.4 ± 2.1 | 72.0 ± 5.5 | 56.8 ± 7.6 | 42.5 ± 5.6 | 72.7 ± 1.8 | 84.3 ± 3.8 | 60.5 ± 1.8 | 68.8 ± 4.1 |
| LDAM | 93.0 ± 1.4 | 99.8 ± 0.2 | 77.7 ± 5.8 | 86.1 ± 2.1 | 65.9 ± 0.8 | 43.6 ± 2.2 | 55.7 ± 2.0 | 41.1 ± 2.6 | 60.8 ± 3.4 | 88.8 ± 3.4 | 42.4 ± 2.8 | 25.7 ± 6.7 | 69.8 ± 2.7 | 89.8 ± 0.8 | 60.9 ± 3.0 | 46.0 ± 7.7 |
| GINIDG | 88.9 ± 1.1 | 100.0 ± 0.0 | 65.1 ± 2.9 | 74.7 ± 8.2 | 66.7 ± 0.5 | 49.0 ± 1.4 | 54.5 ± 0.4 | 29.0 ± 3.9 | 59.1 ± 3.7 | 87.3 ± 1.6 | 42.8 ± 2.1 | 21.3 ± 3.9 | 65.4 ± 1.4 | 85.5 ± 5.5 | 62.6 ± 6.2 | 28.2 ± 4.9 |
| BoDA | 91.3 ± 0.5 | 99.9 ± 0.0 | 79.8 ± 4.8 | 74.1 ± 3.1 | 65.9 ± 1.2 | 44.0 ± 3.5 | 53.6 ± 0.6 | 37.5 ± 1.4 | 59.5 ± 2.5 | 89.4 ± 3.4 | 41.9 ± 3.2 | 21.1 ± 3.2 | 63.8 ± 2.1 | 97.2 ± 0.4 | 44.1 ± 4.1 | 30.2 ± 2.3 |
| SAMALTDG | 88.8 ± 1.2 | 95.2 ± 2.0 | 77.4 ± 3.2 | 86.9 ± 4.4 | 61.8 ± 1.8 | 31.6 ± 3.9 | 56.5 ± 0.4 | 42.2 ± 1.7 | 66.7 ± 2.2 | 73.3 ± 7.5 | 61.0 ± 3.6 | 35.8 ± 2.8 | 73.4 ± 1.7 | 85.4 ± 5.0 | 69.3 ± 6.2 | 56.4 ± 7.1 |
| NDCL | 95.7 ± 0.7 | 100.0 ± 0.0 | 78.9 ± 4.6 | 94.6 ± 0.9 | 65.2 ± 1.0 | 39.9 ± 3.4 | 59.5 ± 2.8 | 41.7 ± 2.5 | 65.3 ± 1.5 | 52.4 ± 4.1 | 58.4 ± 1.7 | 47.9 ± 2.7 | 76.4 ± 0.5 | 90.4 ± 1.6 | 67.6 ± 0.6 | 59.7 ± 3.9 |

Table 18. Detailed results of the target domain on PACS benchmark under the TotalHeavyTail setting.

| | A | | | | C | | | | P | | | | S | | | |
|---|---|---|---|---|---|---|---|---|---|---|---|---|---|---|---|---|
| | Average | Many | Median | Few | Average | Many | Median | Few | Average | Many | Median | Few | Average | Many | Median | Few |
| ERM | 68.6 ± 2.6 | 90.8 ± 1.6 | 59.1 ± 1.8 | 64.8 ± 6.2 | 64.2 ± 2.0 | 93.6 ± 1.0 | 36.1 ± 1.0 | 92.9 ± 0.5 | 92.9 ± 1.5 | 94.7 ± 2.2 | 91.3 ± 0.9 | 88.0 ± 2.5 | 52.6 ± 3.9 | 70.7 ± 0.8 | 61.6 ± 4.4 | 39.7 ± 15.0 |
| IRM | 72.8 ± 1.6 | 91.3 ± 1.4 | 64.9 ± 1.6 | 67.9 ± 4.5 | 67.1 ± 1.0 | 90.4 ± 1.2 | 45.1 ± 1.2 | 89.5 ± 3.7 | 89.5 ± 2.6 | 92.4 ± 1.7 | 84.3 ± 3.6 | 86.3 ± 4.2 | 57.8 ± 4.2 | 74.7 ± 2.9 | 57.5 ± 2.3 | 49.7 ± 8.1 |
| GroupDRO | 75.4 ± 2.2 | 86.1 ± 0.2 | 74.4 ± 1.8 | 65.4 ± 5.2 | 66.9 ± 1.8 | 93.0 ± 0.5 | 44.0 ± 3.0 | 85.6 ± 2.4 | 91.8 ± 1.5 | 96.1 ± 0.4 | 86.6 ± 2.4 | 90.5 ± 1.5 | 54.9 ± 1.5 | 74.2 ± 4.3 | 66.3 ± 8.9 | 38.3 ± 11.1 |
| Mixup | 71.7 ± 0.6 | 90.4 ± 1.1 | 67.7 ± 1.0 | 60.6 ± 1.2 | 63.6 ± 0.6 | 88.3 ± 3.1 | 42.4 ± 2.7 | 85.1 ± 1.3 | 88.7 ± 2.3 | 95.2 ± 2.0 | 79.2 ± 3.1 | 92.7 ± 1.6 | 49.3 ± 2.8 | 72.0 ± 1.7 | 70.5 ± 6.5 | 13.2 ± 3.7 |
| MLDG | 69.8 ± 2.3 | 89.0 ± 1.6 | 66.4 ± 3.3 | 58.6 ± 5.5 | 68.3 ± 1.4 | 92.1 ± 2.2 | 47.7 ± 3.8 | 82.8 ± 4.0 | 92.2 ± 0.8 | 92.2 ± 0.8 | 89.7 ± 2.5 | 87.6 ± 3.3 | 50.3 ± 2.5 | 71.4 ± 2.3 | 70.5 ± 7.2 | 30.3 ± 2.1 |
| CORAL | 65.7 ± 1.8 | 92.3 ± 0.4 | 56.7 ± 4.2 | 52.1 ± 3.8 | 66.7 ± 2.0 | 89.9 ± 2.5 | 45.5 ± 4.8 | 87.6 ± 1.2 | 92.2 ± 1.5 | 92.4 ± 2.2 | 87.6 ± 3.3 | 87.4 ± 3.7 | 55.9 ± 3.2 | 77.8 ± 3.7 | 67.1 ± 7.2 | 22.0 ± 8.2 |
| MMD | 69.5 ± 2.2 | 85.3 ± 1.7 | 64.8 ± 1.6 | 63.8 ± 2.5 | 65.4 ± 1.2 | 93.2 ± 0.9 | 41.2 ± 2.2 | 86.0 ± 0.9 | 90.7 ± 1.5 | 95.3 ± 1.4 | 87.5 ± 0.5 | 87.6 ± 2.0 | 55.5 ± 0.7 | 69.2 ± 5.4 | 75.2 ± 5.2 | 30.6 ± 7.8 |
| DANN | 76.3 ± 2.1 | 92.3 ± 1.0 | 71.9 ± 2.1 | 66.6 ± 3.9 | 69.0 ± 1.3 | 90.9 ± 2.5 | 49.5 ± 3.2 | 85.5 ± 0.9 | 90.7 ± 1.6 | 94.8 ± 2.0 | 83.1 ± 1.3 | 91.2 ± 3.9 | 56.4 ± 1.0 | 67.3 ± 7.2 | 72.3 ± 8.8 | 41.0 ± 6.8 |
| SagNet | 65.1 ± 0.4 | 85.8 ± 2.3 | 60.1 ± 0.8 | 53.6 ± 5.5 | 71.2 ± 1.8 | 87.0 ± 1.2 | 54.6 ± 3.8 | 88.2 ± 1.2 | 93.8 ± 1.4 | 95.7 ± 0.9 | 91.3 ± 2.7 | 91.4 ± 1.8 | 62.5 ± 2.6 | 73.5 ± 3.2 | 71.9 ± 3.0 | 52.4 ± 11.5 |
| VREx | 67.1 ± 1.4 | 85.4 ± 3.8 | 65.1 ± 2.9 | 49.8 ± 5.4 | 63.4 ± 3.1 | 91.0 ± 1.4 | 40.1 ± 2.4 | 88.1 ± 1.2 | 91.4 ± 1.1 | 94.9 ± 1.4 | 86.2 ± 2.5 | 83.7 ± 5.0 | 58.7 ± 2.0 | 75.0 ± 8.3 | 69.7 ± 4.5 | 45.9 ± 11.0 |
| Fish | 74.5 ± 1.5 | 91.7 ± 1.1 | 69.2 ± 3.2 | 67.6 ± 2.5 | 64.5 ± 0.9 | 94.1 ± 1.5 | 35.8 ± 4.5 | 90.5 ± 2.0 | 90.8 ± 1.3 | 94.9 ± 1.4 | 89.2 ± 2.0 | 90.5 ± 1.9 | 51.1 ± 4.3 | 64.2 ± 10.1 | 62.8 ± 6.5 | 34.1 ± 8.6 |
| Fishr | 69.2 ± 0.8 | 90.9 ± 1.0 | 62.2 ± 2.5 | 61.3 ± 3.1 | 64.5 ± 3.0 | 89.1 ± 2.6 | 40.4 ± 2.0 | 89.6 ± 1.4 | 90.5 ± 1.2 | 94.4 ± 0.3 | 84.2 ± 1.1 | 90.5 ± 1.0 | 60.0 ± 1.3 | 73.8 ± 1.5 | 76.7 ± 3.8 | 41.4 ± 10.0 |
| EQRM | 73.7 ± 1.2 | 88.9 ± 1.9 | 68.4 ± 4.0 | 68.8 ± 5.4 | 66.2 ± 1.4 | 91.4 ± 0.9 | 42.7 ± 2.7 | 88.5 ± 1.7 | 92.7 ± 0.5 | 93.7 ± 1.4 | 89.0 ± 0.8 | 87.9 ± 1.4 | 55.9 ± 1.4 | 71.7 ± 3.3 | 72.0 ± 6.0 | 41.7 ± 8.4 |
| RDM | 72.7 ± 1.2 | 90.5 ± 1.0 | 66.6 ± 1.8 | 64.9 ± 3.6 | 64.0 ± 0.4 | 91.5 ± 1.3 | 42.7 ± 2.7 | 89.7 ± 1.3 | 91.8 ± 0.5 | 96.4 ± 0.2 | 88.5 ± 1.9 | 88.5 ± 1.7 | 59.2 ± 3.6 | 77.4 ± 3.2 | 67.7 ± 4.8 | 47.9 ± 4.8 |
| PGrad | 72.4 ± 1.2 | 92.8 ± 0.9 | 66.8 ± 0.6 | 60.9 ± 5.2 | 67.7 ± 1.2 | 93.5 ± 1.9 | 36.6 ± 0.9 | 89.7 ± 1.3 | 92.9 ± 0.4 | 98.1 ± 0.3 | 87.1 ± 1.1 | 90.8 ± 1.8 | 56.1 ± 1.9 | 83.2 ± 1.5 | 66.7 ± 4.4 | 33.3 ± 10.9 |
| TCRI | 71.3 ± 3.6 | 90.5 ± 0.9 | 63.2 ± 4.8 | 63.8 ± 5.8 | 64.7 ± 1.3 | 89.0 ± 0.5 | 47.2 ± 2.3 | 90.8 ± 1.8 | 92.6 ± 0.6 | 95.3 ± 0.2 | 86.1 ± 0.9 | 87.6 ± 2.2 | 62.2 ± 1.6 | 75.8 ± 3.7 | 71.4 ± 3.3 | 45.7 ± 6.3 |
| Focal | 69.3 ± 3.7 | 89.3 ± 1.9 | 65.8 ± 5.1 | 53.4 ± 4.9 | 66.1 ± 1.9 | 92.9 ± 1.2 | 39.3 ± 2.2 | 88.8 ± 0.9 | 92.3 ± 0.6 | 94.5 ± 1.3 | 89.2 ± 1.3 | 88.0 ± 2.5 | 52.7 ± 2.5 | 74.9 ± 7.6 | 64.2 ± 6.1 | 37.1 ± 10.1 |
| ReWeight | 73.8 ± 2.2 | 87.5 ± 3.3 | 71.7 ± 4.5 | 65.1 ± 3.8 | 66.1 ± 1.9 | 94.6 ± 0.5 | 40.1 ± 3.1 | 89.9 ± 0.5 | 92.2 ± 0.6 | 92.3 ± 2.1 | 93.2 ± 1.5 | 95.7 ± 1.4 | 61.9 ± 5.0 | 57.7 ± 8.3 | 76.2 ± 2.9 | 52.1 ± 7.4 |
| BSoftmax | 74.0 ± 0.7 | 88.8 ± 0.5 | 69.7 ± 0.5 | 69.3 ± 4.4 | 68.3 ± 2.9 | 84.7 ± 5.1 | 46.3 ± 7.0 | 95.5 ± 1.3 | 94.8 ± 0.6 | 90.0 ± 3.6 | 91.2 ± 2.3 | 94.8 ± 0.6 | 59.8 ± 1.9 | 48.3 ± 4.7 | 72.4 ± 6.4 | 63.6 ± 5.9 |
| LDAM | 70.6 ± 1.6 | 87.1 ± 0.3 | 65.4 ± 2.0 | 60.9 ± 2.8 | 63.4 ± 2.4 | 90.0 ± 1.0 | 46.3 ± 3.2 | 90.7 ± 3.5 | 93.5 ± 1.0 | 92.8 ± 2.0 | 89.7 ± 1.9 | 94.9 ± 0.5 | 54.9 ± 5.6 | 63.3 ± 5.7 | 60.6 ± 11.5 | 45.8 ± 8.1 |
| GINIDG | 66.8 ± 0.7 | 89.5 ± 0.9 | 57.8 ± 0.8 | 60.3 ± 2.0 | 63.0 ± 1.0 | 91.5 ± 0.6 | 39.2 ± 3.0 | 88.9 ± 2.1 | 91.3 ± 1.7 | 93.6 ± 1.4 | 86.8 ± 3.0 | 89.4 ± 2.8 | 51.6 ± 2.5 | 78.7 ± 5.4 | 62.7 ± 3.6 | 29.7 ± 3.1 |
| BoDA | 69.8 ± 2.8 | 91.0 ± 0.4 | 66.6 ± 3.4 | 50.7 ± 2.7 | 65.3 ± 1.1 | 86.1 ± 1.6 | 45.1 ± 3.0 | 86.0 ± 2.6 | 94.2 ± 2.3 | 94.2 ± 2.3 | 87.4 ± 1.3 | 86.0 ± 2.6 | 54.4 ± 2.3 | 73.0 ± 3.7 | 72.4 ± 2.6 | 32.4 ± 10.0 |
| SAMALTDG | 70.4 ± 1.6 | 89.7 ± 1.1 | 65.0 ± 0.8 | 63.1 ± 5.1 | 66.4 ± 2.2 | 88.3 ± 3.6 | 43.9 ± 5.4 | 92.7 ± 3.2 | 92.9 ± 1.3 | 92.5 ± 2.8 | 91.5 ± 1.6 | 91.1 ± 3.0 | 53.0 ± 1.4 | 58.9 ± 1.4 | 63.5 ± 9.2 | 53.7 ± 10.5 |
| NDCL | 74.6 ± 1.9 | 84.1 ± 1.7 | 74.8 ± 2.3 | 64.9 ± 1.1 | 70.3 ± 1.4 | 86.1 ± 3.5 | 51.9 ± 4.3 | 94.0 ± 0.8 | 93.6 ± 0.7 | 95.7 ± 0.5 | 89.9 ± 1.9 | 91.7 ± 1.0 | 67.2 ± 1.1 | 73.5 ± 5.3 | 72.3 ± 3.5 | 66.4 ± 4.1 |

Table 19. Detailed results of the target domain on OfficeHome benchmark under the TotalHeavyTail setting.

| | A | | | | C | | | | P | | | | R | | | |
|---|---|---|---|---|---|---|---|---|---|---|---|---|---|---|---|---|
| | Average | Many | Median | Few | Average | Many | Median | Few | Average | Many | Median | Few | Average | Many | Median | Few |
| ERM | 41.9 ± 0.3 | 62.5 ± 0.7 | 61.7 ± 1.1 | 20.0 ± 0.9 | 38.5 ± 0.7 | 69.0 ± 1.2 | 58.4 ± 1.2 | 19.9 ± 1.8 | 49.8 ± 0.8 | 84.3 ± 0.9 | 63.8 ± 1.6 | 30.4 ± 0.5 | 54.3 ± 1.4 | 83.9 ± 0.2 | 75.4 ± 1.1 | 31.5 ± 1.8 |
| IRM | 39.3 ± 0.9 | 62.9 ± 1.1 | 58.1 ± 1.9 | 18.6 ± 1.2 | 37.0 ± 0.3 | 63.9 ± 0.7 | 56.6 ± 2.1 | 18.9 ± 0.2 | 46.3 ± 2.0 | 82.9 ± 0.6 | 58.7 ± 4.0 | 27.2 ± 2.5 | 52.7 ± 2.1 | 84.8 ± 0.3 | 71.3 ± 3.1 | 29.6 ± 2.3 |
| GroupDRO | 40.5 ± 0.3 | 61.4 ± 1.1 | 58.9 ± 1.3 | 18.7 ± 0.7 | 39.9 ± 0.4 | 72.3 ± 0.6 | 58.5 ± 1.1 | 21.7 ± 0.9 | 49.5 ± 1.0 | 84.1 ± 0.6 | 64.1 ± 1.6 | 29.7 ± 1.3 | 53.3 ± 0.8 | 84.6 ± 0.7 | 77.6 ± 0.5 | 29.9 ± 0.8 |
| Mixup | 40.6 ± 0.6 | 65.3 ± 1.7 | 58.3 ± 0.5 | 17.9 ± 1.6 | 37.1 ± 1.0 | 65.6 ± 1.8 | 58.5 ± 1.4 | 18.5 ± 1.2 | 48.6 ± 0.7 | 83.9 ± 0.4 | 63.1 ± 1.6 | 28.3 ± 0.9 | 53.3 ± 0.8 | 85.3 ± 0.3 | 77.0 ± 0.5 | 28.0 ± 0.9 |
| MLDG | 40.6 ± 1.2 | 62.2 ± 1.7 | 59.4 ± 0.8 | 18.6 ± 0.9 | 37.6 ± 0.8 | 67.3 ± 2.6 | 55.3 ± 1.8 | 20.7 ± 1.0 | 49.2 ± 0.9 | 81.5 ± 0.7 | 61.5 ± 1.8 | 30.5 ± 1.0 | 54.3 ± 0.9 | 84.1 ± 0.8 | 77.1 ± 1.1 | 29.8 ± 0.7 |
| CORAL | 40.3 ± 0.8 | 64.6 ± 1.5 | 57.5 ± 1.6 | 17.1 ± 0.6 | 39.1 ± 0.7 | 75.8 ± 2.0 | 61.8 ± 0.4 | 17.3 ± 0.9 | 49.3 ± 1.4 | 87.0 ± 0.6 | 61.3 ± 2.4 | 29.2 ± 1.3 | 54.0 ± 0.9 | 86.6 ± 0.5 | 79.0 ± 0.5 | 28.1 ± 1.2 |
| MMD | 38.8 ± 1.3 | 59.2 ± 1.4 | 59.7 ± 1.6 | 17.3 ± 1.1 | 39.7 ± 1.1 | 70.7 ± 1.6 | 59.7 ± 1.0 | 17.6 ± 1.6 | 50.3 ± 0.2 | 83.8 ± 1.3 | 62.2 ± 0.8 | 31.7 ± 0.7 | 52.8 ± 0.9 | 84.7 ± 0.1 | 73.9 ± 1.3 | 28.9 ± 1.1 |
| DANN | 40.1 ± 0.5 | 64.6 ± 2.6 | 58.9 ± 1.6 | 16.6 ± 0.3 | 37.2 ± 1.2 | 68.4 ± 3.1 | 55.8 ± 2.2 | 19.1 ± 1.4 | 47.7 ± 1.2 | 82.3 ± 1.4 | 62.2 ± 1.2 | 26.9 ± 1.5 | 51.7 ± 1.3 | 84.4 ± 0.4 | 75.7 ± 1.0 | 25.8 ± 1.6 |
| SagNet | 42.3 ± 0.2 | 65.2 ± 0.7 | 61.1 ± 1.9 | 19.2 ± 0.5 | 38.9 ± 0.5 | 74.9 ± 0.6 | 60.4 ± 0.2 | 17.9 ± 0.8 | 49.2 ± 1.0 | 84.8 ± 0.2 | 62.9 ± 1.3 | 29.4 ± 1.0 | 54.8 ± 0.7 | 85.5 ± 0.9 | 79.7 ± 1.6 | 29.7 ± 1.0 |
| VREx | 40.5 ± 0.3 | 62.0 ± 0.3 | 58.6 ± 2.5 | 19.7 ± 0.6 | 38.1 ± 1.1 | 68.1 ± 1.2 | 56.8 ± 0.3 | 20.1 ± 2.3 | 49.1 ± 0.2 | 84.9 ± 0.8 | 60.6 ± 0.3 | 29.5 ± 0.5 | 54.4 ± 1.1 | 83.6 ± 0.8 | 76.9 ± 1.6 | 31.3 ± 1.3 |
| Fish | 41.8 ± 0.6 | 64.7 ± 1.0 | 59.7 ± 0.5 | 20.3 ± 0.4 | 38.8 ± 0.3 | 70.2 ± 1.9 | 60.5 ± 1.6 | 18.1 ± 0.7 | 50.2 ± 0.3 | 86.1 ± 1.0 | 63.1 ± 1.0 | 30.1 ± 0.5 | 55.4 ± 0.3 | 86.3 ± 0.2 | 78.9 ± 2.1 | 30.7 ± 0.5 |
| Fishr | 41.5 ± 0.8 | 64.7 ± 0.5 | 60.7 ± 0.4 | 19.3 ± 1.3 | 38.3 ± 1.0 | 71.4 ± 0.6 | 57.3 ± 0.4 | 20.1 ± 0.4 | 50.1 ± 0.9 | 84.2 ± 1.2 | 62.7 ± 0.7 | 31.4 ± 1.2 | 54.7 ± 0.5 | 84.6 ± 0.6 | 78.2 ± 1.2 | 30.3 ± 0.4 |
| EQRM | 40.5 ± 0.3 | 66.4 ± 1.5 | 56.7 ± 0.7 | 18.3 ± 0.5 | 39.0 ± 0.7 | 74.4 ± 0.6 | 58.0 ± 0.4 | 19.7 ± 1.2 | 50.5 ± 0.5 | 84.3 ± 2.1 | 62.2 ± 1.3 | 32.3 ± 1.2 | 55.2 ± 0.5 | 85.8 ± 0.4 | 76.8 ± 0.5 | 31.4 ± 1.0 |
| RDM | 42.1 ± 0.8 | 62.7 ± 1.6 | 59.5 ± 1.5 | 20.7 ± 0.8 | 39.4 ± 0.5 | 69.3 ± 1.8 | 57.3 ± 1.8 | 21.8 ± 1.0 | 51.3 ± 0.4 | 82.8 ± 1.4 | 63.2 ± 1.5 | 32.4 ± 0.8 | 54.9 ± 0.6 | 84.7 ± 0.4 | 76.3 ± 0.9 | 31.6 ± 1.2 |
| PGrad | 42.4 ± 1.0 | 65.4 ± 0.8 | 60.8 ± 1.1 | 19.8 ± 0.6 | 41.5 ± 0.3 | 66.7 ± 1.6 | 61.9 ± 1.5 | 21.5 ± 1.1 | 52.1 ± 1.4 | 85.6 ± 0.7 | 64.4 ± 1.5 | 32.8 ± 0.2 | 55.4 ± 0.7 | 86.4 ± 0.7 | 78.5 ± 1.5 | 31.7 ± 1.5 |
| TCRI | 42.4 ± 0.3 | 60.7 ± 1.3 | 59.3 ± 0.2 | 24.3 ± 0.7 | 39.4 ± 0.5 | 66.7 ± 1.6 | 59.8 ± 0.7 | 21.4 ± 1.0 | 52.7 ± 0.5 | 81.6 ± 0.3 | 62.3 ± 0.8 | 36.7 ± 0.2 | 57.5 ± 0.6 | 84.8 ± 0.3 | 78.5 ± 1.5 | 35.1 ± 0.5 |
| Focal | 40.3 ± 0.9 | 65.4 ± 1.4 | 57.7 ± 1.3 | 17.6 ± 0.5 | 36.6 ± 0.5 | 66.3 ± 2.6 | 56.6 ± 0.3 | 18.6 ± 0.4 | 49.1 ± 0.8 | 83.7 ± 1.9 | 59.0 ± 0.9 | 30.8 ± 1.5 | 54.3 ± 0.7 | 85.4 ± 0.1 | 78.2 ± 0.2 | 29.7 ± 0.8 |
| ReWeight | 41.8 ± 0.8 | 65.6 ± 1.0 | 61.3 ± 1.0 | 18.8 ± 1.2 | 37.8 ± 1.3 | 64.2 ± 3.7 | 54.2 ± 0.7 | 22.4 ± 0.6 | 52.3 ± 0.7 | 83.7 ± 1.3 | 63.1 ± 1.2 | 34.5 ± 0.3 | 56.5 ± 0.3 | 83.6 ± 0.4 | 79.7 ± 0.8 | 33.7 ± 0.6 |
| BSoftmax | 42.2 ± 0.2 | 55.9 ± 0.1 | 56.0 ± 0.8 | 26.5 ± 0.5 | 39.7 ± 1.1 | 59.1 ± 2.2 | 56.2 ± 0.9 | 26.8 ± 1.6 | 53.6 ± 0.6 | 79.2 ± 1.8 | 60.2 ± 0.5 | 39.3 ± 1.3 | 58.8 ± 0.9 | 83.9 ± 0.7 | 75.1 ± 1.1 | 39.4 ± 0.9 |
| LDAM | 41.7 ± 0.8 | 61.2 ± 1.1 | 59.5 ± 0.9 | 21.3 ± 0.1 | 38.6 ± 0.1 | 70.2 ± 1.3 | 57.0 ± 1.2 | 19.7 ± 0.6 | 51.3 ± 0.6 | 82.1 ± 2.1 | 63.9 ± 1.3 | 32.8 ± 1.1 | 53.4 ± 0.5 | 83.1 ± 1.3 | 76.0 ± 0.4 | 30.1 ± 1.0 |
| GINIDG | 39.9 ± 0.2 | 60.3 ± 1.8 | 58.5 ± 1.0 | 19.9 ± 0.5 | 39.0 ± 0.3 | 64.4 ± 1.4 | 56.4 ± 1.2 | 22.2 ± 1.0 | 48.5 ± 1.0 | 81.8 ± 1.2 | 59.0 ± 0.4 | 30.4 ± 1.7 | 52.5 ± 0.9 | 81.9 ± 0.3 | 74.9 ± 2.0 | 29.3 ± 0.4 |
| BoDA | 39.6 ± 0.6 | 62.7 ± 0.7 | 59.3 ± 1.6 | 15.5 ± 0.6 | 39.3 ± 0.5 | 76.2 ± 1.0 | 61.0 ± 1.0 | 17.8 ± 0.9 | 48.0 ± 0.3 | 85.8 ± 1.0 | 61.9 ± 0.5 | 26.9 ± 0.1 | 52.6 ± 0.7 | 85.4 ± 0.2 | 74.4 ± 1.7 | 27.8 ± 1.6 |
| SAMALTDG | 42.4 ± 1.5 | 63.7 ± 1.8 | 59.2 ± 1.3 | 20.5 ± 1.6 | 37.8 ± 0.9 | 63.7 ± 3.4 | 58.5 ± 0.6 | 20.5 ± 0.6 | 49.2 ± 0.9 | 84.6 ± 0.7 | 60.5 ± 0.7 | 30.3 ± 1.2 | 54.8 ± 0.5 | 84.3 ± 0.5 | 77.8 ± 0.5 | 31.2 ± 0.5 |
| NDCL | 43.0 ± 0.0 | 59.2 ± 0.4 | 59.0 ± 1.5 | 22.8 ± 0.6 | 41.0 ± 1.1 | 65.4 ± 4.6 | 60.3 ± 0.3 | 24.9 ± 0.8 | 54.8 ± 0.2 | 78.1 ± 0.4 | 64.3 ± 0.7 | 39.9 ± 0.6 | 57.1 ± 0.4 | 83.7 ± 0.8 | 80.4 ± 0.6 | 34.4 ± 0.5 |

Table 20. Detailed results of the target domain on VLCS benchmark under the Duality setting.

| | C | | | | L | | | | S | | | | V | | | |
|---|---|---|---|---|---|---|---|---|---|---|---|---|---|---|---|---|
| | Average | Many | Median | Few | Average | Many | Median | Few | Average | Many | Median | Few | Average | Many | Median | Few |
| ERM | 74.2 ± 11.2 | 67.2 ± 20.0 | 82.0 ± 4.3 | 81.7 ± 8.5 | 48.6 ± 0.3 | 1.1 ± 1.2 | 62.3 ± 2.0 | 66.8 ± 0.5 | 43.6 ± 1.1 | 11.4 ± 1.1 | 61.5 ± 6.9 | 49.5 ± 0.6 | 53.1 ± 4.0 | 38.3 ± 3.9 | 83.7 ± 0.9 | 63.3 ± 3.6 |
| IRM | 67.5 ± 12.7 | 52.5 ± 21.0 | 88.1 ± 1.7 | 92.8 ± 3.6 | 47.7 ± 1.2 | 3.1 ± 1.4 | 66.6 ± 3.1 | 63.6 ± 2.7 | 42.1 ± 0.2 | 10.6 ± 1.8 | 57.5 ± 3.7 | 47.8 ± 2.4 | 53.2 ± 2.3 | 42.2 ± 2.2 | 81.6 ± 4.1 | 61.4 ± 3.2 |
| GroupDRO | 83.4 ± 4.9 | 81.1 ± 8.1 | 77.5 ± 2.3 | 94.8 ± 0.9 | 48.1 ± 1.1 | 2.1 ± 1.2 | 59.7 ± 1.7 | 69.6 ± 2.9 | 40.6 ± 1.4 | 8.5 ± 1.2 | 67.3 ± 4.2 | 50.3 ± 1.0 | 59.2 ± 5.1 | 53.6 ± 10.8 | 77.6 ± 8.3 | 62.0 ± 2.7 |
| Mixup | 75.4 ± 7.2 | 67.7 ± 13.7 | 78.4 ± 8.7 | 94.0 ± 2.0 | 48.5 ± 1.0 | 2.0 ± 0.9 | 57.3 ± 3.8 | 66.7 ± 2.9 | 41.1 ± 0.5 | 11.6 ± 2.9 | 61.2 ± 10.7 | 50.1 ± 1.8 | 52.5 ± 2.4 | 38.6 ± 5.1 | 80.1 ± 2.3 | 61.4 ± 2.7 |
| MLDG | 65.5 ± 11.9 | 52.1 ± 17.2 | 86.3 ± 7.5 | 88.7 ± 3.9 | 47.2 ± 1.8 | 2.1 ± 1.1 | 68.0 ± 2.8 | 63.0 ± 3.0 | 41.8 ± 0.9 | 11.9 ± 0.8 | 55.8 ± 9.7 | 49.5 ± 1.7 | 50.9 ± 2.1 | 30.6 ± 5.4 | 77.7 ± 2.5 | 67.5 ± 6.7 |
| CORAL | 69.6 ± 8.8 | 62.5 ± 14.3 | 67.2 ± 5.3 | 92.1 ± 3.9 | 50.2 ± 2.0 | 6.3 ± 4.0 | 64.5 ± 1.3 | 65.8 ± 2.9 | 47.0 ± 0.3 | 24.6 ± 2.1 | 61.7 ± 3.7 | 50.5 ± 1.7 | 54.5 ± 1.6 | 52.2 ± 4.3 | 78.8 ± 1.8 | 56.1 ± 3.9 |
| MMD | 64.5 ± 10.2 | 53.6 ± 16.4 | 69.4 ± 2.8 | 89.6 ± 1.8 | 48.2 ± 2.4 | 6.2 ± 4.2 | 58.0 ± 3.5 | 67.2 ± 3.0 | 40.9 ± 0.6 | 8.6 ± 1.0 | 67.2 ± 3.4 | 46.8 ± 1.2 | 48.4 ± 2.0 | 38.1 ± 2.7 | 81.4 ± 2.1 | 53.6 ± 4.5 |
| DANN | 87.2 ± 4.4 | 84.6 ± 7.7 | 91.2 ± 3.1 | 87.8 ± 3.9 | 52.0 ± 2.2 | 9.0 ± 3.5 | 60.6 ± 1.8 | 70.7 ± 1.5 | 48.8 ± 4.7 | 27.3 ± 8.6 | 44.7 ± 13.3 | 48.5 ± 1.7 | 55.7 ± 6.1 | 44.7 ± 7.0 | 83.9 ± 3.4 | 61.8 ± 4.6 |
| SagNet | 69.7 ± 9.8 | 54.5 ± 16.8 | 90.4 ± 2.6 | 96.2 ± 0.7 | 47.4 ± 0.3 | 2.4 ± 0.7 | 65.7 ± 3.4 | 66.8 ± 3.0 | 44.6 ± 0.7 | 13.3 ± 2.3 | 55.4 ± 8.8 | 51.5 ± 0.7 | 52.9 ± 3.2 | 38.3 ± 1.2 | 82.0 ± 2.1 | 65.1 ± 5.8 |
| VREx | 62.0 ± 8.4 | 43.4 ± 13.1 | 87.1 ± 3.7 | 95.1 ± 1.1 | 48.0 ± 0.9 | 2.6 ± 1.3 | 61.6 ± 4.1 | 67.8 ± 1.9 | 42.4 ± 1.3 | 11.4 ± 3.0 | 55.9 ± 8.7 | 49.4 ± 2.0 | 53.6 ± 2.9 | 43.1 ± 2.2 | 83.3 ± 2.7 | 60.0 ± 3.2 |
| Fish | 80.9 ± 10.1 | 75.4 ± 18.1 | 80.8 ± 13.2 | 95.1 ± 1.2 | 49.0 ± 0.5 | 2.5 ± 0.7 | 61.8 ± 0.3 | 69.7 ± 2.2 | 43.8 ± 0.8 | 12.1 ± 0.5 | 59.9 ± 6.5 | 50.7 ± 2.2 | 52.8 ± 1.7 | 36.0 ± 5.1 | 82.3 ± 1.2 | 66.2 ± 2.5 |
| Fishr | 63.4 ± 13.6 | 47.0 ± 22.6 | 86.7 ± 4.2 | 93.4 ± 0.5 | 48.6 ± 1.2 | 3.3 ± 1.0 | 59.4 ± 0.8 | 65.1 ± 2.2 | 44.6 ± 1.1 | 13.4 ± 0.7 | 55.7 ± 6.0 | 51.6 ± 2.5 | 53.4 ± 1.9 | 39.8 ± 2.4 | 80.5 ± 1.8 | 63.2 ± 4.1 |
| EQRM | 67.3 ± 12.2 | 56.7 ± 17.0 | 74.7 ± 7.8 | 86.3 ± 4.4 | 47.9 ± 0.5 | 1.6 ± 0.6 | 61.6 ± 1.5 | 65.9 ± 0.7 | 42.4 ± 1.1 | 9.7 ± 1.5 | 65.4 ± 0.5 | 50.5 ± 2.7 | 53.9 ± 2.5 | 39.3 ± 2.0 | 80.9 ± 1.7 | 64.9 ± 2.8 |
| RDM | 73.9 ± 15.5 | 64.8 ± 26.1 | 80.7 ± 3.3 | 94.3 ± 1.7 | 46.8 ± 1.1 | 1.3 ± 0.1 | 61.5 ± 3.6 | 63.0 ± 0.2 | 42.3 ± 1.2 | 10.3 ± 1.3 | 59.6 ± 5.6 | 48.8 ± 0.4 | 55.9 ± 2.7 | 45.9 ± 1.0 | 84.5 ± 1.4 | 62.2 ± 4.8 |
| PGrad | 76.1 ± 16.1 | 66.4 ± 26.0 | 87.7 ± 2.4 | 96.1 ± 0.6 | 49.0 ± 1.0 | 2.9 ± 1.6 | 63.4 ± 0.4 | 65.3 ± 1.6 | 43.1 ± 0.7 | 11.1 ± 0.7 | 44.3 ± 3.6 | 51.2 ± 0.7 | 54.6 ± 2.4 | 39.4 ± 0.9 | 79.1 ± 2.9 | 67.8 ± 3.6 |
| TCRI | 60.6 ± 11.0 | 40.0 ± 17.9 | 87.8 ± 1.8 | 98.3 ± 0.7 | 49.5 ± 0.2 | 4.4 ± 1.3 | 58.7 ± 1.4 | 67.7 ± 1.2 | 43.2 ± 1.1 | 13.5 ± 3.8 | 50.0 ± 4.2 | 52.1 ± 2.9 | 54.9 ± 4.0 | 40.7 ± 5.0 | 80.7 ± 1.6 | 65.2 ± 4.4 |
| Focal | 66.1 ± 12.3 | 53.7 ± 18.8 | 80.7 ± 6.3 | 93.4 ± 1.0 | 48.0 ± 0.9 | 2.3 ± 1.4 | 62.0 ± 1.9 | 63.7 ± 1.1 | 41.1 ± 1.2 | 11.1 ± 1.3 | 73.3 ± 5.2 | 48.3 ± 0.7 | 50.8 ± 1.2 | 32.8 ± 2.8 | 80.5 ± 3.1 | 64.8 ± 4.7 |
| ReWeight | 64.0 ± 11.6 | 50.4 ± 19.8 | 72.7 ± 5.9 | 96.5 ± 1.1 | 47.3 ± 1.3 | 2.9 ± 1.3 | 61.7 ± 2.7 | 68.6 ± 2.1 | 43.0 ± 0.2 | 12.2 ± 1.3 | 67.2 ± 3.4 | 51.4 ± 1.8 | 51.6 ± 2.8 | 36.9 ± 8.2 | 69.6 ± 10.8 | 65.2 ± 4.1 |
| BSoftmax | 57.0 ± 5.7 | 34.2 ± 9.4 | 90.5 ± 1.9 | 96.5 ± 0.4 | 50.2 ± 1.9 | 7.4 ± 3.3 | 54.3 ± 6.2 | 71.3 ± 3.0 | 43.5 ± 1.0 | 14.4 ± 1.2 | 80.6 ± 4.4 | 48.8 ± 1.4 | 50.4 ± 3.2 | 34.9 ± 2.9 | 78.6 ± 3.7 | 60.1 ± 5.0 |
| LDAM | 59.7 ± 9.3 | 43.8 ± 17.5 | 75.4 ± 12.1 | 93.4 ± 1.0 | 49.8 ± 1.3 | 3.6 ± 2.3 | 62.6 ± 4.4 | 66.0 ± 1.9 | 40.1 ± 1.0 | 7.9 ± 0.6 | 73.1 ± 9.7 | 51.4 ± 2.3 | 51.1 ± 3.2 | 32.6 ± 1.7 | 81.0 ± 5.0 | 64.4 ± 5.7 |
| BoDA | 89.6 ± 1.8 | 90.5 ± 3.7 | 79.9 ± 5.8 | 95.8 ± 0.8 | 49.0 ± 2.4 | 8.5 ± 2.4 | 64.6 ± 3.7 | 62.4 ± 1.7 | 45.3 ± 1.2 | 20.8 ± 1.4 | 69.4 ± 3.7 | 50.3 ± 1.5 | 49.0 ± 3.0 | 41.1 ± 1.2 | 79.7 ± 1.8 | 54.7 ± 4.1 |
| GINIDG | 94.6 ± 1.4 | 97.4 ± 0.7 | 84.4 ± 11.0 | 93.6 ± 1.8 | 46.6 ± 1.9 | 2.5 ± 0.5 | 61.4 ± 2.6 | 60.2 ± 0.5 | 47.5 ± 2.3 | 33.3 ± 9.7 | 53.7 ± 15.5 | 45.2 ± 2.5 | 45.5 ± 3.8 | 29.7 ± 5.8 | 74.2 ± 4.4 | 58.8 ± 6.6 |
| SAMALTDG | 86.7 ± 6.4 | 82.7 ± 8.7 | 93.7 ± 3.7 | 93.2 ± 1.0 | 48.5 ± 0.5 | 1.9 ± 0.3 | 64.8 ± 0.2 | 68.9 ± 2.2 | 43.9 ± 0.6 | 9.1 ± 1.0 | 69.3 ± 3.0 | 53.7 ± 0.5 | 50.8 ± 2.7 | 35.9 ± 1.7 | 79.5 ± 1.3 | 61.1 ± 3.3 |
| NDCL | 94.0 ± 0.9 | 92.8 ± 1.1 | 96.9 ± 1.1 | 96.5 ± 0.5 | 49.4 ± 0.9 | 3.0 ± 1.2 | 59.6 ± 0.8 | 69.6 ± 2.1 | 43.8 ± 0.7 | 11.6 ± 2.0 | 69.2 ± 7.0 | 48.8 ± 1.7 | 59.1 ± 3.7 | 46.5 ± 4.2 | 79.8 ± 1.9 | 67.8 ± 5.7 |

*Table 21.* Detailed results of the target domain on PACS benchmark under the Duality setting.

| Method | Average | A Average | A Many | A Median | A Few | C Average | C Many | C Median | C Few | P Average | P Many | P Median | P Few | S Average | S Many | S Median | S Few |
|---|---|---|---|---|---|---|---|---|---|---|---|---|---|---|---|---|---|
| ERM | 67.3 ± 1.3 | 62.0 ± 1.8 | 27.7 ± 3.5 | 80.4 ± 0.3 | 77.3 ± 1.0 | 62.0 ± 1.8 | 53.1 ± 1.6 | 95.5 ± 0.1 | 43.3 ± 3.9 | 86.8 ± 0.6 | 100.0 ± 0.0 | 94.2 ± 0.6 | 67.5 ± 2.3 | 55.5 ± 2.5 | 54.2 ± 4.5 | 54.5 ± 5.9 | 58.8 ± 4.4 |
| IRM | 70.7 ± 1.3 | 57.8 ± 0.8 | 39.3 ± 5.5 | 82.0 ± 0.7 | 79.6 ± 0.7 | 57.8 ± 0.8 | 37.1 ± 8.9 | 95.7 ± 0.5 | 40.6 ± 3.9 | 84.4 ± 1.8 | 100.0 ± 0.0 | 94.9 ± 0.5 | 60.0 ± 4.8 | 47.9 ± 2.3 | 27.6 ± 14.8 | 48.7 ± 7.2 | 54.0 ± 2.1 |
| GroupDRO | 67.4 ± 1.1 | 60.4 ± 2.1 | 31.4 ± 1.7 | 80.2 ± 0.7 | 75.5 ± 2.1 | 60.4 ± 2.1 | 38.5 ± 5.9 | 96.6 ± 0.4 | 44.1 ± 3.1 | 82.0 ± 0.4 | 100.0 ± 0.0 | 93.2 ± 1.5 | 55.0 ± 1.7 | 56.9 ± 2.8 | 38.5 ± 5.2 | 66.1 ± 7.2 | 55.7 ± 4.9 |
| Mixup | 68.3 ± 0.4 | 56.4 ± 1.9 | 35.7 ± 5.7 | 83.0 ± 1.9 | 73.8 ± 3.3 | 56.4 ± 1.9 | 25.3 ± 4.2 | 94.8 ± 0.6 | 42.0 ± 2.2 | 84.6 ± 1.3 | 100.0 ± 0.0 | 95.6 ± 0.7 | 60.3 ± 3.2 | 46.3 ± 1.6 | 51.0 ± 9.1 | 53.8 ± 6.1 | 39.8 ± 3.0 |
| MLDG | 70.0 ± 0.8 | 59.3 ± 2.6 | 42.8 ± 2.5 | 84.3 ± 0.6 | 74.9 ± 2.7 | 59.3 ± 2.6 | 44.2 ± 1.6 | 96.4 ± 1.0 | 40.4 ± 4.7 | 86.3 ± 2.9 | 100.0 ± 0.0 | 94.5 ± 0.3 | 65.7 ± 8.3 | 48.8 ± 0.2 | 54.0 ± 20.8 | 55.4 ± 6.7 | 50.6 ± 5.6 |
| CORAL | 67.8 ± 1.3 | 55.3 ± 2.0 | 45.1 ± 5.6 | 83.6 ± 0.5 | 68.6 ± 4.2 | 55.3 ± 2.0 | 46.4 ± 5.4 | 97.0 ± 0.5 | 31.1 ± 3.0 | 84.2 ± 1.2 | 100.0 ± 0.0 | 95.9 ± 0.7 | 58.6 ± 4.5 | 52.8 ± 6.2 | 49.0 ± 4.1 | 64.6 ± 8.6 | 51.3 ± 5.7 |
| MMD | 73.5 ± 2.1 | 63.5 ± 1.6 | 47.6 ± 6.9 | 85.1 ± 1.2 | 77.2 ± 3.2 | 63.5 ± 1.6 | 44.2 ± 3.4 | 96.9 ± 0.3 | 47.9 ± 3.6 | 85.0 ± 0.8 | 100.0 ± 0.0 | 93.5 ± 1.2 | 62.8 ± 3.9 | 50.6 ± 4.2 | 45.4 ± 11.4 | 52.3 ± 4.6 | 50.7 ± 4.3 |
| DANN | 72.3 ± 0.9 | 65.3 ± 1.6 | 47.2 ± 4.8 | 85.0 ± 2.0 | 75.4 ± 2.2 | 65.3 ± 1.6 | 59.2 ± 7.1 | 97.2 ± 0.4 | 45.8 ± 5.4 | 87.7 ± 0.8 | 100.0 ± 0.0 | 92.8 ± 1.4 | 72.0 ± 1.1 | 48.0 ± 3.8 | 57.0 ± 11.0 | 57.2 ± 11.2 | 48.2 ± 1.7 |
| SagNet | 70.6 ± 1.4 | 59.2 ± 1.4 | 37.4 ± 8.6 | 86.3 ± 1.3 | 75.7 ± 1.7 | 59.2 ± 1.4 | 52.0 ± 7.4 | 95.9 ± 0.8 | 38.1 ± 4.3 | 84.8 ± 2.6 | 100.0 ± 0.0 | 94.7 ± 1.6 | 62.2 ± 5.5 | 49.5 ± 4.3 | 24.0 ± 3.5 | 53.1 ± 6.0 | 50.9 ± 7.3 |
| VREx | 67.8 ± 2.2 | 56.7 ± 0.1 | 40.0 ± 5.5 | 79.7 ± 2.2 | 75.1 ± 2.3 | 56.7 ± 0.1 | 39.5 ± 0.2 | 97.3 ± 0.5 | 36.6 ± 0.8 | 86.5 ± 0.6 | 100.0 ± 0.0 | 94.0 ± 1.0 | 67.0 ± 2.8 | 51.5 ± 5.0 | 46.5 ± 6.1 | 50.8 ± 6.6 | 57.1 ± 6.4 |
| Fish | 69.8 ± 1.4 | 58.4 ± 1.4 | 39.0 ± 1.3 | 82.0 ± 2.1 | 75.3 ± 2.8 | 58.4 ± 1.4 | 48.2 ± 7.7 | 95.3 ± 0.6 | 40.7 ± 1.6 | 84.7 ± 1.0 | 100.0 ± 0.0 | 94.9 ± 0.7 | 60.8 ± 2.9 | 49.9 ± 2.6 | 58.5 ± 5.5 | 48.2 ± 7.0 | 52.6 ± 5.8 |
| Fishr | 69.1 ± 1.3 | 58.4 ± 1.4 | 45.7 ± 1.1 | 85.4 ± 1.8 | 69.6 ± 5.0 | 58.4 ± 1.4 | 39.7 ± 2.1 | 97.0 ± 0.0 | 39.9 ± 2.6 | 86.1 ± 2.4 | 100.0 ± 0.0 | 94.8 ± 1.0 | 65.2 ± 6.6 | 52.2 ± 3.9 | 58.5 ± 3.0 | 40.7 ± 4.5 | 52.4 ± 3.6 |
| EQRM | 68.1 ± 4.1 | 66.8 ± 3.4 | 32.7 ± 11.7 | 84.2 ± 1.5 | 74.6 ± 3.0 | 66.8 ± 3.4 | 58.0 ± 9.5 | 96.6 ± 0.6 | 49.7 ± 2.6 | 86.5 ± 0.7 | 100.0 ± 0.0 | 92.9 ± 2.4 | 68.3 ± 0.7 | 45.3 ± 2.4 | 21.0 ± 7.6 | 49.8 ± 5.3 | 50.7 ± 4.0 |
| RDM | 72.5 ± 1.8 | 58.9 ± 1.1 | 42.4 ± 9.3 | 83.6 ± 1.4 | 80.2 ± 1.1 | 58.9 ± 1.1 | 43.3 ± 3.8 | 97.1 ± 0.3 | 38.8 ± 3.1 | 82.7 ± 1.5 | 100.0 ± 0.0 | 92.8 ± 0.1 | 57.3 ± 4.2 | 54.4 ± 4.3 | 41.5 ± 2.6 | 60.4 ± 5.4 | 58.1 ± 2.3 |
| PGrad | 75.3 ± 0.5 | 62.2 ± 0.9 | 53.5 ± 3.8 | 85.5 ± 0.9 | 79.1 ± 3.4 | 62.2 ± 0.9 | 44.4 ± 6.9 | 97.7 ± 0.2 | 44.1 ± 3.5 | 87.9 ± 0.4 | 100.0 ± 0.0 | 95.8 ± 0.8 | 69.2 ± 4.6 | 56.9 ± 3.6 | 30.3 ± 10.1 | 56.7 ± 6.5 | 62.3 ± 3.2 |
| TCRI | 68.8 ± 1.4 | 61.9 ± 3.1 | 53.6 ± 3.7 | 81.7 ± 4.0 | 67.2 ± 4.0 | 61.9 ± 3.1 | 56.6 ± 9.1 | 96.5 ± 0.5 | 40.3 ± 4.1 | 87.1 ± 1.1 | 100.0 ± 0.0 | 95.6 ± 1.1 | 66.8 ± 4.4 | 58.3 ± 1.3 | 67.8 ± 4.3 | 62.5 ± 3.6 | 54.3 ± 3.3 |
| Focal | 69.1 ± 1.2 | 63.3 ± 1.6 | 33.5 ± 6.6 | 82.5 ± 1.4 | 78.0 ± 0.7 | 63.3 ± 1.6 | 61.0 ± 10.6 | 95.4 ± 1.1 | 42.8 ± 4.7 | 85.9 ± 2.2 | 100.0 ± 0.0 | 92.9 ± 1.8 | 66.4 ± 4.5 | 52.0 ± 4.3 | 49.4 ± 5.3 | 51.9 ± 14.6 | 55.3 ± 6.6 |
| ReWeight | 70.1 ± 1.6 | 58.0 ± 1.6 | 42.1 ± 5.0 | 83.2 ± 1.9 | 75.2 ± 1.8 | 58.0 ± 1.6 | 31.6 ± 3.5 | 96.7 ± 1.1 | 41.7 ± 4.0 | 86.3 ± 1.7 | 100.0 ± 0.0 | 92.3 ± 2.5 | 68.2 ± 2.3 | 48.4 ± 5.9 | 53.4 ± 10.4 | 35.9 ± 10.0 | 53.7 ± 2.9 |
| BSoftmax | 68.2 ± 1.5 | 60.7 ± 1.4 | 22.4 ± 4.7 | 84.2 ± 1.5 | 78.6 ± 1.7 | 60.7 ± 1.4 | 36.2 ± 5.2 | 96.3 ± 0.2 | 45.0 ± 2.1 | 88.7 ± 1.3 | 100.0 ± 0.0 | 95.0 ± 1.6 | 72.4 ± 3.8 | 53.1 ± 4.7 | 50.9 ± 8.7 | 62.6 ± 8.6 | 52.1 ± 2.9 |
| LDAM | 67.9 ± 1.1 | 59.4 ± 2.8 | 33.5 ± 5.4 | 82.2 ± 0.8 | 74.9 ± 1.8 | 59.4 ± 2.8 | 30.4 ± 1.6 | 96.0 ± 0.5 | 45.5 ± 4.9 | 84.0 ± 0.9 | 100.0 ± 0.0 | 92.3 ± 1.7 | 61.8 ± 1.8 | 55.1 ± 1.3 | 39.3 ± 12.7 | 48.2 ± 5.6 | 64.7 ± 1.1 |
| BoDA | 68.4 ± 0.5 | 56.1 ± 1.2 | 30.2 ± 4.8 | 83.6 ± 0.6 | 76.3 ± 2.4 | 56.1 ± 1.2 | 45.4 ± 2.7 | 97.4 ± 0.3 | 32.6 ± 3.3 | 85.0 ± 0.5 | 100.0 ± 0.0 | 94.6 ± 1.0 | 62.1 ± 2.0 | 48.4 ± 6.3 | 45.3 ± 6.8 | 63.3 ± 9.0 | 45.3 ± 7.3 |
| GINIDG | 64.5 ± 1.7 | 52.4 ± 2.5 | 39.4 ± 3.5 | 78.5 ± 2.0 | 68.2 ± 2.2 | 52.4 ± 2.5 | 42.2 ± 13.4 | 91.5 ± 2.0 | 31.0 ± 1.0 | 85.5 ± 1.4 | 100.0 ± 0.0 | 93.2 ± 2.0 | 65.4 ± 2.3 | 50.4 ± 5.4 | 60.0 ± 19.7 | 50.5 ± 7.7 | 57.4 ± 5.8 |
| SAMALTDG | 67.7 ± 0.6 | 61.0 ± 1.3 | 27.4 ± 1.2 | 81.0 ± 2.2 | 77.3 ± 1.2 | 61.0 ± 1.3 | 40.5 ± 5.1 | 96.5 ± 0.4 | 44.9 ± 1.6 | 86.1 ± 0.5 | 100.0 ± 0.0 | 95.4 ± 0.2 | 64.8 ± 1.6 | 49.7 ± 2.7 | 30.7 ± 11.2 | 53.9 ± 5.3 | 52.7 ± 1.8 |
| **NDCL** | **70.7 ± 1.8** | **62.8 ± 0.6** | **39.8 ± 2.5** | **83.5 ± 3.2** | **78.7 ± 0.2** | **62.8 ± 0.6** | **46.1 ± 9.6** | **94.2 ± 0.8** | **48.0 ± 3.3** | **86.8 ± 1.0** | **100.0 ± 0.0** | **94.6 ± 0.3** | **67.1 ± 2.9** | **65.3 ± 2.2** | **32.0 ± 5.1** | **72.0 ± 3.9** | **69.1 ± 1.4** |

*Table 22.* Detailed results of the target domain on OfficeHome benchmark under the Duality setting.

| Method | Average | A Average | A Many | A Median | A Few | C Average | C Many | C Median | C Few | P Average | P Many | P Median | P Few | R Average | R Many | R Median | R Few |
|---|---|---|---|---|---|---|---|---|---|---|---|---|---|---|---|---|---|
| ERM | 48.6 ± 0.7 | 42.5 ± 0.4 | 48.0 ± 2.6 | 49.3 ± 0.0 | 38.3 ± 1.2 | 42.5 ± 0.4 | 55.1 ± 0.2 | 55.1 ± 0.2 | 36.7 ± 0.7 | 59.6 ± 0.5 | 63.2 ± 4.3 | 75.2 ± 0.4 | 50.1 ± 0.6 | 58.7 ± 0.9 | 87.3 ± 0.5 | 67.2 ± 0.4 | 46.6 ± 1.3 |
| IRM | 44.1 ± 1.4 | 39.6 ± 1.3 | 47.6 ± 0.8 | 42.8 ± 2.3 | 34.7 ± 1.5 | 39.6 ± 1.3 | 54.7 ± 2.3 | 54.7 ± 2.3 | 33.3 ± 1.1 | 56.5 ± 1.4 | 61.7 ± 6.2 | 73.3 ± 1.7 | 46.5 ± 1.1 | 57.5 ± 0.6 | 86.5 ± 1.2 | 69.6 ± 0.7 | 44.8 ± 1.4 |
| GroupDRO | 45.2 ± 0.2 | 42.9 ± 0.6 | 53.3 ± 0.8 | 44.0 ± 2.6 | 35.3 ± 0.5 | 42.9 ± 0.6 | 55.5 ± 0.9 | 55.5 ± 0.9 | 36.7 ± 0.7 | 59.1 ± 0.5 | 64.9 ± 3.3 | 77.4 ± 1.0 | 48.4 ± 0.8 | 59.6 ± 0.3 | 90.6 ± 0.7 | 71.8 ± 0.6 | 46.5 ± 0.1 |
| Mixup | 46.1 ± 1.4 | 42.8 ± 0.7 | 51.8 ± 0.4 | 45.6 ± 1.8 | 36.2 ± 2.3 | 42.8 ± 0.7 | 55.0 ± 0.4 | 55.0 ± 0.4 | 36.9 ± 1.3 | 59.5 ± 0.8 | 70.0 ± 3.3 | 76.9 ± 0.8 | 48.0 ± 0.8 | 60.6 ± 0.7 | 86.7 ± 0.8 | 72.8 ± 0.6 | 47.9 ± 0.5 |
| MLDG | 49.1 ± 0.6 | 42.1 ± 1.4 | 46.5 ± 1.4 | 47.9 ± 2.2 | 39.6 ± 1.3 | 42.1 ± 1.4 | 56.6 ± 1.0 | 56.6 ± 1.0 | 36.6 ± 1.0 | 58.4 ± 1.1 | 61.9 ± 4.8 | 76.9 ± 0.7 | 48.2 ± 1.3 | 60.5 ± 0.7 | 86.5 ± 0.7 | 71.0 ± 0.4 | 48.5 ± 0.9 |
| CORAL | 50.6 ± 0.8 | 45.0 ± 0.2 | 57.4 ± 2.5 | 52.5 ± 0.9 | 39.6 ± 0.9 | 45.0 ± 0.2 | 58.8 ± 0.5 | 58.8 ± 0.5 | 38.5 ± 0.6 | 60.4 ± 0.7 | 70.6 ± 4.3 | 78.8 ± 0.2 | 48.9 ± 0.5 | 61.9 ± 0.4 | 91.2 ± 0.5 | 76.3 ± 0.1 | 48.1 ± 0.4 |
| MMD | 45.7 ± 0.3 | 41.2 ± 1.0 | 51.0 ± 2.1 | 43.1 ± 2.3 | 35.7 ± 1.9 | 41.2 ± 1.0 | 54.1 ± 1.4 | 54.1 ± 1.4 | 35.7 ± 1.9 | 59.3 ± 0.9 | 61.5 ± 3.6 | 75.9 ± 0.7 | 49.1 ± 1.1 | 57.4 ± 0.2 | 89.6 ± 1.6 | 68.2 ± 0.4 | 44.1 ± 0.7 |
| DANN | 49.3 ± 0.6 | 40.6 ± 0.3 | 48.6 ± 1.3 | 43.1 ± 1.9 | 36.6 ± 0.4 | 40.6 ± 0.3 | 54.0 ± 1.3 | 54.0 ± 1.3 | 35.5 ± 1.4 | 58.0 ± 1.4 | 66.2 ± 6.4 | 74.4 ± 2.3 | 47.0 ± 1.6 | 59.2 ± 0.8 | 89.7 ± 1.1 | 72.1 ± 1.9 | 45.7 ± 0.8 |
| SagNet | 48.7 ± 0.0 | 44.3 ± 0.8 | 55.1 ± 2.2 | 47.3 ± 1.1 | 40.7 ± 1.4 | 44.3 ± 0.8 | 56.7 ± 0.9 | 56.7 ± 0.9 | 38.5 ± 0.9 | 60.6 ± 0.3 | 66.5 ± 4.2 | 77.7 ± 0.6 | 49.9 ± 1.1 | 73.6 ± 0.9 | 90.5 ± 1.7 | 73.6 ± 0.9 | 48.9 ± 1.3 |
| VREx | 48.6 ± 1.0 | 43.4 ± 0.9 | 50.6 ± 1.4 | 46.3 ± 1.7 | 39.0 ± 1.2 | 43.4 ± 0.9 | 57.2 ± 0.7 | 57.2 ± 0.7 | 36.9 ± 1.3 | 58.5 ± 0.4 | 65.5 ± 3.1 | 75.6 ± 0.6 | 48.3 ± 0.8 | 60.9 ± 0.2 | 86.5 ± 3.0 | 71.0 ± 1.1 | 49.1 ± 0.6 |
| Fish | 47.5 ± 0.4 | 43.3 ± 1.1 | 48.4 ± 1.3 | 46.6 ± 1.0 | 37.4 ± 0.1 | 43.3 ± 1.1 | 56.9 ± 2.2 | 56.9 ± 2.2 | 37.4 ± 1.0 | 60.2 ± 0.6 | 62.3 ± 4.0 | 76.8 ± 0.6 | 50.5 ± 0.4 | 60.5 ± 0.2 | 89.8 ± 0.9 | 69.9 ± 0.7 | 47.8 ± 0.5 |
| Fishr | 47.5 ± 0.4 | 41.8 ± 0.2 | 51.7 ± 1.7 | 46.3 ± 2.3 | 37.5 ± 0.8 | 41.8 ± 0.2 | 53.2 ± 1.2 | 53.2 ± 1.2 | 36.5 ± 1.0 | 60.3 ± 0.7 | 63.5 ± 3.5 | 78.4 ± 0.2 | 49.8 ± 1.1 | 60.3 ± 0.7 | 89.8 ± 1.2 | 70.2 ± 1.8 | 48.4 ± 0.9 |
| EQRM | 48.3 ± 0.4 | 42.5 ± 0.6 | 48.2 ± 1.0 | 47.2 ± 1.2 | 38.2 ± 0.7 | 42.5 ± 0.6 | 55.5 ± 0.5 | 55.5 ± 0.5 | 37.7 ± 0.4 | 59.4 ± 0.9 | 62.4 ± 3.9 | 78.2 ± 0.5 | 48.7 ± 1.3 | 59.5 ± 1.0 | 86.7 ± 0.5 | 71.2 ± 0.5 | 46.9 ± 1.3 |
| RDM | 48.2 ± 0.6 | 42.8 ± 0.7 | 51.5 ± 1.0 | 46.4 ± 1.3 | 38.2 ± 1.0 | 42.8 ± 0.7 | 55.2 ± 1.4 | 55.2 ± 1.4 | 37.6 ± 1.1 | 60.5 ± 0.8 | 65.5 ± 4.0 | 75.2 ± 1.1 | 51.1 ± 1.5 | 60.1 ± 0.6 | 89.5 ± 1.3 | 69.4 ± 1.1 | 47.6 ± 0.9 |
| PGrad | 51.4 ± 0.3 | 45.6 ± 1.0 | 56.4 ± 1.2 | 50.9 ± 0.9 | 41.8 ± 0.6 | 45.6 ± 1.0 | 58.9 ± 1.5 | 58.9 ± 1.5 | 39.6 ± 0.4 | 61.2 ± 0.3 | 65.7 ± 1.9 | 79.9 ± 0.8 | 50.6 ± 0.7 | 63.5 ± 1.2 | 90.9 ± 0.9 | 73.0 ± 1.4 | 52.1 ± 1.3 |
| TCRI_HSIC | 50.8 ± 1.3 | 43.3 ± 0.5 | 52.1 ± 1.1 | 48.5 ± 4.4 | 42.2 ± 1.3 | 43.3 ± 0.5 | 57.4 ± 2.4 | 57.4 ± 2.4 | 37.3 ± 1.4 | 61.5 ± 0.4 | 67.0 ± 3.4 | 78.1 ± 1.4 | 51.4 ± 0.7 | 62.7 ± 0.8 | 88.6 ± 1.8 | 73.9 ± 1.4 | 50.8 ± 0.9 |
| Focal | 45.9 ± 1.4 | 41.9 ± 0.8 | 49.3 ± 2.9 | 42.5 ± 2.7 | 37.3 ± 1.4 | 41.9 ± 0.8 | 54.3 ± 1.6 | 54.3 ± 1.6 | 36.6 ± 0.7 | 57.1 ± 1.1 | 58.7 ± 2.5 | 75.3 ± 1.5 | 47.3 ± 1.1 | 58.1 ± 0.2 | 88.2 ± 1.7 | 69.0 ± 1.7 | 44.9 ± 0.2 |
| ReWeight | 48.2 ± 0.7 | 42.2 ± 0.7 | 51.0 ± 2.2 | 46.8 ± 1.6 | 40.3 ± 0.1 | 42.2 ± 0.7 | 52.9 ± 0.9 | 52.9 ± 0.9 | 37.3 ± 0.8 | 59.7 ± 0.1 | 64.3 ± 3.5 | 74.9 ± 0.6 | 49.8 ± 0.5 | 59.7 ± 0.5 | 86.3 ± 0.6 | 70.6 ± 1.1 | 47.2 ± 0.7 |
| BSoftmax | 48.2 ± 0.8 | 42.6 ± 1.7 | 47.4 ± 2.0 | 46.1 ± 0.9 | 41.3 ± 1.3 | 42.6 ± 1.7 | 53.1 ± 1.1 | 53.1 ± 1.1 | 38.0 ± 2.1 | 60.5 ± 0.7 | 65.4 ± 3.7 | 78.4 ± 0.6 | 50.3 ± 1.6 | 59.8 ± 0.8 | 88.0 ± 1.5 | 70.8 ± 1.6 | 48.0 ± 0.6 |
| LDAM | 46.2 ± 0.9 | 41.5 ± 0.5 | 48.4 ± 1.3 | 44.0 ± 1.4 | 37.2 ± 0.9 | 41.5 ± 0.5 | 56.1 ± 1.1 | 56.1 ± 1.1 | 35.3 ± 0.6 | 55.3 ± 0.9 | 59.8 ± 3.8 | 70.8 ± 0.6 | 45.4 ± 0.9 | 58.7 ± 1.3 | 88.7 ± 1.3 | 67.6 ± 2.0 | 45.2 ± 1.6 |
| BoDA | 49.3 ± 1.0 | 44.0 ± 1.1 | 61.4 ± 0.3 | 51.6 ± 0.8 | 38.4 ± 1.5 | 44.0 ± 1.1 | 57.5 ± 0.7 | 57.5 ± 0.7 | 36.8 ± 1.8 | 59.5 ± 0.3 | 67.2 ± 2.6 | 78.0 ± 0.6 | 48.4 ± 0.8 | 61.1 ± 0.6 | 91.5 ± 1.4 | 74.9 ± 0.4 | 47.3 ± 0.5 |
| GINIDG | 45.7 ± 0.9 | 41.4 ± 1.0 | 52.2 ± 1.7 | 42.5 ± 1.8 | 36.5 ± 1.6 | 41.4 ± 1.0 | 54.9 ± 0.9 | 54.9 ± 0.9 | 35.2 ± 0.9 | 57.1 ± 0.1 | 63.3 ± 2.8 | 74.0 ± 1.0 | 46.4 ± 0.3 | 56.6 ± 0.9 | 89.4 ± 1.2 | 67.9 ± 0.4 | 43.8 ± 1.2 |
| SAMALTDG | 42.4 ± 1.5 | 37.8 ± 0.9 | 63.7 ± 3.4 | 59.2 ± 1.3 | 20.5 ± 1.6 | 37.8 ± 0.9 | 58.5 ± 0.6 | 58.5 ± 0.6 | 20.5 ± 0.6 | 49.2 ± 0.9 | 84.6 ± 0.7 | 60.5 ± 0.5 | 30.3 ± 1.2 | 54.8 ± 0.5 | 84.3 ± 0.5 | 77.8 ± 0.5 | 31.2 ± 0.5 |
| **NDCL** | **50.0 ± 0.5** | **45.7 ± 0.6** | **52.3 ± 1.5** | **51.5 ± 2.7** | **41.4 ± 0.8** | **45.7 ± 0.6** | **58.2 ± 1.6** | **58.2 ± 1.6** | **41.5 ± 0.8** | **63.5 ± 0.5** | **66.7 ± 4.7** | **80.2 ± 1.4** | **54.4 ± 0.8** | **63.6 ± 0.3** | **90.9 ± 1.8** | **73.2 ± 0.9** | **53.2 ± 1.0** |

