# OpenReview forum: "Negatives-Dominant Contrastive Learning for Generalization in Imbalanced Domains"
_ICML.cc/2026/Conference — ICML 2026 regular_

### Official Review · Reviewer_uVxV · 2026-03-03

**Soundness:** 3
**Presentation:** 3
**Significance:** 3
**Originality:** 2
**Overall Recommendation:** 4
**Confidence:** 4

**Summary:**

This paper addresses the challenge of Imbalanced Domain Generalization (IDG), where models must handle simultaneous domain and label shifts that distort decision boundaries. The authors first establish a theoretical generalization bound that identifies posterior discrepancy and decision margin as key factors. They propose the Negative-Dominant Contrastive Learning (NDCL) framework, which amplifies the signal from negative samples to separate classes and employs alignment strategies to ensure prediction consistency across domains. Extensive experiments on standard benchmarks demonstrate that NDCL effectively mitigates the compounded challenges of label and domain shift.

**Compliance With Llm Reviewing Policy:**

Affirmed.

**Final Justification:**

The paper is well-structured and clearly written. The paper establishes a theoretical generalization bound, and the experimental design is rigorous.

**Key Questions For Authors:**

1. Contrastive learning methods are notoriously sensitive to batch size. If the batch size is small, the set of available negatives is insufficient. In contrast, if the batch size is large, the computational overhead becomes significant, especially for the pairwise similarity calculations in Eq. (1). Could the authors clarify the trade-off between batch size, computational cost, and performance stability? Does NDCL remain practical on memory-constrained hardware where large batch sizes are infeasible?

2. For majority classes, the number of negatives is naturally small. Does the denominator term $\sum_{n\in N(i)}(1-s(p_i,p_n))$ become unstable or small for majority classes, potentially leading to exploding gradients or insufficient repulsion signals? How does the method ensure that the decision boundaries of majority classes remain well-defined?

3. The prediction-central alignment strategy (Eq. (3)) relies on prediction prototypes $\mu_k^d$. But the text does not explicitly detail the update mechanism of these prototypes. Could you clarify the update rule?

**Limitations:**

yes

**Strengths And Weaknesses:**

**Strengths:**

1. The paper establishes a theoretical generalization bound for IDG, formally motivating the need for posterior consistency and decision margin enlargement. This theoretical grounding is a significant strength, providing a principled justification for the proposed method.

2. The proposed NDCL method is technically sound. The reformulation of the contrastive loss to prioritize negative samples is well-motivated by the insight that negatives remain relatively balanced even when positives are long-tailed.

3. The experimental design is rigorous. The authors evaluate the method on three standard benchmarks under three distinct imbalance settings, comparing against a comprehensive set of 21 baselines.

4. The paper addresses a practical problem. Real-world data often exhibits both domain and label shifts, making IDG a crucial area for advancing reliable machine learning.

**Weaknesses:**

1. The proposed method is explicitly positioned as a modification of Supervised Contrastive Learning. However, SupCon or its simple variants are absent from the experimental comparisons. Therefore it remains unclear if the performance gains stem from the novel negative-focused formulation, or simply from the inclusion of a contrastive learning framework.

2. The prediction-central alignment loss relies on computing domain-specific class prototypes. In the proposed settings, the sample size for a specific minority class in a specific domain can be extremely small. Calculating a mean vector is statistically unreliable and results in a high-variance representation.

3. The prediction-central alignment strategy explicitly requires domain indice $d$ to compute domain-specific prototypes. This requirement restricts the method's applicability in real-world scenarios.  Many modern DG approaches aim to be domain-agnostic or do not require explicit domain indices during the batch construction process.

---

> ### Author Rebuttal · Authors · 2026-03-31
>
> We sincerely thank the reviewer for their positive feedback and recognition of our work's contributions in **theoretical grounding**, **technical soundness**, **experimental rigor**, and **practical relevance**.
>
> > **[W1] The proposed method is explicitly positioned as a modification of Supervised...**
>
> **[AW1]**: Thank you for your comment.
> In fact, we reported ablation results in Tab. 8 (Page 24) comparing our framework with standard SupCon and its negative-dominant variant (SupCon-ND) using the same setting.
> The results clearly indicate that the gains **stem fundamentally from our negative-focused formulation**, not merely the contrastive framework itself.
>
> > **[W2 & Q3] The prediction-central alignment loss relies on...**
>
> **[AW2 & AQ3]**: Thank you for your insight comment.
>
> As detailed in Lines 237-238, $\mu_k^d$ is **defined in the prediction space** and computed simply as the in-batch mean of the same class from different domains.
> Therefore, **it does not require an extra update rule**.
>
> While computing means from few samples in a feature space causes high variance, doing so in the prediction space is highly stable.
> To prove this, Tab. R1 tracks the step-wise behavior of extreme minority classes ($\le 2$ samples, OfficeHome).
> The prototype similarity ("Sim") between consecutive steps rapidly stabilizes, and the corresponding $L_{con}$ ("Loss") smoothly converges.
> This empirically confirms that our prototypes do not suffer from high-variance oscillations and remain statistically reliable.
>
> Tab R1: *Step-wise behavior. Headers: steps ($\times 100$).*
>
> ||0|3|6|9|12|15|18|21|24|27|30|33|36|39|42|45|48|50|
> |-|-|-|-|-|-|-|-|-|-|-|-|-|-|-|-|-|-|-|
> |Sim|-|0.14|0.93|0.96|0.99|0.99|0.99|0.99|0.99|0.98|0.99|0.98|0.98|0.99|0.98|0.99|0.99|0.99|
> |Loss|0.62|0.39|0.39|0.39|0.38|0.39|0.38|0.38|0.39|0.39|0.38|0.38|0.38|0.39|0.38|0.38|0.39|0.39|
>
> > **[W3] The prediction-central alignment strategy explicitly requires...**
>
> **[AW3]**: While domain-agnostic DG is an important research direction, utilizing domain indices is still a reasonable and practical design choice.
> Specifically, explicit domain information actively helps capture cross-domain variations and facilitates structured modeling of distribution shifts.
> Furthermore, many recent methods [1, 2] similarly rely on source domain labels.
> Therefore, this requirement strictly follows standard DG protocols and **does not restrict our method's real-world applicability**.
>
> > **[Q1] Contrastive learning methods are notoriously sensitive...**
>
> **[AQ1]**: Thank you for your practical question.
> Following the DomainBed protocol [3] with minor adaptations for IDG, our default batch size is set to 32 per domain across all experiments (sampled from a hyperparameter search range of $[2^3, 2^{5.5}]$).
> By leveraging our augmented negative set, the effective number of samples participating in $L_{con}$ explicitly reaches approximately 240 per step.
>
> This is significantly smaller than the massive batch sizes (1024) typically required by standard contrastive learning methods, e.g., SupCon.
> Nevertheless, our NDCL achieves consistently strong performance across all three experimental settings.
>
> Consequently, our computational costs remain highly manageable.
> The entire framework, utilizing a ResNet-50 backbone, can be trained on a single 24GB GPU, which we hope alleviates concerns regarding its deployment on memory-constrained hardware."
>
> > **[Q2] For majority classes, the number of negatives is naturally small...**
>
> **[AQ2]**: Thank you for your insightful question.
> We empirically observe that the denominator term remains stable throughout training, and the decision boundaries for majority classes are not adversely affected.
>
> To substantiate this, we tracked the maximum class proportion per batch during training.
> Under TotalHeavyTail on PACS (imbalance ratio 132.78), the majority class exceeds 50% of the batch in 95.71% of iterations, but it exceeds 75% in only **0.04%**.
> On OfficeHome, it **never** exceeds 50%.
> This indicates that a sufficient number of negative samples is consistently present in each batch.
> Furthermore, by explicitly leveraging augmented hard negatives near the boundaries, the corresponding $L_{con}$ smoothly converges to 0.32 (similar to Tab. R1), suggesting that the repulsion signal remains effective during training.
>
> Intuitively, majority classes are already well-optimized due to their abundant samples.
> The role of $L_{con}$ is primarily to push minority representations away from majority regions, thereby sharpening class boundaries, while $L_{ce}$ provides a stable anchor for majority classes and prevents boundary degradation.
>
> [1] Zhao, Di, et al. Unlearning during Training: Domain-Specific Gradient Ascent for Domain Generalization. ICLR. 2026.
> [2] Yi, Huang, et al. Domain-Aware Adversarial Domain Augmentation Network for Hyperspectral Image Classification. TIP. 2026.
> [3] Gulrajani, Ishaan, et al. In Search of Lost Domain Generalization. ICLR. 2021.

---

> > ### Author Rebuttal · Reviewer_uVxV · 2026-04-01
> >
> > Thank the authors for the detailed and thoughtful rebuttal. The response has adequately addressed my concerns. In particular, the rebuttal provides a much clearer justification of the method and helps resolve the key questions I raised during the initial review. The response strengthens my confidence in the paper and supports my original assessment, but it does not substantially change my overall evaluation of the work relative to the bar for a higher score. I therefore maintain my original score of 4.

---

> > > ### Author Response · Authors · 2026-04-04
> > >
> > > Thank you very much for your careful reading and valuable comments.

---

### Official Review · Reviewer_q2Rj · 2026-03-05

**Soundness:** 3
**Presentation:** 3
**Significance:** 2
**Originality:** 3
**Overall Recommendation:** 4
**Confidence:** 3

**Summary:**

This paper focuses on solving the problem of imbalanced domain generalization by addressing both domain shift and label shift simultaneously. It points out that existing domain generalization methods usually only concentrate on aligning the input distribution, while under class imbalance, they ignore the consistency of the posterior distribution, causing the decision boundary to be dominated by majority classes and thus degrading the performance of minority classes; therefore, this paper proposes to push away negative samples and enlarge the decision boundary to protect minority classes from being overwhelmed. The paper introduces a negative-guided contrastive learning framework consisting of three loss functions: L_con, which uses mixup-augmented negative samples and pushes away neighboring negatives to enlarge the decision space for minority samples; L_ce, which enhances intra-class compactness; and L_const, which pulls the prototypes of the same class from different domains closer and pushes apart those of different classes to promote posterior consistency.

**Compliance With Llm Reviewing Policy:**

Affirmed.

**Final Justification:**

Because the authors clarified the concerns about novelty and added experiments at a larger-scale dataset, based on these two points, I am currently leaning toward a weak accept. However, I feel the work still does not reach a strong level of novelty.

**Key Questions For Authors:**

1）Please provide more evidences for the novelty of this submission. 2) Please provide experimental resuls on large-scale datasets.

**Limitations:**

Please add specific discussions about the limitations of the proposed method.

**Strengths And Weaknesses:**

Strengths: 1) This paper adapts contrastive learning for the imbalanced domain generalization (IDG) setting by shifting its focus from simply pulling positive samples together to mainly pushing away hard neighboring negatives. It also uses hard negative sample synthesis in the prediction space to increase boundary pressure, and aligns cross-domain class prototypes to make the prediction distributions of the same category more consistent across different domains. In this way, it simultaneously improves the margin and posterior consistency. 2) The motivation of the paper is clear, and the objectives and functions of the three loss functions are well-defined.

Weaknesses：1） The method proposed in this paper is somewhat lacking in innovation, and is more like a recombination and adaptation of existing modules. The so-called negative-guided mechanism is essentially a variant of hard negative mining in contrastive learning.
2） The paper lacks experiments conducted on large-scale datasets.

---

> ### Author Rebuttal · Authors · 2026-03-31
>
> We sincerely thank the reviewer for their positive feedback and recognition of our work's contributions in **clear motivation** and **well-defined objective design**.
>
> > **[W1] The method proposed in this paper is somewhat lacking in innovation, and is more like a recombination and adaptation of existing modules.**
>
> **[AW1]**: Thank you for your comment.
> While our framework draws inspiration from existing contrastive learning techniques, we respectfully clarify that it is **not a simple recombination or a standard variant of hard negative mining**.
> Instead, it represents a fundamental redesign tailored specifically for IDG, supported by rigorous theory.
>
> *1. Theoretical Contributions*
> First, our study establishes **a rigorous generalization bound tailored specifically for IDG**, filling a critical theoretical gap in current literature.
> Second, we provide **an in-depth gradient analysis** of our proposed objective $L_{con}$, which is discussed in detail below.
>
> *2. Methodological Novelty (beyond hard negative mining)*
> Our loss functions are not arbitrarily assembled; rather, they form a unified structure derived directly from the proposed generalization bound.
>
> More importantly, as detailed in Appendix B (Page 14), our negative-dominant contrastive objective $L_{con}$ induces **a qualitatively different optimization behavior**, shifting from pulling positives to
> **adaptively re-weighting hard negative boundaries**.
> This mechanism is not captured by existing contrastive objectives and is particularly critical for IDG, thereby **being a non-trivial discovery**.
>
> Furthermore, unlike hard negative mining [1], which operates primarily as a heuristic sample selection strategy, our $L_{con}$ is designed to **adaptively repel negative samples intrinsically through its optimization objective**.
> Although we incorporate a hard negative augmentation scheme, it **serves purely as an orthogonal training enhancement**.
> As demonstrated in our ablation study (Tab. 4, Page 8), even without this augmentation (NoAug), $L_{con}$ (Baseline vs NoCon vs NoAug) still yields significant improvements, particularly on minority classes.
> This isolates the source of our performance gain, proving that **the effectiveness stems fundamentally from the novel loss design itself rather than auxiliary sampling techniques**.
>
> *3. Empirical Evidence of Non-triviality.*
> If the proposed method were merely a minor variation, its performance would likely mirror that of existing approaches.
> However, as shown in Tab. 8 (Page 24), models utilizing prior contrastive methods instead of our $L_{con}$ **consistently underperform in challenging IDG settings**, especially on minority classes.
> In contrast, our proposed formulation achieves significant and stable improvements, **empirically validating the necessity and novelty of our structural design**.
>
> > **[W2] The paper lacks experiments conducted on large-scale datasets.**
>
> **[AW2]**: Thank you for raising this point. We provide additional results on a large-scale real-world dataset, DomainNet, which is widely recognized as a realistic and challenging benchmark due to its diverse data sources.
>
> Tab. R1 and R2 report these results on DomainNet under two different settings, respectively.
> From these results, we observe that 1) our NDCL consistently yields the highest average accuracy across both settings, demonstrating a clear overall advantage. 2) On minority classes, it achieves top-tier performance, ranking first under the Duality setting and second under the TotalHeavyTail setting, slightly behind BSoftmax.
> Together, these results indicate that our method remains effective on large-scale datasets.
>
> Tab R1: *Overall accuracy of representative methods on DomainNet under TotalHeavyTail*
>
> ||Avg|Many|Medium|Few|
> |-|-|-|-|-|
> |ERM|24.7$\pm$0.6|57.2$\pm$0.7|50.2$\pm$0.3|15.0$\pm$0.1|
> |TRCI|25.4$\pm$0.5|59.0$\pm$0.9|51.9$\pm$0.6|15.2$\pm$0.4|
> |BSofmax|*26.4$\pm$0.5*|45.3$\pm$0.4|45.6$\pm$0.2|**19.4$\pm$0.2**|
> |BoDA|25.9$\pm$0.4|**59.7$\pm$0.5**|*52.8$\pm$0.3*|15.6$\pm$0.1|
> |PgCL|25.7$\pm$0.2|*59.6$\pm$0.1*|**53.9$\pm$0.2**|15.5$\pm$0.5|
> |*Ours*|**27.9$\pm$0.3**|54.4$\pm$0.5|52.2$\pm$0.4|*18.6$\pm$0.2*|
>
> Tab R2: *Overall accuracy of representative methods on DomainNet under Duality*
>
> ||Avg|Many|Medium|Few|
> |-|-|-|-|-|
> |ERM|24.6$\pm$0.2|48.5$\pm$0.6|41.1$\pm$0.5|19.3$\pm$0.1|
> |TRCI|25.9$\pm$0.2|*49.5$\pm$0.7*|*41.8$\pm$0.7*|20.7$\pm$0.3|
> |BSofmax|26.2$\pm$0.6|41.8$\pm$1.3|37.3$\pm$0.7|*22.7$\pm$0.5*|
> |BoDA|*26.6$\pm$0.2*|**53.8$\pm$0.5**|**44.7$\pm$0.2**|21.0$\pm$0.2|
> |PgCL|26.5$\pm$0.3|47.7$\pm$0.2|38.1$\pm$0.4|22.1$\pm$0.3|
> |*Ours*|**27.3$\pm$0.2**|48.6$\pm$0.2|40.4$\pm$0.8|**23.0$\pm$0.1**|
>
>
> [1] Kalantidis, Yannis, et al. Hard negative mixing for contrastive learning. NeurIPS. 2020.

---

> > ### Author Rebuttal · Reviewer_q2Rj · 2026-04-01
> >
> > Since most of my feedback received serious responses, I decided to raise my rating to honor my commitment to AC at that time. However, the information provided in this dataset was not particularly detailed, and I hope subsequent versions will offer more specifics.

---

> > > ### Author Response · Authors · 2026-04-04
> > >
> > > Thank you very much for your careful reading and increasing score.

---

### Official Review · Reviewer_n7JT · 2026-03-12

**Soundness:** 3
**Presentation:** 3
**Significance:** 3
**Originality:** 3
**Overall Recommendation:** 4
**Confidence:** 3

**Summary:**

This paper addresses Imbalanced Domain Generalization (IDG), where domain shift and label shift occur simultaneously. The authors provide a generalization bound for IDG highlighting posterior discrepancy and decision margin, and propose NDCL — a negative-dominant contrastive learning framework that exploits the relative balance of negatives under label imbalance. Experiments on three benchmarks against 23 baselines show consistent improvements.

**Compliance With Llm Reviewing Policy:**

Affirmed.

**Final Justification:**

My concerns have been adequately addressed, so I maintain my original positive score.

**Key Questions For Authors:**

1. Could the authors further clarify how each component of NDCL contributes to tightening the bound terms $\zeta$, $\eta$, and $\lambda_\pi$?
2. The results occasionally show slightly lower performance on "Many" classes. Is this an inherent majority–minority trade-off in the negative-dominant objective, and can it be controlled?
3. How does the quality of the prediction prototypes $\mu_k^d$ affect performance in domains where minority classes have very few samples?

**Limitations:**

The paper discusses several limitations related to domain prior modeling. It would also be helpful to further discuss (1) the generalizability of the approach beyond vision benchmarks and (2) potential majority–minority accuracy trade-offs introduced by the negative-dominant objective.

**Strengths And Weaknesses:**

Strengths:
1. The paper derives a generalization bound for IDG that explicitly highlights posterior discrepancy and decision margin.
2. The paper presents a clear narrative from theory to method, integrating contrastive learning, re-weighted cross-entropy, and prototype alignment in a unified objective.
3. Experiments span three benchmarks with multiple imbalance settings and 23 baselines, further supported by Pearson correlation and Friedman tests.
4. The negative-dominant contrastive objective provides a reasonable modification of the InfoNCE formulation by leveraging the relative abundance of negative samples under long-tailed distributions.

Weaknesses:
1. The connection between bound terms ($\zeta$, $\eta$, $\lambda_\pi$) and the actual training objectives could be more explicit.
2. The negative-balance assumption is not extensively verified beyond the considered benchmarks.
3. Several individual components build upon existing techniques in contrastive learning and imbalance learning; clearer positioning relative to prior contrastive learning and IDG methods would strengthen the contribution.
4. A citation formatting error appears in Appendix A ("(?)Thm. 2]ben2010theory"), which should be corrected before the final version.

---

> ### Author Rebuttal · Authors · 2026-03-31
>
> We sincerely thank the reviewer for their positive feedback and recognition of our work's contributions in **theoretical grounding**, **well-defined objective design**, and **solid experimental validation**.
>
> > **[W1 & Q1] The connection between bound terms ($\zeta$, $\eta$, $\lambda_\pi$) ...**
>
> **[AW1 & AQ1]**: Thank you for your comment.
>
> As defined in Thm. 1, $\zeta$ and $\eta$ denote the prior and posterior divergence, respectively, between the target distribution and its nearest projection onto the space spanned by the source domains.
> According to [1], these terms vanish if the target lies within this spanned space; otherwise, they characterize the **intrinsic generalization error** that cannot be eliminated by any hypothesis.
> Similarly, $\lambda_\pi$ represents the ideal joint error, reflecting the **fundamental compatibility** between domains.
> Consequently, $\zeta$, $\eta$, and $\lambda_\pi$ are inherently intractable and act as constants in our optimization.
>
> > **[W2] The negative-balance assumption is not extensively verified beyond the considered benchmarks.**
>
> **[AW2]**: We clarify that our negative-balance perspective is grounded in the fundamental statistical properties of long-tailed learning, rather than being a dataset-specific assumption.
>
> In long-tailed distributions, minority classes inherently lack sufficient positive samples for traditional feature "pulling."
> Conversely, negative samples from majority classes are universally abundant.
> Therefore, adaptively leveraging the "pushing" force of these negatives is structurally necessary to establish robust decision boundaries for tail classes.
>
> While exhaustive cross-domain verification is practically impossible, our evaluation spans highly diverse and challenging benchmarks (particularly DomainNet, cf. Reviewers rdwX and q2Rj).
> Our consistent effectiveness under various rigorous IDG settings provides strong empirical evidence that this mechanism generalizes well across complex visual scenarios.
>
> > **[W3] Several individual components build upon existing techniques in contrastive learning ...**
>
> **[AW3]**: Thank you for the constructive feedback.
> While our components draw inspiration from existing fields, their integration and optimization dynamics are fundamentally different: 1) Compared to prior contrastive learning, our $L_{con}$ is specifically designed to adaptively re-weight hard negative boundaries rather than just pulling positives (detailed in Lines 194-219 and Appendix B). 2) Compared to IDG methods, our NDCL is theoretically guided by a rigorous generalization bound rather than heuristic adjustments (discussed in Lines 925-939).
> We will further highlight these contributions in the revised version.
>
> > **[W4] A citation formatting error**
>
> **[AW4]**: Thank you for catching this oversight. We have corrected the citation formatting error in the revised manuscript.
>
> > **[Q2] Is this an inherent majority–minority trade-off in the negative-dominant objective, and can it be controlled?**
>
> **[AQ2]**: We would like to clarify that the observed performance drop on Many classes reflects a common majority–minority trade-off, which is not unique to our method but widely observed in imbalance-aware learning approaches, such as BSoftmax, SAMALTDG. Standard DG methods achieve higher majority accuracy simply because they lack imbalance-aware designs, inherently biasing the classifier toward majority classes. Furthermore, this trade-off can be explicitly controlled in our NDCL by adjusting the hyperparameter $\alpha$ in Eq. (5).
>
> > **[Q3] How does the quality of the prediction prototypes...**
>
> **[AQ3]**: Thank you for this insightful question.
> We acknowledge that the quality of prediction prototypes naturally impacts performance, but our NDCL remains highly robust against extreme data scarcity.
>
> Under the TotalHeavyTail setting on OfficeHome, approximately 40% of the classes contain fewer than 2 samples per source domain.
> Although such scarcity inevitably limits accurate prototype estimation, our NDCL still maintains highly competitive performance on minority classes (Tabs. 2, 3, and Fig. 4).
> Conversely, under the Duality setting, where data allows for better prototype quality, our NDCL achieves SOTA results on minority classes.
>
> Furthermore, to explicitly verify stability, Tab. R1 reports the step-wise prototype similarity specifically for minority classes ($\le 2$ samples) under TotalHeavyTail.
> The rapid convergence and high stability (>0.98) demonstrate that our updating mechanism effectively prevents erratic fluctuations, ensuring stable optimization even with minimal samples.
>
> Tab R1: *Step-wise prototype similarity. Headers: steps ($\times 100$).*
>
> |3|6|9|12|15|18|21|24|27|30|33|36|39|42|45|48|50|
> |-|-|-|-|-|-|-|-|-|-|-|-|-|-|-|-|-|
> |0.14|0.93|0.96|0.99|0.99|0.99|0.99|0.99|0.98|0.99|0.98|0.98|0.99|0.98|0.99|0.99|0.99|
>
> [1] Lu, Wang, et al. Towards Optimization and Model Selection for Domain Generalization: A Mixup-guided Solution. SDM. 2024.

---

> > ### Author Rebuttal · Reviewer_n7JT · 2026-04-05
> >
> > I thank the authors for their detailed and thoughtful rebuttal. While the response has adequately addressed my concerns, it does not alter my overall evaluation with respect to a higher score. I maintain my original score of 4.

---

### Official Review · Reviewer_rdwX · 2026-03-13

**Soundness:** 2
**Presentation:** 2
**Significance:** 2
**Originality:** 2
**Overall Recommendation:** 2
**Confidence:** 3

**Summary:**

This study addresses the complex challenge of Imbalanced Domain Generalization (IDG), which involves entangled domain and label shift across heterogeneous long-tailed distributions. To overcome the lack of theoretical foundations in this area, the authors first mathematically establish a generalization bound for IDG based on posterior discrepancy and decision margin, motivating a novel approach to directly steer decision boundaries. Building on this theoretical observation, they propose Negative Dominant Contrastive learning (NDCL)  to enhance discriminability and enforce posterior consistency across domains. Specifically NDCL achieves inter-class separation by emphasizing negative samples to naturally amplify minority-class gradients, encorages intra-class compactness through a reweighted cross-entropy strategy, and ensures cross-domain posterior consistency via prediction-central alignment.

**Compliance With Llm Reviewing Policy:**

Affirmed.

**Key Questions For Authors:**

Are there any relevant methodologies published after 2025 that could be included for comparison? In my humble opinion, the current baselines appear to focus predominantly on class imbalance, including LDAM, alone. A deeper analysis and justification regarding this aspect of the evaluation are required.

**Limitations:**

Please refer to the Key question and Weaknesses part.

**Strengths And Weaknesses:**

## S1. Theoretical Observation
The paper establishes a strong theoretical grounding by deriving a generalization bound for imbalanced domain generalization based on H-divergence.

## S2. Methodological Novelty.
The proposed NDCL framework presents a novel and technically sound solution. It successfully translates the theoretical insights into practice by enhancing discriminability while enforcing posterior consistency across diverse domains.

## S3. Empirical validation.
The effectiveness of NDCL is thoroughly demonstrated through various benchmarks. The evaluation covering various baselines and shows the proposed model's practical utility.

## W1. Outdated baselines
Even though the paper conducts extensive comparative experiments, the selected baselines appear to be largely outdated. Specifically, the compared methods are mostly limited to those published up to 2024, lacking evaluation against the most recent state-of-the-art approaches. This makes it difficult to ascertain the framework's superiority in the current recent landscape.

## W2. Lack of real evaluation.
The empirical validation currently relies heavily on benchmark datasets constructed under specific or artificially induced settings. To convincingly demonstrate the practical applicability of the proposed method, it is required to include additional experiments using actual real-world datasets.

---

> ### Author Rebuttal · Authors · 2026-03-31
>
> We sincerely thank the reviewer for their feedback and recognition of our work's contributions in **theoretical grounding**, **methodological novelty**, and **practical effectiveness**.
>
> > **[W1]: Outdated baselines**
>
> **[AW1]**: Thank you for raising this point.
>
> We would like to clarify that our evaluation is already comprehensive with respect to the existing literature on imbalanced domain generalization (IDG).
> In fact, the number of relevant IDG methods **remains relatively limited**, and **our comparisons cover the vast majority of existing approaches**, rather than focusing on a narrow or outdated subset.
>
> More importantly, the perceived gap stems from a difference in problem scope.
> Prior IDG works mainly focus on **intra-domain class imbalance**, whereas our study additionally investigates **inter-domain imbalance**, which we show (Appendix E.2) can also affect generalization.
> To examine this, we introduce two settings, TotalHeavyTail and Duality.
> The inclusion of standard DG and tailored imbalance methods is intended as a complementary analysis, providing a broader reference to assess how different paradigms behave under more extreme conditions.
> Notably, the results suggest that existing IDG methods remain suboptimal in such regimes (Lines 300-318), highlighting the necessity of our study.
>
> Finally, we note that one very recent work, PgCL [1], was not identified during our initial benchmarking process.
> We will incorporate it in the revised version for completeness.
> **Additional results on the large-scale dataset are provided in the next response**.
>
> > **[W2]: Lack of real evaluation**
>
> **[AW2]**: Thank you for raising this point. We provide additional results on a large-scale real-world dataset, DomainNet, which contains over 600,000 images and is widely recognized as a realistic and challenging benchmark due to its diverse data sources.
>
> Tab. R1 and R2 report these results on DomainNet under two different settings, respectively.
> Note that PgCL is newly included in the comparison.
> From these results, we observe that 1) our NDCL consistently yields the highest average accuracy across both settings, demonstrating a clear overall advantage. 2) On minority classes, it achieves top-tier performance, ranking first under the Duality setting and second under the TotalHeavyTail setting, slightly behind BSoftmax.
> Together, these results indicate that our method remains effective on large-scale datasets.
>
> Tab R1: *Overall accuracy of representative methods on DomainNet under TotalHeavyTail*
>
> |         | Avg          | Many         | Medium       | Few          |
> |---------|--------------|--------------|--------------|--------------|
> | ERM     | 24.7$\pm$0.6 | 57.2$\pm$0.7 | 50.2$\pm$0.3 | 15.0$\pm$0.1 |
> | TRCI    | 25.4$\pm$0.5 | 59.0$\pm$0.9 | 51.9$\pm$0.6 | 15.2$\pm$0.4 |
> | BSofmax | *26.4$\pm$0.5* | 45.3$\pm$0.4 | 45.6$\pm$0.2 | **19.4$\pm$0.2** |
> | BoDA    | 25.9$\pm$0.4 | **59.7$\pm$0.5** | *52.8$\pm$0.3* | 15.6$\pm$0.1 |
> | PgCL   | 25.7$\pm$0.2 | *59.6$\pm$0.1* | **53.9$\pm$0.2** | 15.5$\pm$0.5 |
> | *Ours*  | **27.9$\pm$0.3** | 54.4$\pm$0.5 | 52.2$\pm$0.4 | *18.6$\pm$0.2* |
>
>
> Tab R2: *Overall accuracy of representative methods on DomainNet under Duality*
>
> |         | Avg          | Many         | Medium       | Few          |
> |---------|--------------|--------------|--------------|--------------|
> | ERM     | 24.6$\pm$0.2 | 48.5$\pm$0.6 | 41.1$\pm$0.5 | 19.3$\pm$0.1 |
> | TRCI    | 25.9$\pm$0.2 | *49.5$\pm$0.7* | *41.8$\pm$0.7* | 20.7$\pm$0.3 |
> | BSofmax | 26.2$\pm$0.6 | 41.8$\pm$1.3 | 37.3$\pm$0.7 | *22.7$\pm$0.5* |
> | BoDA    | *26.6$\pm$0.2* | **53.8$\pm$0.5** | **44.7$\pm$0.2** | 21.0$\pm$0.2 |
> | PgCL   | 26.5$\pm$0.3 | 47.7$\pm$0.2 | 38.1$\pm$0.4 | 22.1$\pm$0.3 |
> | *Ours*  | **27.3$\pm$0.2** | 48.6$\pm$0.2 | 40.4$\pm$0.8 | **23.0$\pm$0.1** |
>
> > **[Q1]: Are there any relevant methodologies published...**
>
> **[AQ1]**: *Our NDCL outperforms the very recent IDG method, PgCL.*
>
> As discussed in [AW1], we investigate two extreme IDG settings and include both standard DG methods and tailored imbalance approaches to provide a broader evaluation.
> The justification for this selection is to serve as a complementary analysis, revealing the limitations of IDG methods against traditional paradigms under extreme settings.
>
> The revision will include a deeper analysis of baseline behaviors. For example, 1) TRCI offers basic stability but lacks explicit minority constraints. 2) SAMALTDG struggles compared to PGrad, as finding flat minima for minority classes under domain shift may be highly noisy. 3) Crucially, feature-level alignment (BoDA) fails due to severe cross-domain feature distortion, whereas classifier-level adjustments (BSoftmax & Ours) remain highly robust. This validates our strategy of operating within the stable prediction space.
>
> [1] Wang, M. et al.Probability-Guided Contrastive Learning for Long-Tailed Domain Generalization. IEEE Trans on Big Data. 2026.

---

> > ### Author Rebuttal · Reviewer_rdwX · 2026-04-04
> >
> > Thank you for the detailed response.
> >
> > While some of my concerns have been addressed, I would like to clarify a few remaining points. First, regarding the real-world benchmark, my intention was to inquire about the existence of benchmarks where both domain generalization and class imbalance issues occur simultaneously, rather than an empirical setting for transferring from one setting to the other. For example, as far as I understand, DomainNet is designed to evaluate domain generalization but does not inherently account for imbalance in domain distribution. The reason this point needs further clarification is that if such dataset (incorporating both challenges) does not exist, it implies that the scope for performance comparison is limited. Furthermore, it could be interpreted that scenarios where both issues coexist are themselves quite rare or restricted.
> >
> > Second, the current baseline for class imbalance (2017~2022) are somewhat outdated. While the authors propose PgCL as a new baseline, it is important to note that various class imbalance algorithms have evolved significantly. These methods may already possess inherent domain generalization capabilities. Therefore, a direct comparison with these algorithms is essential to isolate and measure the fundamental contribution of this work.
> >
> > Therefore, I maintain that these concerns need to be fully resolved. I believe that addressing these points in the next step of the review process will significantly help solidify the positioning of this research and clarify its unique value.

---

> > > ### Author Response · Authors · 2026-04-04
> > >
> > > Thank you for your comments.
> > >
> > > First, we would like to clarify that DomainNet is a real-world dataset inherently containing both domain shift and significant class imbalance, **as illustrated in Fig. 1 of [1] and [2], respectively**. Beyond benchmark datasets, this dual challenge can also arise in real-world scenarios. For instance, as discussed in our Introduction (Lines 34-36), specialized hospitals often serve distinct patient populations, which may simultaneously induce domain shifts and imbalanced label distributions across institutions. This suggests that the coexistence of these issues is a plausible and practically relevant setting, rather than a purely restricted case.
> > >
> > > Second, we would like to clarify that our use of classic class-imbalance baselines aims for a broader analysis rather than to directly compete with the latest state-of-the-art imbalance methods. Crucially, this led to **a key discovery overlooked in existing IDG literature**: classic methods can still excel under specific conditions. For example, BSoftmax dominates the "Few" categories under the TotalHeavyTail setting, while its advantage diminishes under the Duality setting. **We analyze the potential reasons behind this phenomenon in lines 325-329**, hypothesizing that the TotalHeavyTail setting might be heavily dominated by the long-tail issue, which effectively overshadows the domain shift. This observation raises an important question: under what specific conditions are specialized IDG methods strictly necessary? **Since our primary contribution lies in revealing this phenomenon and its boundaries**, an exhaustive evaluation of the latest class imbalance algorithms falls outside our current scope. We will explicitly highlight this open question as an important direction for future work.
> > >
> > > [1] Wang, M. et al. Probability-Guided Contrastive Learning for Long-Tailed Domain Generalization. IEEE Trans on Big Data. 2026.
> > > [2] Su H, et al. Sharpness-Aware Model-Agnostic Long-Tailed Domain Generalization. AAAI. 2024.

---

### Decision · Program_Chairs · 2026-04-30

**Decision:**

Accept (regular)

**Comment:**

This paper studies the problem of Imbalanced Domain Generalization. The authors propose a theoretical analysis establishing a generalization bound based on posterior discrepancy and decision margin. Then, they derive an algorithm called Negative Dominant Contrastive learning (NDCL) to enhance discriminability and enforce posterior consistency across domains. This algorithm achieves inter-class separation by focusing on negative samples to to give more importance to minority-class gradients, intra-class compactness is controlled through a reweighted cross-entropy strategy. Experiments are done on 3 benchmarks (VLCS, PACS, and OfficeHome) comparing more than 20 baselines.

Reviewers have identified the following strengths:
-Sound theoretical analysis/generalization bound
-Clear motivation/narrative story, the paper addresses a practical problem.
-Novelty of the methodology, NDCL provides a reasonable modification of the InfoNCE formulation
-Convincing empirical validation.


They also mentioned the following weaknesses:
-Not enough explanations on the connections between bound terms and the training objectives
-The negative-balance assumption is not extensively verified beyond the considered benchmarks.
-The positioning with respect to SOTA should be more precise, the paper may appear like a recombination and adaptation of existing methods (the negative-guided mechanism is essentially a variant of hard negative mining in contrastive learning)
-Outdated baselines (published up to 2024)
-No real world/large-scale dataset in the experiments
-method subject to high variance and knowledge of domain indices


During rebuttal, authors have provided detailed answers to rebuttal.  They add additional experimental results on a large dataset with the DomainNet benchmark and an experiment on the stability of prototypes. They precise some crucial and original aspects on their method and their positioning with respect to SOTA.

During the final evaluation, the reviewers made the final feedbacks:
-reviewer rdwX maintained his rejection score mainly due to the use of outdated baselines and lack of more real benchmarks for addressing imbalanced domain generalization.
-reviewer n7JT indicated that his concerns have been adequately addressed, he maintained his weak accept evaluation but did not want to go higher.
-reviewer q2Rj was satisfied by the answers on his concerns about novelty and the addition of experiments at a larger-scale dataset, and move toward a weak accept. However, he mentioned that he thinks that this work still does not reach a strong level of novelty.
-reviewer uVxV indicated that authors have adequately addressed his concerns, he mentioned that the paper is well-structured and clearly written with a theoretical generalization bound and rigorous experimental design. However, he was not included to give a higher score.

Overall the paper has some weaknesses that limit his scope. However, I retain that 3 reviewers were positives, 2 indicated that the theoretical aspect is strong strength (reviewers rdwX and uVxV). This work can be useful to an important fraction of the ICML community.
I propose then accept.